# Density-Aware Farthest Point Sampling

**Paolo Climaco**[*]    *climacopaolo@gmail.com*
*Institut für Numerische Simulation, Universität Bonn, Germany*
*Department of Mathematics, University of California, Los Angeles, USA*

**Jochen Garcke**    *garcke@ins.uni-bonn.de*
*Institut für Numerische Simulation, Universität Bonn, Germany*
*Fraunhofer SCAI, Sankt Augustin, Germany*

**Reviewed on OpenReview:** *https://openreview.net/forum?id=vI47lgIfYc*

## Abstract

We focus on training machine learning regression models in scenarios where the availability of labeled training data is limited due to computational constraints or high labeling costs. Thus, selecting suitable training sets from unlabeled data is essential for balancing performance and efficiency. For the selection of the training data, we focus on passive and model-agnostic sampling methods that only consider the data feature representations. We derive an upper bound for the expected prediction error of Lipschitz continuous regression models that linearly depends on the weighted fill distance of the training set—a quantity we can estimate simply by considering the data features. We introduce "Density-Aware Farthest Point Sampling" (DA-FPS), a novel sampling method. We prove that DA-FPS provides approximate minimizers for a data-driven estimation of the weighted fill distance, thereby aiming at minimizing our derived bound. We conduct experiments using two regression models across three datasets. The results demonstrate that DA-FPS significantly reduces the mean absolute prediction error compared to other sampling strategies.

## 1 Introduction

Machine learning regression aims to predict continuous numerical values based on input features by learning the underlying prediction patterns from labeled training data. Data quality and data selection are crucial for developing and deploying effective regression models. Without sufficient or reliable data, ML regression models cannot perform well: too little data leads to poor learning, and biased or corrupted data results in inaccurate predictions. This work addresses challenges related to the limited availability of labeled training data, often resulting from computational constraints and high labeling costs. These challenges frequently occur in scientific applications, where labeling typically relies on expensive numerical simulations or laboratory experiments. Our work is motivated by molecular property prediction. The chemical compound space is large. It is estimated to contain over $10^{60}$ molecules (Kirkpatrick & Ellis, 2004). Its exhaustive exploration using classical labeling methods based on quantum-mechanics simulation is impractical. Machine learning regression offers a promising solution by leveraging small, selectively labeled datasets to predict molecular properties at scale (von Lilienfeld et al., 2020). This data-efficient approach has the potential to accelerate the discovery of new drugs and materials.

We consider scenarios where we have access to a large pool of unlabeled data, e.g., molecules for which chemical and physical properties (the labels) are unknown and costly to obtain. We aim to select and label a small subset of the data pool to train a regression model. We show that by selecting suitable training sets we can positively impact the prediction performance of ML regression models. The goal is to improve the average prediction performance of the model on the points in the pool not selected for training. We focus on label-agnostic, passive, and model-agnostic sampling, that is, selection approaches that solely rely

---

[*]Work done while at Universität Bonn. Currently at University of California, Los Angeles.

on the data feature representations, do not consider any active learning procedure, and do not assume any specific structure for the regression model. On the one hand, this promotes the reusability of the labeled samples, ensuring labeling efforts are not wasted on subsets useful only for specific models or tasks. On the other hand, developing such approaches is a non-trivial challenge because we can rely only on very little information, that is, the data feature representations and some assumption on the regularity of the learned function, but not on the model used to learn it, e.g., using a neural network or a kernel approach. As an additional constraint, we are interested in sampling approaches that are computationally efficient and potentially can be used on data pools consisting of hundreds of thousands of high-dimensional data points, which is a realistic size for molecular property prediction tasks.

Training data selection for model performance optimization has been extensively studied. Relevant literature includes Active Learning (AL) (Settles, 2012; Ren et al., 2021) and experimental design (John & Draper, 1975; Yu et al., 2006). However, both focus on methodologies that optimize data selection by leveraging model-specific knowledge. Here, we focus on passive and model-agnostic sampling strategies. Passive sampling was first introduced in Yu & Kim (2010), where the authors highlight its importance in scenarios where labeled data can be difficult, time-consuming, or expensive to obtain. Drawing from Climaco & Garcke (2024), passive sampling strategies can be classified as model-dependent and model-agnostic. Model-dependent approaches optimize the data selection process for specific models or classes of models. We see single-shot batch active learners and experimental design approaches as passive and model-dependent methods. Model-agnostic selection approaches have the potential to benefit multiple classes of regression models, thereby enhancing reusability of costly data labelling.

The Farthest Point Sampling (FPS) is a greedy passive and model-agnostic sampling strategy that provides approximate solutions to the $k$-center problem by minimizing the fill distance of the selected set, which is the maximal distance between a point in the feature space of interest and its closest selected point. The benefits of minimizing the training set fill distance have been studied independently for classification (Sener & Savarese, 2018) and regression (Climaco & Garcke, 2024). In Sener & Savarese (2018) it was shown that minimizing the training set fill distance leads to increased accuracy by reducing the average classification error of Lipschitz continuous models with a softmax output layer. In Climaco & Garcke (2024) we show that for regression tasks minimizing the training set fill distance with the FPS significantly reduces the maximum error of the label predictions of Lipschitz continuous models. However, it was also shown that employing FPS to minimize the maximum prediction error is mostly beneficial in the low data regime and that in such low data regime it does not provide any significant advantage in terms of the average absolute prediction error compared to other passive sampling approaches that are model-agnostic.

This work addresses the limitations of the FPS for molecular property prediction tasks and aims to extend the work done in Climaco & Garcke (2024). In particular, we develop a passive and model-agnostic algorithm, inspired by the FPS greedy procedure and supported by a solid theoretical motivation, to select training sets that can improve the average prediction performance of regression models measured in terms of the mean absolute error (MAE).

We derive an upper bound for the expected prediction error of Lipschitz continuous regression models. Our bound is linear in a novel weighted fill distance, that is, the maximum weighted distance between a point in the feature domain of interest and its closest selected training point. The weights we use to scale the distances provide information on the distributions of the data in the feature space. We introduce a novel passive and model-agnostic data selection strategy called "Density-Aware Farthest Point Sampling" (DA-FPS). This strategy greedily selects sets from a pool of available data points with the aim of minimizing a data-driven estimation of the weighted fill distance of the selected set, thereby minimizing our proposed bound. The ultimate goal of DA-FPS is to select training sets that can enhance the average prediction performance of Lipschitz continuous regression models. We prove that DA-FPS provides approximate minimizers for a data-driven estimation of the weighted fill distance. Additionally, we report experimental results showing that selecting training sets using DA-FPS can decrease the average prediction absolute error of Lipschitz continuous regression models. Compared with other model-agnostic sampling techniques, including FPS, DA-FPS demonstrates its superiority for low training set budgets in terms of mean absolute error (MAE) reduction. We are not aware of any other passive sampling model-agnostic strategy supported by rigorous theoretical work that can improve the mean absolute prediction error of a regression model.

The paper is organized as follows: In section 2 we review related work. In section 3 we formalize the regression problem and define the optimization objective. In section 4 we derive an upper bound for the expected prediction error of Lipschitz continuous regression models that depends linearly on a weighted fill distance of the training set. In section 5-7 we introduce the DA-FPS algorithm and prove that it provides approximate minimizers for a data-driven estimation of the weighted fill distance. In section 8 we report experimental results showing that DA-FPS can reduce the mean absolute prediction error of regression models compared to other passive sampling model-agnostic strategies. In section 9 we summarize our contributions and outline future research directions. After the conclusion, Table 1 provides key notation.

## 2 Related work

There has been a significant research effort to develop approaches that can be used for training data selection aiming at model performance optimization. There is a large body of literature on active learning (AL). See Settles (2012) and Ren et al. (2021) for a review of such approaches. AL typically involves training regression models to predict uncertainties or estimate labels for unlabeled data, selecting the most relevant points for labeling, including them in the training set and repeating the process until stopping criteria are met. It optimizes the training set selection for a specific model or model class, and for a given task, by using computed labels to iteratively update model parameters during selection. Single-shot batch active learners select the training set in one step without iterative training or label knowledge. Examples include methods that minimize maximum mean discrepancy, discrepancy, and nuclear discrepancy, which are quantities that can be estimated solely relying on the data features (Chattopadhyay et al., 2012; Wang & Ye, 2015; Viering et al., 2019). However, these approaches require choosing a kernel function, and their effectiveness in reducing the prediction error is theoretically motivated only assuming that the regression function belongs to the reproducing kernel Hilbert space associated with the chosen kernel (Viering et al., 2019). That is, they optimize the selection for specific model classes, such as kernel regularized least squares, similar to active learning. Moreover, such approaches have been mostly studied for classification tasks. Assuming the knowledge of the learning model may even lead to the development of strategies that reflect some principle of optimality for the selection of the training set, as in the case of approaches developed in the field of experimental design (John & Draper, 1975; Yu et al., 2006).

In this work we focus on passive and model-agnostic sampling strategies. Coresets (Feldman, 2019) provide an effective set of tools for performing model-agnostic passive sampling. Uniform random sampling is considered the natural benchmark for all the other sampling strategies (Feldman, 2019). Importance sampling selects data points that are more important to the problem at hand based on importance weights determining the relevance of each data point. For instance, Xie et al. (2023) present an importance sampling approach for selecting suitable pretraining datasets for large language models. Cluster-based methods, such as $k$-medoids++ and $k$-means++ (Schubert & Rousseeuw, 2021), group similar points together and select representatives from each cluster. Greedy methods such as submodular function maximization algorithms (Krause & Golovin, 2014) start with an empty set and iteratively add the most influential point at each step, according to a given principle. Data twinning partitions the dataset into statistically similar twin sets by attempting to minimize the energy distance, a quantity that can be estimated solely by computing pairwise distances between data features. It has been effectively used for training data selection in regression (Vakayil & Joseph, 2022). In our earlier work Climaco & Garcke (2024) we show that selecting the training set by fill distance minimization using the Farthest Point Sampling (FPS) can reduce the maximum prediction error of Lipschitz continuous regression models. However, we also show that FPS does not offer any significant benefit in terms of the average absolute prediction error when compared to other benchmark passive sampling model-agnostic approaches.

## 3 Problem definition

In this section we formalize the setting of the regression problem we aim to address and define the optimization objective. Our setting is similar to the one introduced in Climaco & Garcke (2024). We consider a supervised regression problem where the feature space $\mathcal{X} \subset \mathbb{R}^d$ is bounded and the label space is $\mathcal{Y} \subset \mathbb{R}$. The goal is to learn a function $g : \mathcal{X} \to \mathcal{Y}$ from a function space $\mathcal{M}$, using a set of labeled data. Each function in $\mathcal{M}$ may

be parameterized by weights $\boldsymbol{w} \in \mathbb{R}^m$, depending on the learning algorithm. To evaluate prediction quality, we use an error function $l : \mathcal{X} \times \mathcal{Y} \times \mathcal{M} \to \mathbb{R}^+$, which measures the prediction error of a trained regression model for a given input feature and associated true label.

Let $\mathcal{D} := \{(\boldsymbol{x}_i, y_i)\}_{i=1}^n$ be a pool of available data points consisting of i.i.d. realizations of two random variables $\boldsymbol{X}$ and $Y$ taking value on $\mathcal{X}$ and $\mathcal{Y}$, respectively, with joint distribution $p_{\mathcal{D}} \in \mathcal{P} := \{p : \mathcal{X} \times \mathcal{Y} \to \mathbb{R}^+ | \int_{\mathcal{X} \times \mathcal{Y}} p(\boldsymbol{x}, y) d\boldsymbol{x} dy = 1\}$. We consider scenarios where the labels $\{y_i\}_{i=1}^n$ are unknown. We aim to select and label a training set $\mathcal{L} := \{(\boldsymbol{x}_{i_j}, y_{i_j})\}_{j=1}^b$ from $\mathcal{D}$, with $i_j \in \{1, \ldots, n\} \ \forall j$, and $b \ll n$. The objective of the training data selection is to maximize the average predictive performance of a regression model $m_{\mathcal{L}} \in \mathcal{M}$, trained on $\mathcal{L}$, on the new data points not selected for training. In the following, without loss of generality, we assume $\mathcal{L} := \{(\boldsymbol{x}_j, y_j)\}_{j=1}^b$ and define $\mathcal{L}_{\mathcal{X}} := \{\boldsymbol{x}_j\}_{j=1}^b$. Following along Wang & Ye (2015), we assume that the training data is selected according to a distribution $p_{\mathcal{L}} \in \mathcal{P}$ with $p_{\mathcal{D}} \neq p_{\mathcal{L}}$ and $p_{\mathcal{D}}(y|\boldsymbol{x}) = p_{\mathcal{L}}(y|\boldsymbol{x})$. In other words, the training data distribution may differ from the data source distribution, but the map connecting a data location $\boldsymbol{x} \in \mathcal{X}$ and its associated label value is independent on how the data is selected. Furthermore, we study a typical scenario involving molecular prediction tasks in which the data source distribution $p_{\mathcal{D}}$ is unknown, and we have only access to the locations $\mathcal{D}_{\mathcal{X}} := \{\boldsymbol{x}_i\}_{i=1}^n$ of the dataset $\mathcal{D}$. Moreover, we are interested in applications in which the labeling process is computationally expensive, therefore, given a budget $b \in \mathbb{N}$ of points to label, the goal is to select a subset $\mathcal{L}_{\mathcal{X}} \subset \mathcal{D}_{\mathcal{X}}$ and label it to obtain a subset $\mathcal{L} \subset \mathcal{D}$ that solves the following optimization problem:

$$\min_{\substack{\mathcal{L} \subset \mathcal{D}, \\ |\mathcal{L}| = b}} \mathbb{E}_{p_{\mathcal{D}}}[l(\boldsymbol{X}, Y, m_{\mathcal{L}})]. \tag{1}$$

In other words, we aim to sample and label a training set $\mathcal{L}$, of cardinality $b$, so that the expected error on the data distribution $p_{\mathcal{D}}$ associated with the function $m_{\mathcal{L}}$ is minimized.

The optimization problem addressed in Climaco & Garcke (2024) and the one we define in (1) are similar in the sense that both focus on minimizing an expected value of the error function by selecting a subset $\mathcal{L}$ from a set of available data points $\mathcal{D}$. The key distinction lies in their objectives: in Climaco & Garcke (2024), the goal is to minimize the maximum expected error over a finite set of known data locations. Differently, in (1), the objective is to minimize the average error, defined as the expected value of the error function over the data source distribution $p_{\mathcal{D}}$ on $\mathcal{X} \times \mathcal{Y}$.

## 4 Bound for the expected prediction error

In this section we develop our main theoretical contribution: an upper bound for the optimization objective in (1) that depends linearly on a novel weighted fill distance of the training set. We first introduce the concept of weighted fill distance of a set. We then establish assumptions and present our main theoretical result.

Since the optimization problem in (1) cannot be solved directly due to the unknown labels of the points in $\mathcal{D}$ and the unknown data source distribution $p_{\mathcal{D}}$, we derive an upper bound for its optimization objective. This bound identifies the weighted fill distance of the training set as a key quantity to minimize for better prediction performance, motivating our Density-Aware Farthest Point Sampling (DA-FPS) algorithm introduced in the next section. We define the weighted fill distance, a quantity we can associate with finite subsets of the feature space $\mathcal{X}$ selected from $p_{\mathcal{X}_{\mathcal{L}}}$, which is the marginal on $\mathcal{X}$ of a distribution $p_{\mathcal{L}} \in \mathcal{P}$, called training data distribution.

**Definition 4.1.** Consider $\mathcal{X} \subset \mathbb{R}^d$ bounded and $\mathcal{L}_{\mathcal{X}} = \{\boldsymbol{x}_j\}_{j=1}^b \subset \mathcal{X}$, set of data locations sampled from $p_{\mathcal{X}_{\mathcal{L}}}$, marginal on $\mathcal{X}$ of a distribution $p_{\mathcal{L}} \in \mathcal{P}$. Moreover, consider $p_{\mathcal{X}_{\mathcal{D}}}$ marginal on $\mathcal{X}$ of a distribution $p_{\mathcal{D}} \in \mathcal{P}$. We define the *weighted fill distance* of $\mathcal{L}_{\mathcal{X}}$ in $\mathcal{X}$ with respect to $p_{\mathcal{X}_{\mathcal{D}}}$ as

$$W_{\mathcal{L}_{\mathcal{X}}, \mathcal{X}}(p_{\mathcal{X}_{\mathcal{L}}} || p_{\mathcal{X}_{\mathcal{D}}}) := \sup_{\boldsymbol{x} \in \mathcal{X}} \min_{\boldsymbol{x}_j \in \mathcal{L}_{\mathcal{X}}} \|\boldsymbol{x} - \boldsymbol{x}_j\|_2 \psi_{\mathcal{L}_{\mathcal{X}}}(\boldsymbol{x}), \tag{2}$$

where $\| \cdot \|_2$ is the $L_2$-norm and the weight function $\psi_{\mathcal{L}_{\mathcal{X}}} : \mathcal{X} \to \mathbb{R}$ is defined as

$$\psi_{\mathcal{L}_{\mathcal{X}}}(\boldsymbol{x}) := \begin{cases} 1 - \frac{p_{\mathcal{X}_{\mathcal{L}}}(\boldsymbol{x})}{p_{\mathcal{X}_{\mathcal{D}}}(\boldsymbol{x})} & \text{if } p_{\mathcal{X}_{\mathcal{D}}}(\boldsymbol{x}) \neq 0, \\ 0 & \text{otherwise,} \end{cases} \tag{3}$$

with $p_{\mathcal{X}_{\mathcal{L}}}(\boldsymbol{x}) := \int_{\mathcal{Y}} p_{\mathcal{L}}(\boldsymbol{x}, y) dy$ and $p_{\mathcal{X}_{\mathcal{D}}}(\boldsymbol{x}) := \int_{\mathcal{Y}} p_{\mathcal{D}}(\boldsymbol{x}, y) dy$.

The weighted fill distance is a deterministic quantity depending on the densities $p_{\mathcal{X}_{\mathcal{L}}}(\boldsymbol{x})$ and $p_{\mathcal{X}_{\mathcal{D}}}(\boldsymbol{x})$, the weight function $\psi_{\mathcal{L}_{\mathcal{X}}}(\boldsymbol{x})$, the selected set $\mathcal{L}_{\mathcal{X}}$ and the distance metric. Here, we use the $L_2$-metric as it is commonly used in machine learning applications. The following results can be generalized to other metrics. The weighted fill distance is non-negative, as we show in Appendix A, and reduces to the standard fill distance when $\psi_{\mathcal{L}_{\mathcal{X}}}(\boldsymbol{x}) = 1$ for all $\boldsymbol{x} \in \mathcal{X}$. Our weight function $\psi_{\mathcal{L}_{\mathcal{X}}}(\boldsymbol{x}) = 1 - \frac{p_{\mathcal{X}_{\mathcal{L}}}(\boldsymbol{x})}{p_{\mathcal{X}_{\mathcal{D}}}(\boldsymbol{x})}$ assigns higher weights to regions where the marginal of training data distribution on $\mathcal{X}$ underrepresents the marginal of the data source distribution, i.e., where $p_{\mathcal{X}_{\mathcal{L}}}(\boldsymbol{x}) < p_{\mathcal{X}_{\mathcal{D}}}(\boldsymbol{x})$. Note that the weighted fill distance can be zero under different circumstances. This occurs, for instance, when $\psi_{\mathcal{L}_{\mathcal{X}}}(\boldsymbol{x}) = 0$ for all $\boldsymbol{x} \in \mathcal{X}$, or when any point in the non-zero support of $p_{\mathcal{X}_{\mathcal{D}}}$ is in $\mathcal{L}_{\mathcal{X}}$, that is when $\min_{\boldsymbol{x}_j \in \mathcal{L}_{\mathcal{X}}} \|\boldsymbol{x} - \boldsymbol{x}_j\|_2 = 0$ for all $\boldsymbol{x}$ such that $p_{\mathcal{D}_{\mathcal{X}}}(\boldsymbol{x}) > 0$. In the first case, the fact that $\psi_{\mathcal{L}_{\mathcal{X}}}(\boldsymbol{x}) = 0$ for all $\boldsymbol{x} \in \mathcal{X}$ implies that the marginals of the training and data source distributions are perfectly aligned, that is, $p_{\mathcal{X}_{\mathcal{L}}}(\boldsymbol{x}) = p_{\mathcal{X}_{\mathcal{D}}}(\boldsymbol{x})$ for all $\boldsymbol{x} \in \mathcal{X}$. This simply follows by how we defined the weight function. In the second case, the fact that each point in the non-zero support of $p_{\mathcal{X}_{\mathcal{D}}}$ is in $\mathcal{L}_{\mathcal{X}}$ implies that $\mathcal{L}_{\mathcal{X}}$ already contains all points associated with data locations where a point from the data source distribution could arise. That is, the relevant regions of the data space are already included in the selected set. In other words, the fact that the weighted fill distance is zero can indicate either perfect distributional alignment or full geometric coverage of regions of the data space where points from the data source distribution could arise, but not necessarily both. Thus, the weighted fill distance serves as a joint measure of geometric coverage and distributional matching between the training and data source marginals.

For our theoretical analysis, we use assumptions from Climaco & Garcke (2024), which we recapture in Appendix B.1 and B.2. In particular, we consider the assumptions to be valid also for $p_{\mathcal{D}}$ and $p_{\mathcal{L}}$. The first assumption states that given the data feature location $\boldsymbol{X} = \boldsymbol{x}_i$, the expected associated label value, $y_i$, is close to the conditional expected value of the random variable $Y$ at that location. That is, $\mathbb{E}\left[ |Y - \mathbb{E}[Y|\boldsymbol{x}_i]| \,\big|\, \boldsymbol{x}_i \right] \leq \epsilon$, with scalar $\epsilon > 0$. This assumption models scenarios where the true feature-label relationship is stochastic or deterministic but subject to random fluctuations with a magnitude parameterized by a fixed scalar $\epsilon$. Further, a Lipschitz continuity assumption relates to the smoothness of the map connecting $\mathcal{X}$ and $\mathcal{Y}$. That is, $\left| \mathbb{E}\left[Y|\hat{\boldsymbol{x}}\right] - \mathbb{E}\left[Y|\tilde{\boldsymbol{x}}\right] \right| \leq \lambda_p \|\hat{\boldsymbol{x}} - \tilde{\boldsymbol{x}}\|_2, \ \forall \, \hat{\boldsymbol{x}}, \tilde{\boldsymbol{x}} \in \mathcal{X}$, where $\lambda_p \in \mathbb{R}^+$. It implies that when two data points are closer in $\mathcal{X}$, their corresponding labels tend to be closer in $\mathcal{Y}$. The second assumption essentially states that the expected error on the training set is bounded. That is, we assume there exist $\epsilon_{\mathcal{L}} \geq 0$, depending on the labeled set $\mathcal{L} \subset \mathcal{D}$ and the trained regression model $m_{\mathcal{L}}$, such that for any labeled point $(\boldsymbol{x}_j, y_j) \in \mathcal{L}$ we have that $\mathbb{E}[l(\boldsymbol{x}_j, Y, m_{\mathcal{L}})|\boldsymbol{x}_j] \leq \epsilon_{\mathcal{L}}$. We also make standard regularity assumptions on the error and learned functions hold in the form of Lipschitz continuity. With that, we are able to establish an upper bound for the optimization objective in (1).

**Theorem 4.2.** *Consider random variables $(\boldsymbol{X}, Y)$ taking value on $\mathcal{X} \times \mathcal{Y} \subset \mathbb{R}^d \times \mathbb{R}$, with $\mathcal{X}$ bounded, data source distribution $p_{\mathcal{D}} \in \mathcal{P}$, labeled dataset $\mathcal{L} := \{(\boldsymbol{x}_j, y_j)\}_{j=1}^b$ arising from training data distribution $p_{\mathcal{L}} \in \mathcal{P}$, regression model $m_{\mathcal{L}} \in \mathcal{M}$ trained on $\mathcal{L}$, and error function $l : \mathcal{X} \times \mathcal{Y} \times \mathcal{M} \to \mathbb{R}^+$. If Assumptions B.1 and B.2 are fulfilled, then we have that*

$$\mathbb{E}_{p_{\mathcal{D}}}\left[l(\boldsymbol{X}, Y, m_{\mathcal{L}})\right] \leq C W_{\mathcal{L}_{\mathcal{X}}, \mathcal{X}}\left(p_{\mathcal{X}_{\mathcal{L}}} \| p_{\mathcal{X}_{\mathcal{D}}}\right) +$$
$$+ \Big( \underbrace{\lambda_{l_{\mathcal{Y}}} \epsilon}_{\substack{labels \\ uncertainty}} + \underbrace{\epsilon_{\mathcal{L}}}_{\substack{max \ error \\ training \ set}} \Big) \mathbb{P}_{\mathcal{L}}\left[p_{\mathcal{X}_{\mathcal{D}}} < p_{\mathcal{X}_{\mathcal{L}}}\right] + \underbrace{\mathbb{E}_{p_{\mathcal{L}}}\left[l(\boldsymbol{X}, Y, m_{\mathcal{L}})\right]}_{\substack{expected \ error \\ training \ distribution}}, \quad (4)$$

*where $\mathbb{P}_{\mathcal{L}}\left[p_{\mathcal{X}_{\mathcal{D}}} < p_{\mathcal{X}_{\mathcal{L}}}\right] := \int_{\mathcal{X}} p_{\mathcal{X}_{\mathcal{L}}}(\boldsymbol{x}) \cdot \mathbf{1}\left\{p_{\mathcal{X}_{\mathcal{D}}}(\boldsymbol{x}) < p_{\mathcal{X}_{\mathcal{L}}}(\boldsymbol{x})\right\} d\boldsymbol{x}$ quantifies distributional mismatch between $p_{\mathcal{X}_{\mathcal{D}}}$ and $p_{\mathcal{X}_{\mathcal{L}}}$. The indicator function $\mathbf{1}\left\{p_{\mathcal{X}_{\mathcal{D}}}(\boldsymbol{x}) < p_{\mathcal{X}_{\mathcal{L}}}(\boldsymbol{x})\right\}$ equals 1 when the condition $p_{\mathcal{X}_{\mathcal{D}}}(x) < p_{\mathcal{X}_{\mathcal{L}}}(x)$ is satisfied. $C := (\lambda_{l_{\mathcal{X}}} + \lambda_{l_{\mathcal{Y}}} \lambda_p)$. $\lambda_p$ Lipschitz constant and $\epsilon$ labels' uncertainty from Assumption B.1. $\lambda_{l_{\mathcal{X}}}$ and $\lambda_{l_{\mathcal{Y}}}$ are Lipschitz constants of the error function from Assumption B.2. $\epsilon_{\mathcal{L}}$ is the maximum error of $m_{\mathcal{L}}$ on the training set, from Assumption B.2. $W_{\mathcal{L}_{\mathcal{X}}, \mathcal{X}}\left(p_{\mathcal{X}_{\mathcal{L}}} \| p_{\mathcal{X}_{\mathcal{D}}}\right)$ is the weighted fill distance defined as in (2). Moreover, $p_{\mathcal{X}_{\mathcal{D}}}$ and $p_{\mathcal{X}_{\mathcal{L}}}$ are the marginals of $p_{\mathcal{D}}$ and $p_{\mathcal{L}}$, respectively, defined on $\mathcal{X}$.*

Theorem 4.2, proof in Appendix C, provides a qualitative upper bound for the expected value of the error function on the data source distribution $p_{\mathcal{D}}$ depending linearly on the weighted fill distance of the selected

training set. Note that if $p_{\mathcal{X}_{\mathcal{L}}}(\boldsymbol{x}) = p_{\mathcal{X}_{\mathcal{D}}}(\boldsymbol{x}) \ \forall \ \boldsymbol{x}$ in the non-zero support of $p_{\mathcal{X}_{\mathcal{D}}}$, the weighted fill distance and $\mathbb{P}_{\mathcal{L}}\left[p_{\mathcal{X}_{\mathcal{D}}} < p_{\mathcal{X}_{\mathcal{L}}}\right]$ would be zero and the bound reduces to the trivial but tight inequality $\mathbb{E}_{p_{\mathcal{X}_{\mathcal{D}}}}\left[l(\boldsymbol{X}, Y, m_{\mathcal{L}})\right] \leq \mathbb{E}_{p_{\mathcal{D}}}\left[l(\boldsymbol{X}, Y, m_{\mathcal{L}})\right]$. This shows that when the training and data source distributions are perfectly aligned, our bound reduces to the best achievable bound in this scenario.

The bound in (4) depends on five main quantities: the maximum error on the training set ($\epsilon_{\mathcal{L}}$), which we can only compute after training a regression model, the label uncertainty ($\epsilon$), the expected error on the training data distribution ($\mathbb{E}_{p_{\mathcal{L}}}\left[l(\boldsymbol{X}, Y, m_{\mathcal{L}})\right]$), which we may not know or be able to compute, $\mathbb{P}_{\mathcal{L}}\left[p_{\mathcal{X}_{\mathcal{D}}} < p_{\mathcal{X}_{\mathcal{L}}}\right]$ quantifying the mismatch between $p_{\mathcal{X}_{\mathcal{D}}}$ and $p_{\mathcal{X}_{\mathcal{L}}}$, and the weighted fill distance. The last two quantities do not depend on the data labels, contrary to the first three. The expected error on the training data distribution and the maximum error on the training set indicate that the bound also depends on how well the unknown trained model fits the training data distribution. We do not consider minimizing these quantities because we cannot estimate them without training a regression model. We are considering a passive and model-agnostic sampling scenario where we have no knowledge about the data labels and the regression model at the time of selection. In what follows, we work under the general assumption that the employed trained model works well on the training data and that these quantities are negligible. A similar assumption has been considered in previous work (Just et al., 2023). Under such assumptions, the smaller the weighted fill distance of the selected training set and $\mathbb{P}_{\mathcal{L}}\left[p_{\mathcal{X}_{\mathcal{D}}} < p_{\mathcal{X}_{\mathcal{L}}}\right]$, the smaller the bound for the expected approximation error on data from the source distribution $p_{\mathcal{D}}$.

In this work, we focus on minimizing the weighted fill distance of the selected set. We only consider minimizing the weighted fill distance for the following reasons: First, the weighted fill distance provides a quantification of distributional mismatch between the marginals of training and data source distributions by means of the weight function. If there is no distributional mismatch the fill distance is zero. Thus, minimizing the weighted fill distance is related to decreasing $\mathbb{P}_{\mathcal{L}}\left[p_{\mathcal{X}_{\mathcal{D}}} < p_{\mathcal{X}_{\mathcal{L}}}\right]$ as well. Second, by taking into account point distances, the weighted fill distance provides a more comprehensive measure that captures both geometric coverage and distributional alignment aspects. Third, it is more tractable to minimize one of the two quantities than to minimize both. Our goal is to design a data sampling strategy, inducing a training data distribution marginal $p_{\mathcal{X}_{\mathcal{L}}}$, to sample a set $\mathcal{L}_{\mathcal{X}}$ with minimal weighted fill distance.

## 5 Density-Aware Farthest Point Sampling (DA-FPS)

In this section we present our main algorithmic contribution: the DA-FPS method for selecting training sets aiming at minimizing the weighted fill distance. We first explain the core algorithmic idea. We then detail how we estimate the weighted fill distance from data. Specifically, we describe the density estimation approach using $k$-nearest neighbors and discuss the estimated weighted fill distance formulation. After that, we present the complete DA-FPS algorithm. The section concludes by addressing computational challenges, providing implementation details, and discussing key algorithmic choices.

DA-FPS is a data selection procedure aiming at minimizing the weighted fill distance of the selected set: $W_{\mathcal{L}_{\mathcal{X}}, \mathcal{X}}\left(p_{\mathcal{X}_{\mathcal{L}}} \| p_{\mathcal{X}_{\mathcal{D}}}\right)$. Unfortunately, given a set $\mathcal{L}_{\mathcal{X}} \subset \mathcal{X}$, its associated weighted fill distance cannot be directly computed as the weights explicitly depend on the marginal of the data source distribution $p_{\mathcal{X}_{\mathcal{D}}}$, which we consider to be unknown. Still, we have access to a set $\mathcal{D}_{\mathcal{X}} = \{\boldsymbol{x}_i\}_{i=1}^n$ of unlabeled data points arising from $p_{\mathcal{X}_{\mathcal{D}}}$. Thus, we compute an estimation $\hat{p}_{\mathcal{X}_{\mathcal{D}}}$ of $p_{\mathcal{X}_{\mathcal{D}}}$ from the unlabeled set $\mathcal{D}_{\mathcal{X}}$. In this work we assume we do not know a priori the explicit form of the training data distribution marginal $p_{\mathcal{X}_{\mathcal{L}}}$ that we want to use to sample points. If we knew it we could sample according to it and would not have the need to develop a data selection strategy. Instead, we develop a data selection strategy that induces a training data distribution marginal $p_{\mathcal{X}_{\mathcal{L}}}$, which we estimate from the selected $\mathcal{L}_{\mathcal{X}}$ as $\hat{p}_{\mathcal{X}_{\mathcal{L}}}$. The selection procedure we propose aims at selecting a set $\mathcal{L}_{\mathcal{X}}$ that minimizes a data-driven estimation of the weighted fill distance based on the dataset $\mathcal{D}_{\mathcal{X}}$, the selected subset $\mathcal{L}_{\mathcal{X}}$ and the two estimated marginals $\hat{p}_{\mathcal{X}_{\mathcal{D}}}$ and $\hat{p}_{\mathcal{X}_{\mathcal{L}}}$.

Before diving into the technicalities of our proposed sampling method, let us briefly describe its core procedure. Our approach involves iteratively sampling data points at a maximal weighted distance from those already selected. Initially the weights are set to one for all available points, thus, in this initial stage, our procedure coincides with FPS. After a portion of the data has been selected, the weights adapt dynamically during the selection process according to the chosen training set. The weights reflect the relationship

between $\hat{p}_{\mathcal{X}_{\mathcal{D}}}$ and $\hat{p}_{\mathcal{X}_{\mathcal{L}}}$, which we estimate from the data locations. For this reason we call our approach Density-Aware Farthest Point Sampling (DA-FPS). The weights dynamically change because the estimation of the marginal of the training data distribution, $\hat{p}_{\mathcal{X}_{\mathcal{L}}}$, is data-driven and depends on the current training set. As we add new points to the selected set, $\hat{p}_{\mathcal{X}_{\mathcal{L}}}$ must be updated.

### 5.1 Density estimation

To estimate the multivariate marginal distributions $p_{\mathcal{X}_{\mathcal{D}}}$ and $p_{\mathcal{X}_{\mathcal{L}}}$, we follow along Wang et al. (2009) and choose an adaptive $k$-nearest-neighbor ($k$NN) density estimation approach. Note that, we opted for $k$NN density estimators over other strategies for density or density ratio estimation because they are computationally efficient. Using $k$NNs, we can estimate the density at a specific data point by only considering its distance relationships with its $k$-nearest neighbors, without having to perform additional comparisons with other points in the data set. This reduces the computational costs involved in the iterative process of density estimation. The multivariate $k$NN density estimations of marginal distributions, $p_{\mathcal{X}_{\mathcal{L}}}$ and $p_{\mathcal{X}_{\mathcal{D}}}$, are based on a selected set $\mathcal{L}_{\mathcal{X}}$, the dataset $\mathcal{D}_{\mathcal{X}}$ and a kernel function $K : \mathbb{R}^d \to \mathbb{R}^+$. At a point $\boldsymbol{x} \in \mathcal{X}$ the estimated densities have the form

$$\hat{p}^k_{\mathcal{X}_{\mathcal{D}}}(\boldsymbol{x}) := \frac{\sum_{\boldsymbol{x}_i \in \mathcal{D}_{\mathcal{X}}} K\left(\frac{\boldsymbol{x}-\boldsymbol{x}_i}{r^k_{\mathcal{L}_{\mathcal{X}}}(\boldsymbol{x})}\right)}{|\mathcal{D}_{\mathcal{X}}| \left(r^k_{\mathcal{L}_{\mathcal{X}}}(\boldsymbol{x})\right)^d} \quad \text{and} \quad \hat{p}^k_{\mathcal{X}_{\mathcal{L}}}(\boldsymbol{x}) := \frac{\sum_{\boldsymbol{x}_j \in \mathcal{L}_{\mathcal{X}}} K\left(\frac{\boldsymbol{x}-\boldsymbol{x}_j}{r^k_{\mathcal{L}_{\mathcal{X}}}(\boldsymbol{x})}\right)}{|\mathcal{L}_{\mathcal{X}}| \left(r^k_{\mathcal{L}_{\mathcal{X}}}(\boldsymbol{x})\right)^d}, \tag{5}$$

where

$$r^k_{\mathcal{L}_{\mathcal{X}}}(\boldsymbol{x}) := \min\left\{\min_{\bar{\boldsymbol{x}} \in \mathcal{L}_{\mathcal{X}}} \|\boldsymbol{x}-\bar{\boldsymbol{x}}\|_2 + \frac{\epsilon_{\mathcal{X}}}{|\mathcal{L}_{\mathcal{X}}|}, \rho_k(\boldsymbol{x})\right\} \quad \text{and} \quad K(\boldsymbol{x}) := \begin{cases} \frac{1}{V_d}, & \text{if } \|\boldsymbol{x}\|_2 \leq 1, \\ 0, & \text{otherwise,} \end{cases} \tag{6}$$

with $\rho_k(\boldsymbol{x}) := \min\{r \in \mathbb{R}^+ \text{ such that } |\{\tilde{\boldsymbol{x}} \in \mathcal{D}_{\mathcal{X}} \mid \|\boldsymbol{x}-\tilde{\boldsymbol{x}}\|_2 \leq r\}| \geq k\}$ the distance between $\boldsymbol{x}$ and its $k$-nearest neighbor in $\mathcal{D}_{\mathcal{X}}$, and $V_d := \frac{\pi^{d/2}}{\Gamma(d/2+1)}$ is the volume of the unit ball in $\mathbb{R}^d$, where $\Gamma(r) = (r-1)!$ is the so-called Gamma function. The quantity $r^k_{\mathcal{L}_{\mathcal{X}}}(\boldsymbol{x})$ defines an adaptive neighborhood size, depending on the selected set $\mathcal{L}_{\mathcal{X}}$, aiming at reducing the estimation bias at finite sample sizes (Wang et al., 2009). $\epsilon_{\mathcal{X}}$ in the definition of $r^k_{\mathcal{L}_{\mathcal{X}}}(\boldsymbol{x})$ is an arbitrarily small positive scalar that prevents the denominator in the $k$NN density estimations from becoming zero on points in $\mathcal{L}_{\mathcal{X}}$. Note that, we are interested in estimating the densities only for points in $\mathcal{D}_{\mathcal{X}} \backslash \mathcal{L}_{\mathcal{X}} := \{\boldsymbol{x} \in \mathcal{D}_{\mathcal{X}} \text{ such that } \boldsymbol{x} \notin \mathcal{L}_{\mathcal{X}}\}$, which are not already in $\mathcal{L}_{\mathcal{X}}$, and we may want to select depending on the value they would provide, which we quantify with the associated weighted distance. Nonetheless, the kernel estimations in (5) are well-defined for all points in $\mathcal{X}$. In Appendix D we show that, under some assumptions, $\hat{p}^k_{\mathcal{X}_{\mathcal{L}}}$ and $\hat{p}^k_{\mathcal{X}_{\mathcal{D}}}$ are asymptotically unbiased estimations of $p_{\mathcal{X}_{\mathcal{L}}}$ and $p_{\mathcal{X}_{\mathcal{D}}}$, respectively.

### 5.2 Estimated weighted fill distance

We compute a data-driven estimate of the weighted fill distance of a set $\mathcal{L}_{\mathcal{X}} \subset \mathcal{D}_{\mathcal{X}}$ using the density estimations in (5) as follows

$$\max_{\boldsymbol{x} \in \mathcal{D}_{\mathcal{X}} \backslash \mathcal{L}_{\mathcal{X}}} \left[\left(1 - \frac{\hat{p}^k_{\mathcal{X}_{\mathcal{L}}}(\boldsymbol{x})}{\hat{p}^k_{\mathcal{X}_{\mathcal{D}}}(\boldsymbol{x})}\right) \min_{\boldsymbol{x}_j \in \mathcal{L}_{\mathcal{X}}} \|\boldsymbol{x}-\boldsymbol{x}_j\|_2\right]. \tag{7}$$

Unfortunately, the behavior of the estimated weight function in (7) is inconsistent with that of the true weight function in (3). To see such inconsistency, we consider $\mathcal{N}_k(\boldsymbol{x}) := \{\tilde{\boldsymbol{x}} \in \mathcal{D}_{\mathcal{X}} \text{ such that } \|\boldsymbol{x}-\tilde{\boldsymbol{x}}\|_2 \leq \rho_k(\boldsymbol{x})\} \subset \mathcal{D}_{\mathcal{X}}$ as the set of data points of $\mathcal{D}_{\mathcal{X}}$ in the $k$-neighborhood of $\boldsymbol{x}$, and introduce the weights

$$\omega^k_{\mathcal{L}_{\mathcal{X}}}(\boldsymbol{x}) := \begin{cases} |\{\bar{\boldsymbol{x}} \in \mathcal{D}_{\mathcal{X}} \text{ s.t. } \|\boldsymbol{x}-\bar{\boldsymbol{x}}\|_2 \leq r^k_{\mathcal{L}_{\mathcal{X}}}(\boldsymbol{x})\}|, & \text{if } |\mathcal{L}_{\mathcal{X}}| > 0, \\ k, & \text{if } \mathcal{L}_{\mathcal{X}} = \emptyset \end{cases} \tag{8}$$

defining the number of points in $\mathcal{D}_{\mathcal{X}}$ contained in the ball centered in $\boldsymbol{x}$ with radius $r^k_{\mathcal{L}_{\mathcal{X}}}(\boldsymbol{x})$. Note that, $1 \leq \omega^k_{\mathcal{L}_{\mathcal{X}}}(\boldsymbol{x}) \leq k$. For completeness, we set the weights associated with the empty set equal to $k$. Now that

---

**Algorithm 1** Density-Aware Farthest Point Sampling (DA-FPS)

---

**Input:** Dataset $\mathcal{D}_\mathcal{X} = \{\boldsymbol{x}_i\}_{i=1}^n \subset \mathcal{X}$, data budget $b \in \mathbb{N}$, neighborhood size $k \in \mathbb{N}$, with $b$, $k \ll n$. Set $\mathcal{L}_\mathcal{X} \subset \mathcal{D}_\mathcal{X}$ with $|\mathcal{L}_\mathcal{X}| \ll b$, $u \in \mathbb{N}$ with $u < b$.
**Output:** Subset $\mathcal{L}_\mathcal{X} \subset \mathcal{D}_\mathcal{X}$ with $|\mathcal{L}_\mathcal{X}| = b$.

1: **if** $\mathcal{L}_\mathcal{X} = \emptyset$ **then**
2:      Choose $\hat{\boldsymbol{x}} \in \mathcal{D}_\mathcal{X}$ randomly and set $\mathcal{L}_\mathcal{X} = \{\hat{\boldsymbol{x}}\}$
3: **end if**
4: **while** $|\mathcal{L}_\mathcal{X}| < b$ **do**
5:      **if** $|\mathcal{L}_\mathcal{X}| < u$ **then**
6:          $\bar{\boldsymbol{x}} = \underset{x \in \mathcal{D}_\mathcal{X}}{\arg\max} \left[ \underset{x_j \in \mathcal{L}_\mathcal{X}}{\min} \|\boldsymbol{x} - \boldsymbol{x}_j\|_2 \right].$
7:      **else**
8:          Compute $\omega_{\mathcal{L}_\mathcal{X}}^k(\boldsymbol{x}) \; \forall \; \boldsymbol{x} \in \mathcal{D}_\mathcal{X}/\mathcal{L}_\mathcal{X}$ as in (8).
9:          $\bar{\boldsymbol{x}} = \underset{x \in \mathcal{D}_\mathcal{X}/\mathcal{L}_\mathcal{X}}{\arg\max} \left[ \underset{x_j \in \mathcal{L}_\mathcal{X}}{\min} \|\boldsymbol{x} - \boldsymbol{x}_j\|_2 \omega_{\mathcal{L}_\mathcal{X}}^k(\boldsymbol{x}) \right].$
10:      **end if**
11:      $\mathcal{L}_\mathcal{X} \leftarrow \mathcal{L}_\mathcal{X} \cup \{\bar{\mathbf{x}}\}$
12: **end while**

---

we have introduced the weights in (8) we can rewrite the estimated weight function in (7), evaluated on a point $\boldsymbol{x} \in \mathcal{X}$, as follows

$$1 - \frac{\hat{p}_{\mathcal{X}_\mathcal{L}}^k(\boldsymbol{x})}{\hat{p}_{\mathcal{X}_\mathcal{D}}^k(\boldsymbol{x})} = \begin{cases} 1, & \text{if } \nexists \; \boldsymbol{x}_j \in \mathcal{L}_\mathcal{X} \cap \mathcal{N}_k(\boldsymbol{x}), \\ 1 - \frac{|\mathcal{D}_\mathcal{X}|}{\omega_{\mathcal{L}_\mathcal{X}}^k(\boldsymbol{x})|\mathcal{L}_\mathcal{X}|}, & \text{otherwise.} \end{cases} \tag{9}$$

To see this consider that the numerator of $\hat{p}_{\mathcal{X}_\mathcal{L}}^k(\boldsymbol{x})$, as defined in (5), is either $\frac{1}{V_d}$ or zero depending on whether there exists an $\boldsymbol{x}_j \in \mathcal{L}_\mathcal{X} \cap \mathcal{N}_k(\boldsymbol{x})$ or not. Depending on the ratio $\frac{|\mathcal{D}_\mathcal{X}|}{\omega_{\mathcal{L}_\mathcal{X}}^k(\boldsymbol{x})|\mathcal{L}_\mathcal{X}|}$ and the value of $k$, the estimated weight function in (9) may allow for negative values on data points in $\mathcal{D}_\mathcal{X} \backslash \mathcal{L}_\mathcal{X}$, not yet included in the training set, where we would expect $\hat{p}_{\mathcal{X}_\mathcal{L}}^k(\boldsymbol{x}) \leq \hat{p}_{\mathcal{X}_\mathcal{D}}^k(\boldsymbol{x})$. This behavior is not consistent with the behavior of the true weight function defined in (3), which associates non-negative values with points where $p_{\mathcal{X}_\mathcal{L}}(\boldsymbol{x}) \leq p_{\mathcal{X}_\mathcal{D}}(\boldsymbol{x})$. This artifact is directly linked to the fact that the training and data densities are estimated using sets with different amount of points, i.e., if $|\mathcal{D}_\mathcal{X}| = |\mathcal{L}_\mathcal{X}|$ the estimated weight function in (9) would be non-negative.

Thus, the approach we use to compute the densities' ratio is affected by scaling issues related to the sets' imbalance. But, we observe that the values of the estimated weight function are directly correlated with those of the weights $\omega_{\mathcal{L}_\mathcal{X}}^k(\boldsymbol{x})$ defined in (8). Given $\tilde{\boldsymbol{x}}, \bar{\boldsymbol{x}} \in \mathcal{X}$ we have $\left(1 - \frac{\hat{p}_{\mathcal{X}_\mathcal{L}}^k(\tilde{\boldsymbol{x}})}{\hat{p}_{\mathcal{X}_\mathcal{D}}^k(\tilde{\boldsymbol{x}})}\right) > \left(1 - \frac{\hat{p}_{\mathcal{X}_\mathcal{L}}^k(\bar{\boldsymbol{x}})}{\hat{p}_{\mathcal{X}_\mathcal{D}}^k(\bar{\boldsymbol{x}})}\right) \Rightarrow$ $\omega_{\mathcal{L}_\mathcal{X}}^k(\tilde{\boldsymbol{x}}) \geq \omega_{\mathcal{L}_\mathcal{X}}^k(\bar{\boldsymbol{x}})$ and $\omega_{\mathcal{L}_\mathcal{X}}^k(\tilde{\boldsymbol{x}}) > \omega_{\mathcal{L}_\mathcal{X}}^k(\bar{\boldsymbol{x}}) \Rightarrow \left(1 - \frac{\hat{p}_{\mathcal{X}_\mathcal{L}}^k(\tilde{\boldsymbol{x}})}{\hat{p}_{\mathcal{X}_\mathcal{D}}^k(\tilde{\boldsymbol{x}})}\right) > \left(1 - \frac{\hat{p}_{\mathcal{X}_\mathcal{L}}^k(\bar{\boldsymbol{x}})}{\hat{p}_{\mathcal{X}_\mathcal{D}}^k(\bar{\boldsymbol{x}})}\right)$. The value of $\omega_{\mathcal{L}_\mathcal{X}}^k(\boldsymbol{x})$ is simply the number of points in $\mathcal{D}_\mathcal{X}$ that are close to $\boldsymbol{x}$. Based on these observations, we avoid the scaling issue of the estimated weight function in (9) by replacing it with weights defined in (8).

That is, instead of attempting the minimization of the quantity in (7) we consider the following minimization problem:

$$O_\mathcal{X} \in \underset{\substack{\mathcal{L}_\mathcal{X} \subset \mathcal{D}_\mathcal{X} \\ |\mathcal{L}_\mathcal{X}| = b}}{\arg\min} \left[ \underset{\boldsymbol{x} \in \mathcal{D}_\mathcal{X}}{\max} \left( \omega_{\mathcal{L}_\mathcal{X}}^k(\boldsymbol{x}) \underset{\boldsymbol{x}_j \in \mathcal{L}_\mathcal{X}}{\min} \|\boldsymbol{x} - \boldsymbol{x}_j\|_2 \right) \right]. \tag{10}$$

For a more compact notation, we define the quantity between the squared brackets as follows

$$W_{\mathcal{L}_\mathcal{X}, \mathcal{D}_\mathcal{X}}^k := \underset{\boldsymbol{x} \in \mathcal{D}_\mathcal{X}}{\max} \left( \omega_{\mathcal{L}_\mathcal{X}}^k(\boldsymbol{x}) \underset{\boldsymbol{x}_j \in \mathcal{L}_\mathcal{X}}{\min} \|\boldsymbol{x} - \boldsymbol{x}_j\|_2 \right), \tag{11}$$

and refer to it as *estimated weighted fill distance* of $\mathcal{L}_\mathcal{X}$ in $\mathcal{D}_\mathcal{X}$. The value $k$ in (11) is the amount of nearest neighbors we consider to compute the distance weights as in (8). The optimization problem in (10) is a weighted version of the fill distance minimization problem and is therefore at least NP hard.

### 5.3 DA-FPS algorithm

To address the estimated weighted fill distance minimization problem in (10) we propose the novel DA-FPS, described in Algorithm 1. DA-FPS takes in input a finite dataset $\mathcal{D}_\mathcal{X} \subset \mathbb{R}^d$, a data budget $b \in \mathbb{N}$, a neighborhood size $k \in \mathbb{N}$, with $b$, $k \ll n$, a subset $\mathcal{L}_\mathcal{X} \subset \mathcal{D}_\mathcal{X}$, with $|\mathcal{L}_\mathcal{X}| \ll b$ and an additional hyperparameter $u \in \mathbb{N}$, $u < b$. The input hyperparameter $u \in \mathbb{N}$ allows for a greedy selection with uniform weights until the size of the selected set reaches the value described by the hyperparameter. Later on, the objective of our algorithm is to greedily augment the input subset $\mathcal{L}_\mathcal{X}$ by selecting points from the dataset $\mathcal{D}_\mathcal{X}$ so that the maximum weighted distance of any point in $\mathcal{D}_\mathcal{X}$ to its nearest selected point in $\mathcal{L}_\mathcal{X}$ is minimized. The weights used to scale the distances are defined in (8), depend on the selected set $\mathcal{L}_\mathcal{X}$. Consequently, they must be iteratively updated any time a new point is selected and included in $\mathcal{L}_\mathcal{X}$. The algorithm stops when the number of elements in the selected set $\mathcal{L}_\mathcal{X}$ reaches the data budget $b$. After that, $\mathcal{L}_\mathcal{X}$ is the output.

The behavior of the algorithm is primarily controlled by two hyperparameters: the neighborhood size $k$ for the $k$NN density estimation and the parameter $u$ for initial uniform selection. The parameter $k$ controls how we estimate density and directly affects the weight values $\omega^k_{\mathcal{L}_\mathcal{X}}(\boldsymbol{x})$. Small values of $k$ make the weights very responsive to local changes in density, which may lead to high variance in the values of the weights, and consequently, to noticeable differences in the values of weights associated with points lying in regions with similar true density. Large values of $k$ produce more stable weights but may over-smooth important density variations. As we select more points, the impact of the parameter $k$ becomes less relevant because the weight value at a given location is increasingly determined by how close the nearest selected point is, rather than by the distance from the $k$-th nearest neighbor. This happens because each weight value counts points within a neighborhood whose size depends on the distance to the closest selected point. The parameter $u$ determines the amount of points selected with uniform weight: setting $u = 0$ recovers pure density-aware selection from the start, while $u = b$ reduces to standard FPS. In Section 8, where we report experimental results, we comment on the influence of $k$ and $u$ on DA-FPS performance.

Note that we are considering the Euclidean distance in our formulation of the (estimated) weighted fill distance. However, other distance metrics can be used depending on the data characteristics and the specific application. We follow common practice and use the Euclidean distance for general-purpose applications. We remark that alternative formulations for the estimated weighted fill distance could be considered during the development of DA-FPS. For instance, we chose multiplicative weighting $(\omega^k_{\mathcal{L}_\mathcal{X}}(\boldsymbol{x}) \min_{\boldsymbol{x}_j \in \mathcal{L}_\mathcal{X}} \|\boldsymbol{x} - \boldsymbol{x}_j\|_2)$ rather than additive weighting $(\min_{\boldsymbol{x}_j \in \mathcal{L}_\mathcal{X}} \|\boldsymbol{x} - \boldsymbol{x}_j\|_2 + \alpha \omega^k_{\mathcal{L}_\mathcal{X}}(\boldsymbol{x}))$ because multiplicative weighting was motivated by our theoretical result and avoids the need to tune an additional scaling parameter $\alpha$. Further investigation into alternative formulations and their empirical performance could yield valuable insights and potentially lead to improved sampling strategies.

## 6 Simple illustration of DA-FPS

In this section we provide an intuitive understanding of DA-FPS behavior through a visual 2D example. We compare DA-FPS with uniform random sampling and standard FPS on a synthetic dataset with varying density regions. The illustration demonstrates how DA-FPS balances coverage of the feature space with density awareness, selecting points from high-density regions while still covering sparse areas.

Fig. 1 (top row) shows a dataset of 1000 two-dimensional points in the unit square. The dataset contains a high-density cluster (650 points) in the center, a smaller cluster (200 points) in the lower left, and 150 points scattered uniformly at random. The blue background is a 2D kernel density estimation (KDE) plot, where darker blue means higher density. For the KDE plot we use the Seaborn Python library (Waskom, 2021). The second row of Fig. 1 shows 100-point subsets selected by uniform random sampling, FPS, and DA-FPS. For DA-FPS, we set $u = 0$ and $k = 100$. From a global perspective, DA-FPS balances two objectives: covering the data space evenly while giving greater importance to higher-density regions. Consequently, the

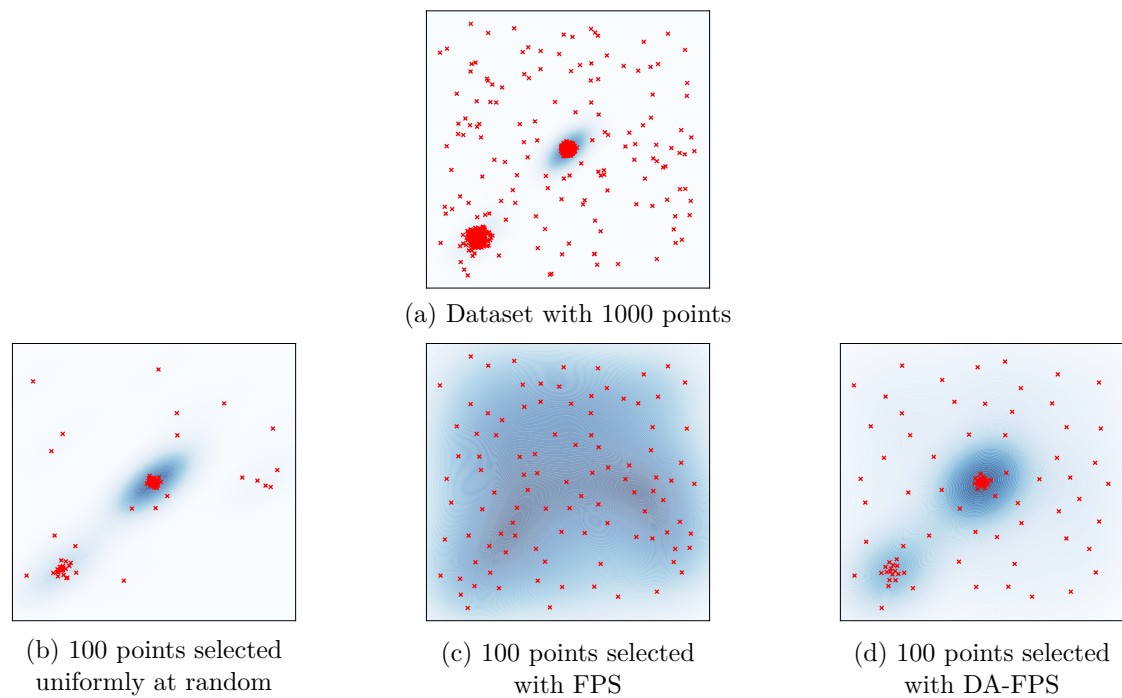

(a) Dataset with 1000 points

(b) 100 points selected uniformly at random

(c) 100 points selected with FPS

(d) 100 points selected with DA-FPS

Figure 1: Illustration of DA-FPS, FPS, and uniform random sampling on a synthetic 2D dataset. Top row: (a) A dataset with 1000 points in the unit square. The dataset consists of a high-density central cluster (650 points), a smaller lower-left cluster (200 points), and uniformly scattered points (150 points). The background shows a 2D kernel density estimation, where darker blue indicates higher density. Bottom row: 100 points selected by each method. (b) Uniform random sampling mostly selects from the dense cluster and may miss sparse regions. (c) FPS selects points evenly across the space, ignoring density. (d) DA-FPS selects more points from dense regions but still covers sparse areas, balancing density and coverage.

sampled data approximates the original distribution but with reduced density differences across the data space. In contrast, FPS consistently selects points to provide a uniform representation of the data space, while uniform random sampling mainly focuses on the high-density clusters, potentially neglecting sparser regions, e.g., no point is selected in the lower right corner when uniform random sampling is used. Locally, both FPS and DA-FPS tend to select points that are evenly distributed without clustering. This is due to their optimization processes, which lead to maximize pairwise (weighted) distances between the selected points, creating a "repulsion effect". One of the key advantages of DA-FPS for sampling training sets lies in its ability to balance the representation of both dense and sparse regions in the data. By ensuring that high-density regions are well-represented while also covering sparser areas, DA-FPS mitigates the risk of over-fitting or under-fitting to dominant clusters, which is a common issue when using random sampling and FPS, respectively.

# 7 Analysis of DA-FPS

In this section we provide a theoretical analysis of DA-FPS performance. We establish that DA-FPS achieves a $2k$-approximation for the estimated weighted fill distance minimization problem in (10), where $k$ is the number of nearest neighbors used in DA-FPS. Following is a discussion of the relationship between the parameter $k$ and approximation quality, as well as computational complexity considerations. We also address limitations of our density estimation approach and implementation details. The following theorem establishes that DA-FPS achieves a $2k$-approximation for the fill distance minimization problem in (10).

**Theorem 7.1.** *Given set of data locations $\mathcal{D}_\mathcal{X} = \{\boldsymbol{x}_i\}_{i=1}^n \subset \mathbb{R}^d$, subset $O_\mathcal{X} \subset \mathcal{D}_\mathcal{X}$, optimal solution to the problem in (10) with $|O_\mathcal{X}| = b \in \mathbb{N}^+$, $b < n$, and $\mathcal{L}_\mathcal{X} \subset \mathcal{D}_\mathcal{X}$, $|\mathcal{L}_\mathcal{X}| = b$, subset selected with Algorithm 1 initialized with $\mathcal{L}_\mathcal{X} = \emptyset$ and $u = 0$, we have*

$$W_{\mathcal{L}_\mathcal{X}, \mathcal{D}_\mathcal{X}}^k \leq 2k W_{O_\mathcal{X}, \mathcal{D}_\mathcal{X}}^k, \tag{12}$$

*where $W_{\mathcal{L}_\mathcal{X}, \mathcal{D}_\mathcal{X}}^k$ and $W_{O_\mathcal{X}, \mathcal{D}_\mathcal{X}}^k$ are the estimated weighted fill distances of $\mathcal{L}_\mathcal{X}$ and $O_\mathcal{X}$ in $\mathcal{D}_\mathcal{X}$, respectively, defined as in (11).*

Proof is in Appendix E. Theorem 7.1 gives an upper bound on the approximation error of DA-FPS for the estimated weighted fill distance minimization problem defined in (10). The theorem indicates that increasing the number $k$ increases the approximation error. Recall that $k$ is the largest possible value of the weights $\omega_{\mathcal{L}_\mathcal{X}}^k(\boldsymbol{x})$. The presence of $k$ in the bound quantifies the worst case approximation error of the weights values, which depend on the selected set $\mathcal{L}_\mathcal{X}$. In particular, the weights associated with the set selected by DA-FPS can differ significantly from those of an optimal set $\mathcal{O}_\mathcal{X}$ of size $b$, especially in the early stages of sampling when $\mathcal{L}_\mathcal{X}$ contains only a few points. Appendix N provides an additional result stating that the approximation error of the DA-FPS estimation depends on the relationship between the weights of an optimal set and those of the set selected by DA-FPS.

It is important to note that the bound in Theorem 7.1 does not account for the quality of the density ratio estimation itself, which also depends on the parameter $k$. In other words, it does not quantify how well the estimated weighted fill distance in (11) relates to the true weighted fill distance in (2). Addressing this issue is an open problem for future work and is closely related to the broader field of density ratio estimation.

The estimated weighted fill distance in (11) may not accurately reflect the true weighted fill distance in (2), particularly when the training set is small or the data are high-dimensional. According to multivariate density estimation theory (Wang et al., 2009), limited sample sizes lead to biased density estimates in (5), which in turn cause the behavior of the estimated weights $\omega_{\mathcal{L}_\mathcal{X}}^k(\boldsymbol{x})$ to deviate from that of the true weights $\psi_{\mathcal{L}_\mathcal{X}}(\boldsymbol{x})$ in (3). Consequently, DA-FPS may appear effective for the estimated problem in (10) but may fail to provide reliable performance for the true problem, especially for small training sets under high-dimensional settings where density estimation is challenging. To mitigate these limitations, we first sample attempting to ensure broad coverage of the data space by using the hyperparameter $u > 0$. That is, during the initial steps of the sampling process, when density estimates are unreliable, we consider constant weights, which we then iteratively update according to (8) after selecting the first $u$ points. Thus, in early stages, DA-FPS coincides with FPS and focuses on minimizing the maximum distance between any point in the dataset $\mathcal{D}_\mathcal{X}$ and its closest selected element. We note that, the theoretical approximation guarantee in Theorem 7.1 applies only to $u = 0$. Nonetheless, using $u > 0$ offers practical benefits: it ensures coverage across different regions of the data space before density-aware sampling begins, and it allows us to accumulate sufficient points to improve the reliability of density estimation needed for effective weight computation. The choice of $u$ is critical to DA-FPS effectiveness. For instance, if set too small, sparse regions may be underrepresented in the training set. This imbalance can result in low model performances in sparse regions of the dataset with a consequential increase in the average error. We speculate that setting $u > 0$ may lead to improved approximation guarantees for DA-FPS with respect to minimizing the true weighted fill distance in (2). This is because the initial uniform sampling phase helps ensure that the selected set $\mathcal{L}_\mathcal{X}$ has a more balanced representation of the data space, which can lead to more accurate density estimates and weight computations in subsequent iterations. However, formalizing this intuition and deriving precise approximation bounds for DA-FPS with $u > 0$ remains an open problem for future research.

Note that, Algorithm 1 can be implemented using $\mathcal{O}(|\mathcal{D}|k)$ memory and takes $\mathcal{O}(db|\mathcal{D}|k)$ time, where $d$ is the data dimension. To give a qualitative understanding, with an implementation in PyTorch (Paszke et al., 2019) it takes approximately 970 seconds to select 26000 points ($\approx 20\%$ of the total) from the QM9 (Ruddigkeit et al., 2012; Ramakrishnan et al., 2014) dataset consisting 130202 data points of dimension 100.[1] In Appendix F we provide more experimental results on the efficiency of DA-FPS.

---

[1]We used a 48-cores CPU with 384 GB RAM.

## 8 Numerical Experiments

In this section we empirically validate DA-FPS on molecular property prediction tasks. We describe our experimental setup, introduce the baseline sampling strategies and regression models, and analyze performance using mean absolute error as our primary evaluation metric.

**Experimental setup**   We focus on molecular property prediction, a regression task in quantum chemistry. We use the QM7, QM8, and QM9 datasets, which contain 7,165, 21,766, and 130,202 molecules, respectively. Our setup is based on Climaco & Garcke (2024), but with important differences. While Climaco & Garcke (2024) studied FPS mainly for small training sets (1% to 10% of the data), we evaluate DA-FPS for larger training set sizes, from 5% up to 20% of the data pool. This allows us to address challenges with high-dimensional density estimation in very small data regimes (which DA-FPS requires), and to test DA-FPS in scenarios where FPS alone does not significantly outperform standard passive sampling in terms of average prediction error. Our data preprocessing is similar to Climaco & Garcke (2024), but we reduce the feature vector dimension (up to 276 instead of up to 1300) to further mitigate issues with high-dimensional density estimation. Full details on datasets, preprocessing, and descriptors are in Appendix G. Appendix H provides empirical evidence that the datasets satisfy the required data assumption B.2. We use two regression models that have been utilized in previous works for molecular property prediction tasks: kernel ridge regression with a Gaussian kernel (KRR) (Stuke et al., 2019; Deringer et al., 2021) and feed-forward neural networks (FNNs) (Pinheiro et al., 2020). Both models are Lipschitz continuous, which is required for our theoretical results (see Appendix I for details on the models and hyperparameter tuning). We evaluate prediction performance using the mean absolute error (MAE), defined as $\text{MAE} := \frac{1}{n_u} \sum_{i=1}^{n_u} |y_i - \tilde{y}_i|$, where $y_i$ are true values, $\tilde{y}_i$ are predictions, and $n_u$ is the number of unlabeled points in the data pool. The code necessary to reproduce the results presented in this section is available in our GitHub repository.[2] Let us remark that the final goal of our experiments is to empirically show the benefits of using DA-FPS compared to other model-agnostic state-of-the-art sampling approaches and investigate the benefits of complementing classical passive sampling approaches with the FPS. We do not make any claims on the general prediction quality of the employed models on any of the studied datasets.

**Baseline sampling strategies**   We compare our approach with four sampling techniques, among those analyzed in well-established papers related to data selection (Sener & Savarese, 2018; Ghorbani & Zou, 2019; Mirzasoleiman et al., 2020; Killamsetty et al., 2021), that fit our application scenario and problem constraints: passive sampling model-agnostic data selection techniques that do not rely on the knowledge of the labels. Specifically, we consider random sampling (RDM), Farthest Point Sampling (FPS), facility location (Frieze, 1974) and $k$-medoids++ (Mannor et al., 2011). Both facility location and $k$-medoids++ attempt to minimize a sum of pairwise squared distances. However, the fundamental difference is that facility location is a greedy technique, while $k$-medoids++ is based on a segmentation of the data points into clusters.

**Molecular property prediction on QM datasets**   Fig. 2 shows the MAE for the regression tasks on the QM7, QM8 and QM9 datasets using KRR (top row) and FNN (bottom row). For each combination of training set size and sampling strategy, the MAE is computed considering all the data points in the available dataset that have not been selected for training. Our experiments indicate that selecting the training sets using DA-FPS leads to better prediction performances than the baselines RDM, facility location, FPS and $k$-medoids++ in terms of the MAE, particularly on the larger QM8 and QM9 and for larger training set sizes (> 5% of the available data points). Comparing the MAE obtained with KRR and FNN, reveals that the standard deviation of the MAE, represented by the error bars, tends to be larger for the FNN, especially on the smaller QM7. This occurs because the FNN training process is sensitive to the limited amount of training data available with the QM7. More specifically, the FNN determines its regression parameters by solving a non-linear optimization problem using stochastic gradient descent, which is inherently prone to variability and instability. As a result, metrics like the MAE often exhibit larger standard deviations. In contrast, KRR solves a convex optimization problem with a closed-form solution. This deterministic approach results

---

[2]https://github.com/PaClimaco/DA-FPS

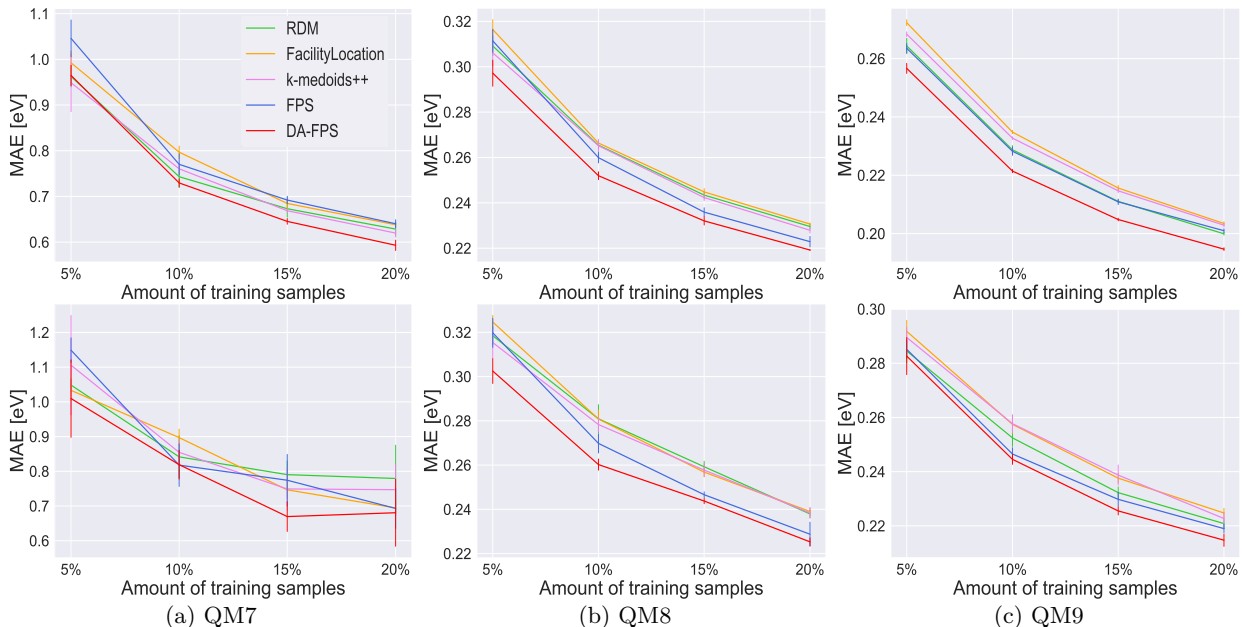

Figure 2: MAE for regression tasks on QM datasets using KRR with Gaussian kernel (top row) and FNN (bottom row) trained on sets of various sizes, expressed as a percentage of the available data points, and selected with different sampling strategies. Error bars represent the standard deviation over five runs. DA-FPS (red lines) outperforms the baselines. The legend in the top-row leftmost graph applies to all graphs. DA-FPS is initialized with $\mathcal{L}_{\mathcal{X}} = \emptyset$, $k = 100$, and $u = 3\%$ of the available data, independently of the dataset.

in lower variability in predictions and lower standard deviation of the MAE. In Appendix J we evaluate DA-FPS considering the root mean squared error (RMSE $:= \sqrt{\frac{1}{n_u} \sum_{i=1}^{n_u} |y_i - \tilde{y}_i|^2}$) and show that DA-FPS is still the most competitive approach. In Appendix K we provide results with two additional datasets, not related to the quantum chemistry domain, two additional baseline selection strategies and one additional regression model. The results of the additional experiments in Appendix K indicate that using DA-FPS leads to the most competitive results in terms of MAE and RMSE.

It is important to note that, as explained in Section 7, during the initial phase of the sampling process DA-FPS uses uniform weights. Meaning that, DA-FPS initially coincide with FPS. The amount of points selected with uniform weights is controlled by the hyperparameter $u$. In Appendix L we empirically investigate the benefits of combining FPS with the baselines used to benchmark DA-FPS. The experiments in Appendix L show that augmenting the baselines approaches with FPS during the initial steps of the sampling process consistently leads to a decrease in the MAE of the predictions. Still, DA-FPS remains the most competitive approach. This investigation highlights that the advantages of DA-FPS stem not only from initially using FPS but also from its density-aware weighting, taking place after the first $u$ points have been selected.

Regarding the DA-FPS hyperparameter $u$, determining the amount of points selected with uniform weights, we make observations that can guide a user to a suitable choice. Our ablation study on the ZINC dataset (Appendix M) shows that for larger training set sizes ($> 10\%$), the choice of $u$ has little effect on DA-FPS performance compared to the baselines. However, for smaller training set sizes, $u$ has a stronger impact. When $u$ is large, DA-FPS behaves more like FPS, which is known to reduce the maximum absolute prediction error (MAXAE $:= \max_{1 \le i \le n_u} |y_i - \tilde{y}_i|$), but may not lead to better average error. The work in Climaco & Garcke (2024) found that, on the QM datasets, selecting just the first $2\%$ of points with FPS, then switching to random sampling, achieves similar MAXAE as using FPS alone. This suggests that $u$ should not be chosen too large (e.g., $< 5\%$ of the available data points). Based on these observations we think that an "optimal" choice $u$ depends on what the user wants to achieve. Qualitatively, if the user is interested in ensuring low MAXAE, at the cost of higher MAE, a larger $u$ is recommended. If the user is mainly interested in performing well on average in the region of high data density, a smaller $u$ is recommended.

In Appendix M, we also analyze how changing the DA-FPS hyperparameter $k$ affects performance. $k$ defines the amount of $k$-nearest neighbors considered for computing the weights. The results show that DA-FPS works well for a range of $k$ values, but choosing $k$ that is too small can reduce its effectiveness. The rough magnitude of $k$ likely depends on the data dimension and distribution and is generally investigated in the context of density estimation. The problem of choosing $k$ is common for $k$NN-based density estimation methods. We expect further insights from that domain. We think that the choice of $k$ should take into account at least two aspects: First, $k$ should not be too small, because it would lead to high variance in the values of the weights, and consequently, to noticeable differences in the values of weights associated with points lying in regions with similar true density. Second, $k$ should not be too large as the computational cost of DAFPS depends linearly on $k$.

## 9    Conclusion

We proposed DA-FPS, a passive model-agnostic sampling algorithm to select training sets by weighted fill distance minimization. We provided an upper bound for the approximation error of DA-FPS to the estimated weighted fill distance minimization problem we defined. Our numerical results demonstrated that utilizing DA-FPS to select training sets has a positive impact on the average prediction of the regression models, particularly for larger datasets, in line with our theoretical motivation.

A key aspect of DA-FPS is the need to estimate the data densities. Poor density estimation in DA-FPS can lead to ineffective weights that cause oversampling of high-density regions, generating redundant information in the selected set, and undersampling of low-density regions, leading to underrepresentation of important areas of the data space in the training set. This can lead to degraded model performance, for instance, through increased prediction errors in underrepresented regions. In this paper, we used a $k$-nearest neighbors (kNN) approach for density estimation. Using the kNN was computationally efficient but led to scaling issues when estimating the weight function used in the definition of the weighted fill distance. Alternative methods that overcome this issue while remaining effective and computationally efficient could be developed. Our ablation study on the hyperparameter $k$ in Appendix M provides empirical evidence that choosing $k$ too small can negatively impact performance, highlighting the importance of appropriate density estimation. Future research directions should investigate alternatives for the data density estimations or approaches for direct density ratio estimation. Concerning the choice of DA-FPS hyperparameter $u$, we hypothesize that statistical properties, such as density or size of the tails of the dataset, may be exploited by a user to take an informed decision. Developing adaptive strategies for automatically selecting the hyperparameters $k$ and $u$ based on dataset characteristics would enhance the robustness and applicability of DA-FPS.

## Notation

### Broader Impact Statement

Selecting points aiming to reduce the weighted fill distance of the training set is beneficial when traditional labeling methods, such as numerical simulations or laboratory experiments, are expensive or time-consuming. In such applications ML regression models provide fast label predictions for new data points. The quality of these predictions depends on the quality of the training data. Therefore, selecting an effective training set is crucial to ensure accurate predictions. Our research on minimizing the training set weighted fill distance can potentially identify sets that can enhance average prediction quality for various regression models and tasks, preventing the wastage of expensive labeling resources on training sets that may only benefit a specific learning model or task.

### Acknowledgments

This work was partly supported by the German Federal Ministry of Education and Research (BMBF) within the projects MaGriDo (Mathematics for Machine Learning Methods for Graph-Based Data with Integrated Domain Knowledge), FKZ 05M20PDB and Rhine-Ruhr Center for Scientific Data Literacy (DKZ.2R), FKZ 16DKZ2030C.

| Symbol | Description |
|---|---|
| $\mathcal{X}$ | Feature space, subset of $\mathbb{R}^d$ |
| $\mathcal{Y}$ | Label space, subset of $\mathbb{R}$ |
| $\mathcal{D}$ | Pool of available data points, $\{(\boldsymbol{x}_i, y_i)\}_{i=1}^n \subset \mathcal{X} \times \mathcal{Y}$ |
| $\mathcal{D}_{\mathcal{X}}$ | Set of feature vectors of points in $\mathcal{D}$, $\{\boldsymbol{x}_i\}_{i=1}^n \subset \mathcal{X}$ |
| $\mathcal{L}$ | Training set $\{(\boldsymbol{x}_{i_j}, y_{i_j})\}_{j=1}^b \subset \mathcal{D}$, with $i_j \in \{1, \ldots, n\}$ |
| $\mathcal{L}_{\mathcal{X}}$ | Set of feature vectors of points in $\mathcal{L}$, $\{\boldsymbol{x}_{i_j}\}_{j=1}^b \subset \mathcal{D}_{\mathcal{X}}$ |
| $p_{\mathcal{D}}$ | Data source distribution on $\mathcal{X} \times \mathcal{Y}$ |
| $p_{\mathcal{L}}$ | Training data distribution on $\mathcal{X} \times \mathcal{Y}$ |
| $p_{\mathcal{X}_{\mathcal{D}}}$ | Marginal of $p_{\mathcal{D}}$ on $\mathcal{X}$ |
| $p_{\mathcal{X}_{\mathcal{L}}}$ | Marginal of $p_{\mathcal{L}}$ on $\mathcal{X}$ |
| $\hat{p}_{\mathcal{X}_{\mathcal{D}}}$ | General density estimator of $p_{\mathcal{X}_{\mathcal{D}}}$ |
| $\hat{p}_{\mathcal{X}_{\mathcal{L}}}$ | General density estimator of $p_{\mathcal{X}_{\mathcal{L}}}$ |
| $\hat{p}_{\mathcal{X}_{\mathcal{D}}}^k$ | $k$-NN density estimator of $p_{\mathcal{X}_{\mathcal{D}}}$ |
| $\hat{p}_{\mathcal{X}_{\mathcal{L}}}^k$ | $k$-NN density estimator of $p_{\mathcal{X}_{\mathcal{L}}}$ |
| $\psi_{\mathcal{L}_{\mathcal{X}}}(\boldsymbol{x})$ | Weight function: $1 - \frac{p_{\mathcal{X}_{\mathcal{L}}}(\boldsymbol{x})}{p_{\mathcal{X}_{\mathcal{D}}}(\boldsymbol{x})}$ |
| $W_{\mathcal{L}_{\mathcal{X}}, \mathcal{X}}(p_{\mathcal{X}_{\mathcal{L}}} \| p_{\mathcal{X}_{\mathcal{D}}})$ | Weighted fill distance of $\mathcal{L}_{\mathcal{X}}$ in $\mathcal{X}$ |
| $W_{\mathcal{L}_{\mathcal{X}}, \mathcal{D}_{\mathcal{X}}}^k$ | Estimated weighted fill distance using $k$-NN |
| $r_{\mathcal{L}_{\mathcal{X}}}^k(\boldsymbol{x})$ | Adaptive neighborhood radius for density estimation |
| $\omega_{\mathcal{L}_{\mathcal{X}}}^k(\boldsymbol{x})$ | Number of points in $k$-neighborhood of $\boldsymbol{x}$ within a distance $r_{\mathcal{L}_{\mathcal{X}}}^k(\boldsymbol{x})$ |
| $\rho_k(\boldsymbol{x})$ | Distance to $k$-th nearest neighbor of $\boldsymbol{x}$ in $\mathcal{D}_{\mathcal{X}}$ |
| $m_{\mathcal{L}}$ | Regression model trained on $\mathcal{L}$ |
| $l(\boldsymbol{x}, y, m_{\mathcal{L}})$ | Error function measuring prediction error |

Table 1: Table of notation used throughout the paper.

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

# A    The weighted fill distance is non-negative

*Remark* A.1. The weighted fill distance defined in (2) is non-negative.

*Proof.* In the definition of the weighted fill distance in (2) the distances are by definition non-negative. Therefore, it is enough to show that there always exists a point $\boldsymbol{x} \in \mathcal{X}$ such that $p_{\mathcal{X}_{\mathcal{D}}}(\boldsymbol{x}) \geq p_{\mathcal{X}_{\mathcal{L}}}(\boldsymbol{x})$, that is, there always exists a point where the weight function is non-negative. Let us proceed by contradiction. Let us assume that $\forall \boldsymbol{x} \in \mathcal{X}$ we have that $p_{\mathcal{X}_{\mathcal{D}}}(\boldsymbol{x}) < p_{\mathcal{X}_{\mathcal{L}}}(\boldsymbol{x})$. Next, let us note that $\int_{\mathcal{X}} p_{\mathcal{X}_{\mathcal{L}}}(\boldsymbol{x})d\boldsymbol{x} = \int_{\mathcal{X}} p_{\mathcal{X}_{\mathcal{D}}}(\boldsymbol{x})d\boldsymbol{x} = 1$, which follows from the fact that both, $p_{\mathcal{X}_{\mathcal{D}}}$ and $p_{\mathcal{X}_{\mathcal{L}}}$ are probability distributions on $\mathcal{X}$. However, by our assumption we have that

$$1 = \int_{\mathcal{X}} p_{\mathcal{X}_{\mathcal{D}}}(\boldsymbol{x})d\boldsymbol{x} < \int_{\mathcal{X}} p_{\mathcal{X}_{\mathcal{L}}}(\boldsymbol{x})d\boldsymbol{x} = 1,$$

which is a contradiction. $\qquad\qquad\square$

# B    Assumptions

We recapture two assumptions from Climaco & Garcke (2024) that underlie our theoretical results. The first assumption concerns the data being analyzed and the relationship between data features and labels.

**Assumption B.1.** We assume that for any feature vector $\boldsymbol{x}_q \in \mathcal{X}$ we have that

$$\mathbb{E}\left[|Y|\big|\boldsymbol{x}_q\right] := \int_{\mathcal{Y}} |y|\, p(y|\boldsymbol{x}_q)dy < \infty \tag{13}$$

and that there exists $\epsilon \geq 0$ such that

$$\mathbb{E}\left[|Y - \mathbb{E}[Y|\boldsymbol{x}_q]|\,\big|\,\boldsymbol{x}_q\right] := \int_{\mathcal{Y}} \big|y - \mathbb{E}[Y|\boldsymbol{x}_q]\big|\, p(y|\boldsymbol{x}_q)dy \leq \epsilon, \tag{14}$$

where

$$p(y|\boldsymbol{x}_q) := \frac{p_{\mathcal{Z}}(\boldsymbol{x}_q, y)}{p_{\mathcal{X}}(\boldsymbol{x}_q)} \quad \text{and} \quad p_{\mathcal{X}}(\boldsymbol{x}_q) := \int_{\mathcal{Y}} p_{\mathcal{Z}}(\boldsymbol{x}_q, y)dy. \tag{15}$$

We refer to "$\epsilon$" as the *labels' uncertainty*. Moreover, we assume that

$$\big|\,\mathbb{E}\left[Y|\hat{\boldsymbol{x}}\right] - \mathbb{E}\left[Y|\tilde{\boldsymbol{x}}\right]\big| \leq \lambda_p \|\hat{\boldsymbol{x}} - \tilde{\boldsymbol{x}}\|_2, \tag{16}$$

$\forall\, \hat{\boldsymbol{x}}, \tilde{\boldsymbol{x}} \in \mathcal{X}$, where $\lambda_p \in \mathbb{R}^+$.

The assumption in (14) states that given the data feature location $\boldsymbol{X} = \boldsymbol{x}_i$, the associated label value is close to the conditional expected value of the random variable $Y$ at that location. This assumption models scenarios where the true feature-label relationship is stochastic or deterministic but subject to random fluctuations with a magnitude parametrized by the scalar $\epsilon$. The Lipschitz continuity assumption in equation (16) relates to the smoothness of the map connecting $\mathcal{X}$ and $\mathcal{Y}$. It implies that when two data points are closer in $\mathcal{X}$, their corresponding labels tend to be closer in $\mathcal{Y}$.

The second assumption concerns the error function used to evaluate the performance of the model and the prediction quality of the model on the training set. Firstly, to formalize the notion that the prediction error of a trained model on the training set is bounded. Secondly, to limit our analysis to error functions that exhibit a certain degree of regularity, which also reflects the regularity of the regression model.

**Assumption B.2.** We assume there exist $\epsilon_{\mathcal{L}} \geq 0$, depending on the labeled set $\mathcal{L} \subset \mathcal{D} := \{(\boldsymbol{x}_q, y_q)\}_{q=1}^n \subset \mathcal{X} \times \mathcal{Y}$ and the trained regression model $m_{\mathcal{L}}$, such that for any labeled point $(\boldsymbol{x}_j, y_j) \in \mathcal{L}$ we have that

$$\mathbb{E}[l(\boldsymbol{x}_j, Y, m_{\mathcal{L}})\big|\boldsymbol{x}_j] \leq \epsilon_{\mathcal{L}}. \tag{17}$$

We consider $\epsilon_{\mathcal{L}}$ as the maximum expected prediction error of the trained model $m_{\mathcal{L}}$ on the labeled data $\mathcal{L}$. Moreover, we assume that for any $y \in \mathcal{Y}$ and $\mathcal{L} \subset \mathcal{D}$ the error function $l(\cdot, y, m_{\mathcal{L}})$ is $\lambda_{l_{\mathcal{X}}}$-Lipschitz and that for any $\boldsymbol{x} \in \mathcal{X}$ and $\mathcal{L} \subset \mathcal{D}$, $l(\boldsymbol{x}, \cdot, m_{\mathcal{L}})$ is $\lambda_{l_{\mathcal{Y}}}$-Lipschitz, convex and $\mathbb{E}[|l(\boldsymbol{x}, Y, m_{\mathcal{L}})|\big|\boldsymbol{x}] < \infty$.

With (17) we assume that the expected error on the training set is bounded. Moreover, with the Lipschitz continuity assumptions we limit our study to error functions that show a certain regularity. However, these regularity assumptions on the error function are not too restrictive and are connected with the regularity of the evaluated trained model as already shown in Climaco & Garcke (2024). For instance, the $\lambda_{l_{\mathcal{Y}}}$-Lipschitz regularity and the convexity in the second argument are verified by all $L_p$-norm error functions, with $1 \leq p < \infty$. We also assume that $\mathbb{E}[|l(\boldsymbol{x}, Y, m_{\mathcal{L}})|\,\|\boldsymbol{x}] < \infty$ for any given $\boldsymbol{x} \in \mathcal{X}$. This assumption formalizes the intuitive fact that, in applications, independently of the trained model and the feature vector considered, we can expect the error function to take finite values.

## C   Proof Theorem 4.2

*Proof.* First, let us notice that

$$
\begin{aligned}
&\mathbb{E}_{p_{\mathcal{D}}} \left[ l(\boldsymbol{X}, Y, m_{\mathcal{L}}) \right] \\
&= \mathbb{E}_{p_{\mathcal{D}}} \left[ l(\boldsymbol{X}, Y, m_{\mathcal{L}}) \right] - \mathbb{E}_{p_{\mathcal{L}}} \left[ l(\boldsymbol{X}, Y, m_{\mathcal{L}}) \right] + \mathbb{E}_{p_{\mathcal{L}}} \left[ l(\boldsymbol{X}, Y, m_{\mathcal{L}}) \right] \\
&= \int_{\mathcal{X}} \mathbb{E} \left[ l(\boldsymbol{x}, Y, m_{\mathcal{L}})|\boldsymbol{x} \right] p_{\mathcal{X}_{\mathcal{D}}}(\boldsymbol{x}) d\boldsymbol{x} - \int_{\mathcal{X}} \mathbb{E} \left[ l(\boldsymbol{x}, Y, m_{\mathcal{L}})|\boldsymbol{x} \right] p_{\mathcal{X}_{\mathcal{L}}}(\boldsymbol{x}) d\boldsymbol{x} + \mathbb{E}_{p_{\mathcal{L}}} \left[ l(\boldsymbol{X}, Y, m_{\mathcal{L}}) \right] \\
&\leq \int_{\substack{\mathcal{X}, \\ p_{\mathcal{X}_{\mathcal{D}}} \geq p_{\mathcal{X}_{\mathcal{L}}}}} \mathbb{E} \left[ l(\boldsymbol{x}, Y, m_{\mathcal{L}})|\boldsymbol{x} \right] \left( p_{\mathcal{X}_{\mathcal{D}}}(\boldsymbol{x}) - p_{\mathcal{X}_{\mathcal{L}}}(\boldsymbol{x}) \right) d\boldsymbol{x} + \mathbb{E}_{p_{\mathcal{L}}} \left[ l(\boldsymbol{X}, Y, m_{\mathcal{L}}) \right].
\end{aligned}
\tag{18}
$$

Note that in (18) the expectation $\mathbb{E} \left[ l(\boldsymbol{x}, Y, m_{\mathcal{L}})|\boldsymbol{x} \right]$ is independent of $p_{\mathcal{D}}$ and $p_{\mathcal{L}}$. This is because, as mentioned in Section 3, we consider a scenario where $p_{\mathcal{D}}$ may differ from $p_{\mathcal{L}}$ but the map connecting a data location $\boldsymbol{x} \in \mathcal{X}$ and its associated label value is independent on how the data is selected, that is, $p_{\mathcal{D}} \neq p_{\mathcal{L}}$ and $p_{\mathcal{D}}(y|\boldsymbol{x}) = p_{\mathcal{L}}(y|\boldsymbol{x})$. In what follows we define $p(y|\boldsymbol{x}) := p_{\mathcal{D}}(y|\boldsymbol{x}) = p_{\mathcal{L}}(y|\boldsymbol{x})$. Next, fixed $\boldsymbol{X} = \tilde{\boldsymbol{x}} \in \mathcal{X}$, we want to bound $\mathbb{E} \left[ l(\tilde{\boldsymbol{x}}, Y, m_{\mathcal{L}})|\tilde{\boldsymbol{x}} \right]$. To do that we use a result from Climaco & Garcke (2024): For fixed $\tilde{\boldsymbol{x}} \in \mathcal{X}$ and $\boldsymbol{x}_j \in \mathcal{L}_{\mathcal{X}}$ we have

$$
\begin{aligned}
\mathbb{E} \left[ l(\tilde{\boldsymbol{x}}, Y, m_{\mathcal{L}})|\tilde{\boldsymbol{x}} \right] &= \int_{\mathcal{Y}} l(\tilde{\boldsymbol{x}}, y, m_{\mathcal{L}}) p(y|\tilde{\boldsymbol{x}}) dy \\
&\leq \int_{\mathcal{Y}} |l(\tilde{\boldsymbol{x}}, y, m_{\mathcal{L}}) - l(\boldsymbol{x}_j, y, m_{\mathcal{L}})|\, p(y|\tilde{\boldsymbol{x}}) dy + \int_{\mathcal{Y}} l(\boldsymbol{x}_j, y, m_{\mathcal{L}}) p(y|\tilde{\boldsymbol{x}}) dy \\
&\leq \|\tilde{\boldsymbol{x}} - \boldsymbol{x}_j\|_2 \lambda_{l_{\mathcal{X}}} + \int_{\mathcal{Y}} l(\boldsymbol{x}_j, y, m_{\mathcal{L}}) p(y|\tilde{\boldsymbol{x}}) dy
\end{aligned}
\tag{19}
$$

The second inequality in (19) follows from the $\lambda_{l_{\mathcal{X}}}$-Lipschitz continuity of the error function. We can bound the remaining term as follows

$$
\begin{aligned}
&\int_{\mathcal{Y}} l(\boldsymbol{x}_j, y, m_{\mathcal{L}}) p(y|\tilde{\boldsymbol{x}}) dy \\
&\leq \int_{\mathcal{Y}} |l(\boldsymbol{x}_j, y, m_{\mathcal{L}}) - l(\boldsymbol{x}_j, \mathbb{E} \left[ Y|\tilde{\boldsymbol{x}} \right], m_{\mathcal{L}})|\, p(y|\tilde{\boldsymbol{x}}) dy \\
&\quad + \int_{\mathcal{Y}} |l(\boldsymbol{x}_j, \mathbb{E} \left[ Y|\tilde{\boldsymbol{x}} \right], m_{\mathcal{L}}) - l(\boldsymbol{x}_j, \mathbb{E} \left[ Y|\boldsymbol{x}_j \right], m_{\mathcal{L}})|\, p(y|\tilde{\boldsymbol{x}}) dy \\
&\quad + \int_{\mathcal{Y}} l(\boldsymbol{x}_j, \mathbb{E} \left[ Y|\boldsymbol{x}_j \right], m_{\mathcal{L}}) p(y|\tilde{\boldsymbol{x}}) dy \\
&\leq \lambda_{l_{\mathcal{Y}}} \int_{\mathcal{Y}} |y - \mathbb{E} \left[ Y|\tilde{\boldsymbol{x}} \right]|\, p(y|\tilde{\boldsymbol{x}}) dy \\
&\quad + \lambda_{l_{\mathcal{Y}}} \int_{\mathcal{Y}} |\mathbb{E} \left[ Y|\tilde{\boldsymbol{x}} \right] - \mathbb{E} \left[ Y|\boldsymbol{x}_j \right]|\, p(y|\tilde{\boldsymbol{x}}) dy
\end{aligned}
$$

$$+ \int_{\mathcal{Y}} \mathbb{E}[l(\boldsymbol{x}_j, Y, m_{\mathcal{L}})|\boldsymbol{x}_j] p(y|\tilde{\boldsymbol{x}}) dy$$

$$\leq \lambda_{l_{\mathcal{Y}}} \epsilon + \lambda_{l_{\mathcal{Y}}} \int_{\mathcal{Y}} (\lambda_p \|\tilde{\boldsymbol{x}} - \boldsymbol{x}_j\|_2) \, p(y|\tilde{\boldsymbol{x}}) dy + \int_{\mathcal{Y}} \epsilon_{\mathcal{L}} \, p(y|\tilde{\boldsymbol{x}}) dy$$

$$\leq \lambda_{l_{\mathcal{Y}}} \epsilon + \lambda_{l_{\mathcal{Y}}} \lambda_p \|\tilde{\boldsymbol{x}} - \boldsymbol{x}_j\|_2 + \epsilon_{\mathcal{L}}.$$

The second inequality follows from the $\lambda_{l_{\mathcal{Y}}}$-Lipschitz continuity of the error function and Jensen's inequality, which is used to obtain the conditional expectation in the integrand of the last term. The third inequality follows from the definition of labels' uncertainty, the $\lambda_p$-Lipschitz continuity of the conditional expectation of the random variable $Y$ and the assumption that the expected error on the training set is bounded by $\epsilon_{\mathcal{L}}$. The fourth inequality is obtained by taking out the constants from the integrals in the second and third terms and noticing that, from the definition of $p(y|\tilde{\boldsymbol{x}})$, we have $\int_{\mathcal{Y}} p(y|\tilde{\boldsymbol{x}}) dy = 1$.

By taking the minimum over $\boldsymbol{x}_j \in \mathcal{L}_{\mathcal{X}}$ we get

$$\mathbb{E}\left[l(\tilde{\boldsymbol{x}}, Y, m_{\mathcal{L}})|\tilde{\boldsymbol{x}}\right] \leq \min_{\boldsymbol{x}_j \in \mathcal{L}_{\mathcal{X}}} \|\tilde{\boldsymbol{x}} - \boldsymbol{x}_j\|_2 (\lambda_{l_{\mathcal{X}}} + \lambda_{l_{\mathcal{Y}}} \lambda_p) + \lambda_{l_{\mathcal{Y}}} \epsilon + \epsilon_{\mathcal{L}}. \tag{20}$$

Next, we define $C := (\lambda_{l_{\mathcal{X}}} + \lambda_{l_{\mathcal{Y}}} \lambda_p)$ and apply (20) to (18). Consequently, we have

$$\mathbb{E}_{p_{\mathcal{D}}}\left[l(\boldsymbol{X}, Y, m_{\mathcal{L}})\right]$$

$$\leq \int_{\substack{\mathcal{X}, \\ p_{\mathcal{X}_{\mathcal{D}}} \geq p_{\mathcal{X}_{\mathcal{L}}}}} \left(\min_{\boldsymbol{x}_j \in \mathcal{L}_{\mathcal{X}}} \|\boldsymbol{x} - \boldsymbol{x}_j\|_2 C + \lambda_{l_{\mathcal{Y}}} \epsilon + \epsilon_{\mathcal{L}}\right) (p_{\mathcal{X}_{\mathcal{D}}}(\boldsymbol{x}) - p_{\mathcal{X}_{\mathcal{L}}}(\boldsymbol{x})) \, d\boldsymbol{x} + \mathbb{E}_{p_{\mathcal{L}}}\left[l(\boldsymbol{X}, Y, m_{\mathcal{L}})\right]$$

$$= \int_{\substack{\mathcal{X}, \\ p_{\mathcal{X}_{\mathcal{D}}} \geq p_{\mathcal{X}_{\mathcal{L}}}}} \min_{\boldsymbol{x}_j \in \mathcal{L}_{\mathcal{X}}} \|\boldsymbol{x} - \boldsymbol{x}_j\|_2 C \, (p_{\mathcal{X}_{\mathcal{D}}}(\boldsymbol{x}) - p_{\mathcal{X}_{\mathcal{L}}}(\boldsymbol{x})) \, d\boldsymbol{x}$$

$$+ \int_{\substack{\mathcal{X}, \\ p_{\mathcal{X}_{\mathcal{D}}} \geq p_{\mathcal{X}_{\mathcal{L}}}}} (\lambda_{l_{\mathcal{Y}}} \epsilon + \epsilon_{\mathcal{L}}) \, (p_{\mathcal{X}_{\mathcal{D}}}(\boldsymbol{x}) - p_{\mathcal{X}_{\mathcal{L}}}(\boldsymbol{x})) \, d\boldsymbol{x} + \mathbb{E}_{p_{\mathcal{L}}}\left[l(\boldsymbol{X}, Y, m_{\mathcal{L}})\right]$$

$$\leq \int_{\substack{\mathcal{X}, \\ p_{\mathcal{X}_{\mathcal{D}}} \geq p_{\mathcal{X}_{\mathcal{L}}}}} \min_{\boldsymbol{x}_j \in \mathcal{L}_{\mathcal{X}}} \|\boldsymbol{x} - \boldsymbol{x}_j\|_2 C \, (p_{\mathcal{X}_{\mathcal{D}}}(\boldsymbol{x}) - p_{\mathcal{X}_{\mathcal{L}}}(\boldsymbol{x})) \, d\boldsymbol{x}$$

$$+ (\lambda_{l_{\mathcal{Y}}} \epsilon + \epsilon_{\mathcal{L}}) \left( \int_{\mathcal{X}} p_{\mathcal{X}_{\mathcal{D}}}(\boldsymbol{x}) d\boldsymbol{x} - \int_{\substack{\mathcal{X}, \\ p_{\mathcal{X}_{\mathcal{D}}} \geq p_{\mathcal{X}_{\mathcal{L}}}}} p_{\mathcal{X}_{\mathcal{L}}}(\boldsymbol{x}) d\boldsymbol{x} \right) + \mathbb{E}_{p_{\mathcal{L}}}\left[l(\boldsymbol{X}, Y, m_{\mathcal{L}})\right]$$

$$\leq \int_{\substack{\mathcal{X}, \\ p_{\mathcal{X}_{\mathcal{D}}} \geq p_{\mathcal{X}_{\mathcal{L}}} \\ p_{\mathcal{X}_{\mathcal{D}}} > 0}} \min_{\boldsymbol{x}_j \in \mathcal{L}_{\mathcal{X}}} \|\boldsymbol{x} - \boldsymbol{x}_j\|_2 C \left(1 - \frac{p_{\mathcal{X}_{\mathcal{L}}}(\boldsymbol{x})}{p_{\mathcal{X}_{\mathcal{D}}}(\boldsymbol{x})}\right) p_{\mathcal{X}_{\mathcal{D}}}(\boldsymbol{x}) d\boldsymbol{x} + (\lambda_{l_{\mathcal{Y}}} \epsilon + \epsilon_{\mathcal{L}}) \left(1 - \int_{\substack{\mathcal{X}, \\ p_{\mathcal{X}_{\mathcal{D}}} \geq p_{\mathcal{X}_{\mathcal{L}}}}} p_{\mathcal{X}_{\mathcal{L}}}(\boldsymbol{x}) d\boldsymbol{x} \right) + \mathbb{E}_{p_{\mathcal{L}}}\left[l(\boldsymbol{X}, Y, m_{\mathcal{L}})\right]$$

$$\leq C \sup_{\boldsymbol{x} \in \mathcal{X}} \left(\min_{\boldsymbol{x}_j \in \mathcal{L}_{\mathcal{X}}} \|\boldsymbol{x} - \boldsymbol{x}_j\|_2 \left(1 - \frac{p_{\mathcal{X}_{\mathcal{L}}}(\boldsymbol{x})}{p_{\mathcal{X}_{\mathcal{D}}}(\boldsymbol{x})}\right)\right) + (\lambda_{l_{\mathcal{Y}}} \epsilon + \epsilon_{\mathcal{L}}) \left( \int_{\substack{\mathcal{X}, \\ p_{\mathcal{X}_{\mathcal{D}}} < p_{\mathcal{X}_{\mathcal{L}}}}} p_{\mathcal{X}_{\mathcal{L}}}(\boldsymbol{x}) d\boldsymbol{x} \right) + \mathbb{E}_{p_{\mathcal{L}}}\left[l(\boldsymbol{X}, Y, m_{\mathcal{L}})\right]$$

$$= C W_{\mathcal{L}_{\mathcal{X}}, \mathcal{X}} \left(p_{\mathcal{X}_{\mathcal{L}}} \| p_{\mathcal{X}_{\mathcal{D}}}\right) + (\lambda_{l_{\mathcal{Y}}} \epsilon + \epsilon_{\mathcal{L}}) \mathbb{P}_{\mathcal{L}}\left[p_{\mathcal{X}_{\mathcal{D}}} < p_{\mathcal{X}_{\mathcal{L}}}\right] + \mathbb{E}_{p_{\mathcal{L}}}\left[l(\boldsymbol{X}, Y, m_{\mathcal{L}})\right],$$

where $\mathbb{P}_{\mathcal{L}}\left[p_{\mathcal{X}_{\mathcal{D}}} < p_{\mathcal{X}_{\mathcal{L}}}\right] := \int_{\substack{\mathcal{X}, \\ p_{\mathcal{X}_{\mathcal{D}}} < p_{\mathcal{X}_{\mathcal{L}}}}} p_{\mathcal{X}_{\mathcal{L}}}(\boldsymbol{x}) d\boldsymbol{x} = \int_{\mathcal{X}} p_{\mathcal{X}_{\mathcal{L}}}(\boldsymbol{x}) \cdot \mathbf{1}\left\{p_{\mathcal{X}_{\mathcal{D}}}(\boldsymbol{x}) < p_{\mathcal{X}_{\mathcal{L}}}(\boldsymbol{x})\right\} d\boldsymbol{x}$ is the probability, under $p_{\mathcal{X}_{\mathcal{L}}}$,

that a randomly drawn sample is assigned higher likelihood by $p_{\mathcal{X}_{\mathcal{L}}}$ than by $p_{\mathcal{X}_{\mathcal{D}}}$. It quantifies the mismatch between $p_{\mathcal{X}_{\mathcal{L}}}$ and $p_{\mathcal{X}_{\mathcal{D}}}$. The last inequality follows from taking the supremum over $\boldsymbol{x} \in \mathcal{X}$, taking the constant terms out of the integral and noticing that $\int_{\substack{\mathcal{X}, \\ p_{\mathcal{X}_{\mathcal{D}}} \geq p_{\mathcal{X}_{\mathcal{L}}} \\ p_{\mathcal{X}_{\mathcal{D}}} > 0}} p_{\mathcal{X}_{\mathcal{D}}}(\boldsymbol{x}) d\boldsymbol{x} \leq \int_{\mathcal{X}} p_{\mathcal{X}_{\mathcal{D}}}(\boldsymbol{x}) d\boldsymbol{x} = 1.$ □

## D  Asymptotic unbiasedness of the density estimations

In this appendix we show that the $k$NN data-driven estimates $\hat{p}^k_{\mathcal{X}_{\mathcal{D}}}$ and $\hat{p}^k_{\mathcal{X}_{\mathcal{L}}}$, defined in (5) and derived from the finite sets $\mathcal{D}_{\mathcal{X}}, \mathcal{L}_{\mathcal{X}} \subset \mathbb{R}^d$, are asymptotically unbiased estimations of some densities $p_{\mathcal{X}_{\mathcal{D}}}$ and $p_{\mathcal{X}_{\mathcal{L}}}$, respectively, for any $2 \leq k < n$. We assume that $p_{\mathcal{X}_{\mathcal{D}}}$ and $p_{\mathcal{X}_{\mathcal{L}}}$ are uniformly continuous and that $\mathcal{D}_{\mathcal{X}}$ and $\mathcal{L}_{\mathcal{X}}$ consist of random samples drawn from $p_{\mathcal{X}_{\mathcal{D}}}$ and $p_{\mathcal{X}_{\mathcal{L}}}$, respectively. To show this we use results from Cacoullos (1966), which we recapture in the following theorem in a formulation that is more suitable for our purposes.

**Theorem D.1.** *Let us consider a $d$-dimensional Euclidean space $\mathcal{X}$ and $K : \mathcal{X} \to \mathbb{R}$, Borel function on $\mathcal{X}$, such that*

$$\sup_{\boldsymbol{x} \in \mathcal{X}} |K(\boldsymbol{x})| < \infty, \quad \int_{\mathcal{X}} |K(\boldsymbol{x})| d\boldsymbol{x} < \infty, \quad \int_{\mathcal{X}} K(\boldsymbol{x}) d\boldsymbol{x} = 1 \quad and \quad \lim_{\|\boldsymbol{x}\|_2 \to \infty} \|\boldsymbol{x}\|_2^d |K(\boldsymbol{x})| = 0. \tag{21}$$

*Additionally, let us consider a sequence of positive scalar numbers $\{r_i\}_{i=1}^{\infty} \subset \mathbb{R}$ such that $\lim_{i \to \infty} r_i = 0$. Let us also consider $\{\boldsymbol{x}_i\}_{i=1}^m \subset \mathcal{X}$, $m \in \mathbb{N}^+$, set of $m$ independent realization of a random variable $\boldsymbol{X}$ with uniformly continuous distribution density $p$. Then, for any $\boldsymbol{x} \in \mathcal{X}$ in the non-zero support of $p$ we have that*

$$p_m(\boldsymbol{x}) = \frac{1}{m(r_m)^d} \sum_{i=1}^m K\left(\frac{\boldsymbol{x} - \boldsymbol{x}_i}{r_m}\right) \tag{22}$$

*is an asymptotically unbiased estimator of $p(\boldsymbol{x})$, i.e., $\lim_{m \to \infty} \mathbb{E}_p[p_m(\boldsymbol{x})] = p(\boldsymbol{x})$.*

*Proof.* The theorem recaptures results from Theorem 3.1 of Cacoullos (1966), which is proved using results from Theorem 2.1 and Lemma 2.1 of the same paper. The sketch of the proof is as follows: First, fix $\boldsymbol{x} \in \mathcal{X}$, and let

$$\varepsilon_m(\boldsymbol{x}) := \frac{1}{(r_m)^d} K\left(\frac{\boldsymbol{x} - \boldsymbol{X}}{r_m}\right). \tag{23}$$

Then, under the mentioned assumptions on $K$ and $p$, given a positive integer $l$, it is possible to show that

$$\lim_{m \to \infty} r_m^{d(l-1)} \mathbb{E}_p[\varepsilon_m^l(\boldsymbol{x})] = p(\boldsymbol{x}) \int K^l(\boldsymbol{x}) d\boldsymbol{x}. \tag{24}$$

Next, the theorem follows from noting that

$$\mathbb{E}_p[p_m(\boldsymbol{x})] = \mathbb{E}_p[\varepsilon_m(\boldsymbol{x})]. \tag{25}$$

□

Next we provide a corollary of the above theorem showing that the data-driven $k$NN density estimations $\hat{p}^k_{\mathcal{X}_{\mathcal{D}}}$ and $\hat{p}^k_{\mathcal{X}_{\mathcal{L}}}$ are asymptotically unbiased estimators of $p_{\mathcal{X}_{\mathcal{D}}}$ and $p_{\mathcal{X}_{\mathcal{L}}}$, respectively, for any $2 \leq k < n$.

**Corollary D.2.** *Let us consider $\mathcal{X} \subset \mathbb{R}^d$ and the function $K : \mathbb{R}^d \to \mathbb{R}^+$ defined in (6). Let us also consider $\mathcal{D}_n, \mathcal{L}_b \subset \mathcal{X}$, $n, b \in \mathbb{N}^+$ such that $\mathcal{D}_n := \{\boldsymbol{x}_i\}_{i=1}^n$ and $\mathcal{L}_b := \{\bar{\boldsymbol{x}}_j\}_{j=1}^b$ are independent realizations of random variables $\boldsymbol{X}_{\mathcal{D}}$ and $\boldsymbol{X}_{\mathcal{L}}$ with uniformly continuous densities $p_{\mathcal{X}_{\mathcal{D}}}$ and $p_{\mathcal{X}_{\mathcal{L}}}$, respectively. Next, consider the*

points $\boldsymbol{x}, \bar{\boldsymbol{x}} \in \mathbb{R}^d$ in the non-zero support of $p_{\mathcal{X}_\mathcal{D}}$ and $p_{\mathcal{X}_\mathcal{L}}$, respectively, and the kNN data-driven density estimations $\hat{p}^k_{\mathcal{X}_{\mathcal{D}_n}}(\boldsymbol{x})$ and $\hat{p}^k_{\mathcal{X}_{\mathcal{L}_b}}(\bar{\boldsymbol{x}})$ defined as follows

$$\hat{p}^k_{\mathcal{X}_{\mathcal{D}_n}}(\boldsymbol{x}) := \frac{\sum\limits_{\boldsymbol{x}_i \in \mathcal{D}_n} K\left(\frac{\boldsymbol{x}-\boldsymbol{x}_i}{r^k_{b,n}(\boldsymbol{x})}\right)}{n\left(r^k_{b,n}(\boldsymbol{x})\right)^d} \quad and \quad \hat{p}^k_{\mathcal{X}_{\mathcal{L}_b}}(\bar{\boldsymbol{x}}) := \frac{\sum\limits_{\bar{\boldsymbol{x}}_j \in \mathcal{L}_b} K\left(\frac{\bar{\boldsymbol{x}}-\bar{\boldsymbol{x}}_j}{r^k_{b,n}(\bar{\boldsymbol{x}})}\right)}{b\left(r^k_{b,n}(\bar{\boldsymbol{x}})\right)^d}, \tag{26}$$

where, for each $\tilde{\boldsymbol{x}} \in \mathbb{R}^d$, $r^k_{b,n}(\tilde{\boldsymbol{x}}) := \min\left\{\min\limits_{\bar{\boldsymbol{x}}_j \in \mathcal{L}_b} \|\tilde{\boldsymbol{x}} - \bar{\boldsymbol{x}}_j\|_2 + \frac{\epsilon_\mathcal{X}}{b}, \rho_{k,n}(\tilde{\boldsymbol{x}})\right\}$, with the scalar $\epsilon_\mathcal{X} > 0$ arbitrary small and $\rho_{k,n}(\tilde{\boldsymbol{x}})$ the distance between $\tilde{\boldsymbol{x}}$ and its k-nearest neighbor in $\mathcal{D}_n$. Then we have that, for any $2 \le k < n$, $\hat{p}^k_{\mathcal{X}_{\mathcal{D}_n}}(\boldsymbol{x})$ and $\hat{p}^k_{\mathcal{X}_{\mathcal{L}_b}}(\bar{\boldsymbol{x}})$ are asymptotically unbiased estimators of $p_{\mathcal{X}_\mathcal{D}}(\boldsymbol{x})$ and $p_{\mathcal{X}_\mathcal{L}}(\bar{\boldsymbol{x}})$, respectively, i.e.,

$$\lim_{n\to\infty} \mathbb{E}_{p_{\mathcal{X}_\mathcal{D}}}[\hat{p}^k_{\mathcal{X}_{\mathcal{D}_n}}(\boldsymbol{x})] = p_{\mathcal{X}_\mathcal{D}}(\boldsymbol{x}) \quad and \quad \lim_{b\to\infty} \mathbb{E}_{p_{\mathcal{X}_\mathcal{L}}}[\hat{p}^k_{\mathcal{X}_{\mathcal{L}_b}}(\bar{\boldsymbol{x}})] = p_{\mathcal{X}_\mathcal{L}}(\bar{\boldsymbol{x}}). \tag{27}$$

*Proof.* First note that the function $K : \mathbb{R}^d \to \mathbb{R}^+$ defined in (6) satisfies all the requirements in (21). In particular, we have that

$$\sup_{\boldsymbol{x}\in\mathbb{R}^d} |K(\boldsymbol{x})| = \frac{1}{V_d} \quad and \quad \int_{\mathbb{R}^d} K(\boldsymbol{x})d\boldsymbol{x} = \frac{1}{V_d}\int_{B^d(0,1)} 1d\boldsymbol{x} = 1, \tag{28}$$

where $V_d$ is the volume of the $d$-dimensional unit ball, which we write as $B^d(0,1) := \{\boldsymbol{x} \in \mathbb{R}^d \text{ such that } \|\boldsymbol{x}\|_2 \le 1\}$. Moreover, we have that $|K(\boldsymbol{x})| = 0$ for $\|\boldsymbol{x}\|_2 \ge 1$, thus, $\lim\limits_{\|\boldsymbol{x}\|_2\to\infty} \|\boldsymbol{x}\|_2^d|K(\boldsymbol{x})| = 0$. The only thing left to show to apply Theorem D.1 to $\hat{p}^k_{\mathcal{X}_{\mathcal{D}_n}}(\boldsymbol{x})$ and $\hat{p}^k_{\mathcal{X}_{\mathcal{L}_b}}(\bar{\boldsymbol{x}})$ is that $\lim\limits_{b\to\infty} r^k_{b,n}(\bar{\boldsymbol{x}}) = 0$ and $\lim\limits_{n\to\infty} r^k_{b,n}(\boldsymbol{x}) = 0$.

To show that $\lim\limits_{b\to\infty} r^k_{b,n}(\bar{\boldsymbol{x}}) = 0$ it is sufficient to observe that $\lim\limits_{b\to\infty}\min\limits_{\bar{\boldsymbol{x}}_j \in \mathcal{L}_b} \|\tilde{\boldsymbol{x}} - \bar{\boldsymbol{x}}_j\|_2 = 0$ because as the number of samples in $\mathcal{L}_b$ increases, the distance between $\bar{\boldsymbol{x}}$ and its nearest neighbor in $\mathcal{L}_b$ decreases, tending to zero. Moreover, $\lim\limits_{b\to\infty} \frac{\epsilon_\mathcal{X}}{b} = 0$. Similarly, we have that $\lim\limits_{n\to\infty} r^k_{b,n}(\boldsymbol{x}) = 0$ from the fact that $\lim\limits_{n\to\infty} \rho_{k,n}(\boldsymbol{x}) = 0$ which follows from the observation that as the number of samples in $\mathcal{D}_n$ increases, the distance between $\boldsymbol{x}$ and its $k$-nearest neighbor decreases, tending to zero. Thus, we can apply Theorem D.1 to $\hat{p}^k_{\mathcal{X}_{\mathcal{D}_n}}(\boldsymbol{x})$ and $\hat{p}^k_{\mathcal{X}_{\mathcal{L}_b}}(\bar{\boldsymbol{x}})$ showing that they are asymptotically unbiased estimators of $p_{\mathcal{X}_\mathcal{D}}(\boldsymbol{x})$ and $p_{\mathcal{X}_\mathcal{L}}(\bar{\boldsymbol{x}})$, respectively, for any $2 \le k < n$.

$\square$

# E    Proof Theorem 7.1

*Proof.* Let us fix the budget $b \in \mathbb{N}^+$, and define, for each $i = 1, \ldots, b+1$, $\mathcal{L}_i := \{\boldsymbol{x}_1, \ldots, \boldsymbol{x}_i\}$ as the set of cardinality $i$ obtained after $i-1$ iterations of the "While" loop in line 3 of Algorithm 1. To simplify the notation, for the rest of the proof, for each $\mathcal{L}_\mathcal{X} \subset \mathcal{D}_\mathcal{X}$ we define $W_{\mathcal{L}_\mathcal{X}} := W^k_{\mathcal{L}_\mathcal{X}, \mathcal{D}_\mathcal{X}}$ omitting the $k$ and $\mathcal{D}_\mathcal{X}$ from the notation of the estimated weighted fill distance. The following proof consists of three main steps.

**Step 1**    The first step of the proof consists of showing that for each $1 \le i < b+1$ we have

$$W_{\mathcal{L}_{i+1}} \le W_{\mathcal{L}_i}.$$

To prove this first step we notice that $\mathcal{L}_{i+1} = \mathcal{L}_i \cup \boldsymbol{x}_{i+1}$. Thus, for each $\boldsymbol{x} \in \mathcal{D}_\mathcal{X}$ we have that

$$\min_{\boldsymbol{x}_j \in \mathcal{L}_{i+1}} \|\boldsymbol{x} - \boldsymbol{x}_j\|_2 \le \min_{\boldsymbol{x}_j \in \mathcal{L}_i} \|\boldsymbol{x} - \boldsymbol{x}_j\|_2.$$

This is because adding a point to the selected set $\mathcal{L}_i$ ensures that the distance from any $\boldsymbol{x} \in \mathcal{D}_\mathcal{X}$ to its closest selected element either remains the same or decreases. Consequently, $\omega^k_{\mathcal{L}_{i+1}}(\boldsymbol{x}) \le \omega^k_{\mathcal{L}_i}(\boldsymbol{x})$ also holds.

To see this, recall that $\omega^k_{\mathcal{L}_i}(\boldsymbol{x})$ is the number of data points in $\mathcal{D}_{\mathcal{X}}$ contained within the ball centered at $\boldsymbol{x}$ with radius $r^k_{\mathcal{L}_i}(\boldsymbol{x}) := \min \left\{ \min_{\boldsymbol{x}_j \in \mathcal{L}_i} \|\boldsymbol{x} - \boldsymbol{x}_j\|_2 + \frac{\epsilon_{\mathcal{X}}}{|\mathcal{L}_i|}, \rho_k(\boldsymbol{x}) \right\}$, where $\rho_k(\boldsymbol{x})$ is the distance between $\boldsymbol{x}$ and its $k$-th nearest neighbor and $\epsilon_{\mathcal{X}}$ positive scalar value, which we consider arbitrary small. Adding a point to the selected set $\mathcal{L}_i$ ensures that the distance between any $\boldsymbol{x} \in \mathcal{D}_{\mathcal{X}}$ and its closest selected element does not increase. As a result, the value of $r^k_{\mathcal{L}_i}(\boldsymbol{x})$ is non-increasing, which in turn implies that the weights $\omega^k_{\mathcal{L}_i}(\boldsymbol{x})$ are also non-increasing. Therefore, we have that for each $1 \leq i < b+1$

$$\min_{\boldsymbol{x}_j \in \mathcal{L}_{i+1}} \|\boldsymbol{x} - \boldsymbol{x}_j\|_2 \omega^k_{\mathcal{L}_{i+1}}(\boldsymbol{x}) \leq \min_{\boldsymbol{x}_j \in \mathcal{L}_i} \|\boldsymbol{x} - \boldsymbol{x}_j\|_2 \omega^k_{\mathcal{L}_i}(\boldsymbol{x}),$$

Since the above inequality holds for each $\boldsymbol{x} \in \mathcal{D}_{\mathcal{X}}$, by taking the maximum over the points in $\mathcal{D}_{\mathcal{X}}$ we prove the first claim.

**Step 2** The second step of the proof shows that for each $2 \leq i \leq b+1$ and $1 \leq l < m \leq i$ we have that

$$W_{\mathcal{L}_{i-1}} \leq \|\boldsymbol{x}_m - \boldsymbol{x}_l\|_2 \omega^k_{\mathcal{L}_{m-1}}(\boldsymbol{x}_m), \tag{29}$$

where $\boldsymbol{x}_m$ and $\boldsymbol{x}_l$ are the points selected by Algorithm 1 at the $m$-th and $l$-th iterations, respectively. To prove this second step we proceed by induction. For the base step we have $i = 2$, $m = 2$, $l = 1$. Then we have

$$W_{\mathcal{L}_1} = \max_{\boldsymbol{x} \in \mathcal{D}_{\mathcal{X}}} \|\boldsymbol{x} - \boldsymbol{x}_1\|_2 \omega^k_{\mathcal{L}_1}(\boldsymbol{x}) = \|\boldsymbol{x}_2 - \boldsymbol{x}_1\|_2 \omega^k_{\mathcal{L}_1}(\boldsymbol{x}_2)$$

which verifies the base step. The second equality follows from how the selection strategy in Algorithm 1 is defined. Next, let us assume the assumption in (29) is true for $i-1$. Then we have for each $1 \leq l < m \leq i-1$

$$W_{\mathcal{L}_{i-1}} \leq W_{\mathcal{L}_{i-2}} \leq \|\boldsymbol{x}_m - \boldsymbol{x}_l\|_2 \omega^k_{\mathcal{L}_{m-1}}(\boldsymbol{x}_m).$$

Where the first inequality follows from the first step of the proof and the second inequality is our inductive assumption. Now notice that for each $1 \leq r < i$ we have

$$W_{\mathcal{L}_{i-1}} = \min_{\boldsymbol{x}_j \in \mathcal{L}_{i-1}} \|\boldsymbol{x}_i - \boldsymbol{x}_j\|_2 \omega^k_{\mathcal{L}_{i-1}}(\boldsymbol{x}_i) \leq \|\boldsymbol{x}_i - \boldsymbol{x}_r\|_2 \omega^k_{\mathcal{L}_{i-1}}(\boldsymbol{x}_i),$$

which proves the inductive step.

**Step 3** Consider now a set $\mathcal{C} \subset \mathcal{D}_{\mathcal{X}}$, with $|\mathcal{C}| = b$. Observe that by the definition of weighted fill distance we have that for each $\boldsymbol{x} \in \mathcal{D}_{\mathcal{X}}$ there exists $\boldsymbol{c} \in \mathcal{C}$ such that $\omega^k_{\mathcal{C}}(\boldsymbol{x}) \|\boldsymbol{x} - \boldsymbol{c}\|_2 \leq W_{\mathcal{C}}$. Next, notice that given $\mathcal{L}_{b+1}$, with $|\mathcal{L}_{b+1}| = b+1$, selected with Algorithm 1, by the pigeonhole principle we have that there exists $\boldsymbol{x}_m, \boldsymbol{x}_l \in \mathcal{L}_{b+1}$ with $1 \leq l < m \leq b+1$ that have a common closest element $\bar{\boldsymbol{c}} \in \mathcal{C}$. Therefore, $\max\{\|\boldsymbol{x}_m - \bar{\boldsymbol{c}}\|_2 \omega^k_{\mathcal{C}}(\boldsymbol{x}_m), \|\boldsymbol{x}_l - \bar{\boldsymbol{c}}\|_2 \omega^k_{\mathcal{C}}(\boldsymbol{x}_l)\} \leq W_{\mathcal{C}}$. Thus, we have

$$\begin{aligned}
W_{\mathcal{L}_b} &\leq \|\boldsymbol{x}_m - \boldsymbol{x}_l\|_2 \omega^k_{\mathcal{L}_{m-1}}(\boldsymbol{x}_m) \\
&\leq (\|\boldsymbol{x}_m - \bar{\boldsymbol{c}}\|_2 + \|\boldsymbol{x}_l - \bar{\boldsymbol{c}}\|_2) \, \omega^k_{\mathcal{L}_{m-1}}(\boldsymbol{x}_m) \\
&\leq (\|\boldsymbol{x}_m - \bar{\boldsymbol{c}}\|_2 \omega^k_{\mathcal{C}}(\boldsymbol{x}_m) + \|\boldsymbol{x}_l - \bar{\boldsymbol{c}}\|_2 \omega^k_{\mathcal{C}}(\boldsymbol{x}_l)) \, \omega^k_{\mathcal{L}_{m-1}}(\boldsymbol{x}_m) \\
&\leq 2\omega^k_{\mathcal{L}_{m-1}}(\boldsymbol{x}_m) W_{\mathcal{C}} \\
&\leq 2k W_{\mathcal{C}}.
\end{aligned} \tag{30}$$

The first inequality follows from the second step of the proof, the second inequality follows from the triangular inequality of the distance considered, the third and fifth inequalities follow from the fact that, by its definition, we have $1 \leq \omega^k_{\mathcal{C}}(\boldsymbol{x}) \leq k$ for all $\boldsymbol{x} \in \mathcal{D}_{\mathcal{X}}$ and $\mathcal{C} \subset \mathcal{D}_{\mathcal{X}}$. Since the above inequality holds for each $\mathcal{C} \subset \mathcal{D}_{\mathcal{X}}$, it holds for the optimal subset $O_{\mathcal{X}}$ as well. $\qquad\square$

## F  Computational efficiency DA-FPS

Given the novelty of DA-FPS, it is important to investigate its computational complexity. DA-FPS can be implemented using $\mathcal{O}(|\mathcal{D}|k)$ memory and the greedy selection takes $\mathcal{O}(db|\mathcal{D}|k)$ time. $|\mathcal{D}|$ is the amount of available data points, $k$ the amount of nearest neighbors we consider for the density approximation, $b$ is the amount of points we select and $d$ is the dimension of the data points. The computational cost of DA-FPS is determined by the weights update (line 7 in Algorithm 1) taking $\mathcal{O}(d|\mathcal{D}|k)$ at each of the $b$ iterations. In the current implementation the weights update involves iterating over all points in $\mathcal{D}$ and compute the distances between the new selected point and the points' $k$-nearest neighbors, which costs $\mathcal{O}(d|\mathcal{D}|k)$. Note that, initializing DA-FPS requires the computation of the $k$-nearest neighbors matrix, which can be a potential bottleneck. In our implementation we query the $k$-nearest neighbors using the cKDtree algorithm from the SciPy python library (Virtanen et al., 2020). The algorithm takes $\mathcal{O}(d|\mathcal{D}|\log|\mathcal{D}|)$ for building the balanced tree in the worst case scenario. After that, it queries the $k$-nearest neighbors with a worst-case cost of $\mathcal{O}(|\mathcal{D}|^{1-\frac{1}{d}})$ and an average cost of $\mathcal{O}(\log|\mathcal{D}|)$. Additionally, it is important to note that if the nearest neighborhood size to compute the weights as in (8) is set to $k = 1$, the optimization problem in (10) coincides with the fill distance minimization problem and Algorithm 1 reduces to the well known Farthest Point Sampling algorithm (FPS), thus providing 2-optimal solution (Har-Peled, 2011)., which is the best approximation factor attainable in polynomial time with theoretical guarantees (Hochbaum & Shmoys, 1985).

We implemented two versions of DA-FPS: one using NumPy (van der Walt et al., 2011) and the other with PyTorch (Paszke et al., 2019). The average computation times (over five runs) to select 20% of data points from QM7, QM8, QM9, and ZINC are respectively as follows: NumPy - 6, 48, 1974, and 70 seconds; PyTorch - 4, 31, 968, and 33 seconds. This was conducted on a 48-core CPU with 384 GB RAM. DA-FPS was initialized with $u = 0$ and $k = 100$ independently of the dataset. DA-FPS' PyTorch implementation is faster than the NumPy implementation. This is because PyTorch can run computations exploiting multiple CPU cores. It uses libraries like OpenMP(Dagum & Menon, 1998) and Math Kernel Library (Wang et al., 2014) to perform operations on multiple CPU cores, leading to faster computations.

## G  Datasets

This appendix draws from (Climaco & Garcke, 2024) and provides additional information related to the datasets, preprocessing procedures and molecular descriptors used for the experiments reported in Section 8

QM7 (Blum & Reymond, 2009; Rupp et al., 2012) contains 7,165 small organic molecules (up to 23 atoms, including C, N, O, S). Each molecule is represented by the upper triangular entries of its Coulomb matrix (Rupp et al., 2012), resulting in a feature vector in $\mathbb{R}^{276}$. The regression target is the atomization energy (in eV), i.e., the energy required to separate all atoms in a molecule.

QM8 (Ruddigkeit et al., 2012; Ramakrishnan et al., 2015) contains 21,766 organic molecules with up to 8 heavy atoms. For each molecule it provides its SMILES representation. We use Mordred (Moriwaki et al., 2018) to compute 1,826 molecular descriptors from SMILES, set missing values to zero, remove 530 features with zero variance, and normalize all features to (0, 1).Thus, each molecule in QM8 is represented by a vector in $\mathbb{R}^{1296}$. We then apply PCA to reduce the feature dimension to 100. The regression target is the lowest singlet transition energy (E1, in eV), computed using the PBE0 functional.

QM9 (Ruddigkeit et al., 2012; Ramakrishnan et al., 2014) consists of 130,202 small organic molecules with up to 9 heavy atoms. For each molecule it provides its SMILES string and 19 computed properties. Following along (Climaco & Garcke, 2024) we preprocess QM9 by removing molecules that fail consistency checks, cannot be parsed by RDKit (Landrum, 2012), or have duplicate SMILES. Molecular features are computed using Mordred (Moriwaki et al., 2018), with missing values set to zero, features with zero variance removed, normalization to (0, 1), and PCA to 100 dimensions. The regression target is the HOMO-LUMO energy gap (in eV), an important indicator of molecular stability.

The ZINC dataset (Gómez-Bombarelli et al., 2018) consists of about 250,000 molecules with up to 38 heavy atoms selected from the ZINC database (Sterling & Irwin, 2015), which contains over 120 million purchasable

organic molecules. To reduce the computational effort of our analysis, we follow along Dwivedi et al. (2023) and consider a subset of the ZINC dataset. Specifically, we use a subset of ZINC consisting of 24000 molecules selected uniformly at random. The molecular representation we employ is based on the Mordred (Moriwaki et al., 2018) library, as for the QM8 and QM9 datasets. We normalize the features provided by the Mordred library, to scale them independently in the interval $(0, 1)$. Again, we set to zero all the descriptor values that Mordred could not compute, removed the features for which the values across the dataset have zero variance and applied PCA to reduce the dimension of the feature vectors to 100. The label value to predict is the water-octanal partition coefficient (LogP), describing the molecules' solubility.

### G.1  Datasets for additional experiments in Appendix K

In this section we provide a more detailed description of the additional datasets unrelated to quantum chemistry used for experiments in Appendix K, including information on the preprocessing procedures.

**The Concrete Compressive Strength dataset** (Yeh, 1998) downloaded from the UCI Machine Learning Repository (Dua & Graff, 2017) contains 1030 data points and is used for regression tasks. It includes eight features: the amount of cement, blast furnace slag, fly ash, water, superplasticizer, coarse aggregate, and fine aggregate, as well as the age of the concrete in days. We remove 34 data points having identical descriptors as at least one other point in the dataset, obtaining a reduced dataset of 996 data points. Furthermore, we normalize the features to scale them independently in the interval $(0, 1)$. The target variable is the compressive strength of the concrete, measured in megapascals (MPa). This dataset is used to test machine learning models to predict material properties.

In the additional experiments illustrated in Appendix K, we use the Twinning algorithm implementation from Vakayil & Joseph (2022), which selects subsets based on an integer $r$, the inverse of the partitioning ratio. Since the algorithm strictly partitions the dataset according to this ratio, we remove 6 points from the Concrete dataset, resulting in a reduced dataset with 990 points. The points were selected randomly. This adjustment ensures that the subset size determined by the Twinning algorithm matches the percentages used to select the subsets, eliminating any discrepancies. Similarly, for the experiments in Appendix K, we remove 66 points from the QM8 dataset, creating a reduced dataset of 21700 points. QM8 is preprocessed with the same procedure used in Section 8.

**The Electrical Grid Stability Simulated dataset** from the UCI Machine Learning Repository (Dua & Graff, 2017) contains 10000 data points and is designed for both classification and regression tasks. Each data point in this dataset is represented by 12 features that describe characteristics of a simulated power grid. We normalize the features to scale them independently in the interval $(0, 1)$. For regression tasks, the target variable is the stability margin, which quantifies the power grid's stability.

## H  Data assumptions

We note that, for the experiments to be consistent with the theory the datasets we use should respect the data assumptions required in Theorem 4.2. We focus on Assumption B.2, more specifically Formula (16), indicating that if two data points have close representations in the feature space, then the conditional expectations of the associated labels are also close. This assumption is necessary to attain the theoretical result in Theorem 4.2.

For the QM datasets, the data assumptions have been already tested in Climaco & Garcke (2024) where the authors perform experiments related to FPS. However, for the experiments with DA-FPS we use different feature vectors. Thus, it is worth to verify whether the new feature vectors we consider still respect the required assumptions or not. We use the same procedure employed in Climaco & Garcke (2024) and study the correlation between pairwise distances in the feature and labels spaces by computing the Pearson's ($\rho_p$) and Spearman's ($\rho_s$) correlation coefficients. We recall that these coefficients measure and quantify the correlation between the quantities of interest. We compute the correlation coefficients for the pairwise distances in the feature and label spaces on the QM7, QM8, QM9 and ZINC. We consider the features and labels used for the experiments illustrated in Section 8 and Appendix M. Due to memory issue in storing the distance matrix, for the QM9 we computed the correlation coefficients on 50% of randomly selected data

points. We selected random subsets and computed the associated correlation coefficients for five times. The results we report for the QM9 are the average over the five runs. The computed coefficients are 0.15, 0.22, 0.26 and 0.22 for $\rho_p$, and 0.27, 0.19, 0.22 and 0.19 for $\rho_s$, for QM7, QM8, QM9 and ZINC, respectively. Thus, both the Pearson's and Spearman's coefficients indicate a positive correlation between the pairwise distances of the data features and labels, suggesting that the data assumption considered in Theorem 4.2 is respected for each of the considered datasets.

## I   Regression models: KRR and FNN

In this work, we follow along Climaco & Garcke (2024) and use two regression models already used in previous works for molecular property prediction: kernel ridge regression with a Gaussian kernel (KRR) and feed-forward neural networks (FNN). In this appendix we recapture the key aspects of the KRR and FNN regression models. For a more in depth analysis of KRR and FNN for molecular property prediction see Deringer et al. (2021) and Pinheiro et al. (2020), respectively.

Kernel ridge regression (KRR) is a machine learning method that combines ridge regression with kernel functions to perform regression in a flexible, non-linear way (Deringer et al., 2021). In this work, we use the Gaussian kernel. Given two data points $\boldsymbol{x}_q, \boldsymbol{x}_l \in \mathcal{X}$, the Gaussian kernel is defined as $k(\boldsymbol{x}_l, \boldsymbol{x}_q) := e^{-\gamma \|\boldsymbol{x}_q - \boldsymbol{x}_l\|_2^2}$, where $\gamma > 0$ is a parameter that controls the width of the kernel. Suppose we have a training set $\mathcal{L} = \{(\boldsymbol{x}_j, y_j)\}_{j=1}^b$ and a set of weights $\boldsymbol{\alpha} = [\alpha_1, \alpha_2, \ldots, \alpha_b]^T \in \mathbb{R}^b$, the predicted label value $m_{\mathcal{L}, \boldsymbol{\alpha}}(\boldsymbol{x}) \in \mathbb{R}$ associated with a data location $\boldsymbol{x} \in \mathbb{R}^d$ of a new data point is defined as follows $m_{\mathcal{L}, \boldsymbol{\alpha}}(\boldsymbol{x}) := \sum_{j=1}^b \alpha_j k(\boldsymbol{x}, \boldsymbol{x}_j)$. The scalar $m_{\mathcal{L}, \boldsymbol{\alpha}}(\boldsymbol{x})$ is the label predicted by the KRR method associated with the training data locations $\{\boldsymbol{x}_j\}_{j=1}^b$ and weights $\boldsymbol{\alpha}$. The goal of KRR is to find weights $\boldsymbol{\alpha} = [\alpha_1, \alpha_2, \ldots, \alpha_b]^T$ that minimize the following objective:

$$\boldsymbol{\alpha} = \arg\min_{\bar{\boldsymbol{\alpha}}} \sum_{j=1}^b (m_{\mathcal{L}}(\boldsymbol{x}_j) - y_j)^2 + \lambda \bar{\boldsymbol{\alpha}}^T \boldsymbol{K}_{\mathcal{L}} \ \bar{\boldsymbol{\alpha}}, \tag{31}$$

where $\boldsymbol{K}_{\mathcal{L}}$ is the kernel matrix with entries $\boldsymbol{K}_{\mathcal{L}}(q, r) = k(\boldsymbol{x}_q, \boldsymbol{x}_r)$, and $\lambda > 0$ is a regularization parameter that helps prevent overfitting. The solution to this problem is given by: $\boldsymbol{\alpha} = (\boldsymbol{K}_{\mathcal{L}} + \lambda \boldsymbol{I})^{-1} \boldsymbol{y}$, where $\boldsymbol{y} = [y_1, y_2, \ldots, y_b]^T$. To predict the label for a new data point $\boldsymbol{x} \in \mathcal{X}$, we use: $y(\boldsymbol{x}) := m_{\mathcal{L}}(\boldsymbol{x}) = \sum_{j=1}^b \alpha_j k(\boldsymbol{x}, \boldsymbol{x}_j)$. The hyperparameters $\gamma$ and $\lambda$ are selected by cross-validation grid search on random training subsets. For each training set size we select the best pairs of parameters. Next, we compute the average of the best pairs across all set sizes and use it for testing. The cross-validation tensor-grid uses 6 values per parameter from $10^{-6}$ to $10^{-1}$. We use the same set of hyperparameters for all selection strategies and training set sizes so that the only variable affecting the model performances is the selected training set.

Feed-forward neural networks (FNNs) (Goodfellow et al., 2016) are a type of deep neural network used for regression. An FNN predicts the label $y(\boldsymbol{x})$ for input $\boldsymbol{x} \in \mathcal{X}$ by passing it through a sequence of layers. With $l$ layers, the output is:

$$y(\boldsymbol{x}) = \psi_l \circ \sigma_l \circ \psi_{l-1} \circ \sigma_{l-1} \circ \cdots \circ \psi_1(\boldsymbol{x}), \tag{32}$$

where each $\psi_i$ is an affine transformation ($\psi_i(\boldsymbol{x}) = \boldsymbol{W}_i \boldsymbol{x} + \boldsymbol{b}_i$) and each $\sigma_i$ is a nonlinear activation function (here, ReLU). Following (Pinheiro et al., 2020), we use $l = 3$ layers and ReLU activations. The weights $\boldsymbol{W}_i$ and biases $\boldsymbol{b}_i$ are learned by minimizing the mean absolute error on the training set. To train the FNN and learn the weight matrices $\boldsymbol{W}_i$ and biases $\boldsymbol{b}_i$, we use the PyTorch (Paszke et al., 2019) Adam optimizer with a learning rate of 0.001, betas (0.9, 0.999), and weight decay 0.001. We use a batch size of 516 and train for 1000 epochs, regardless of the dataset. To ensure that performance differences are only due to the choice of training set, we always initialize the FNN with the same random weights.

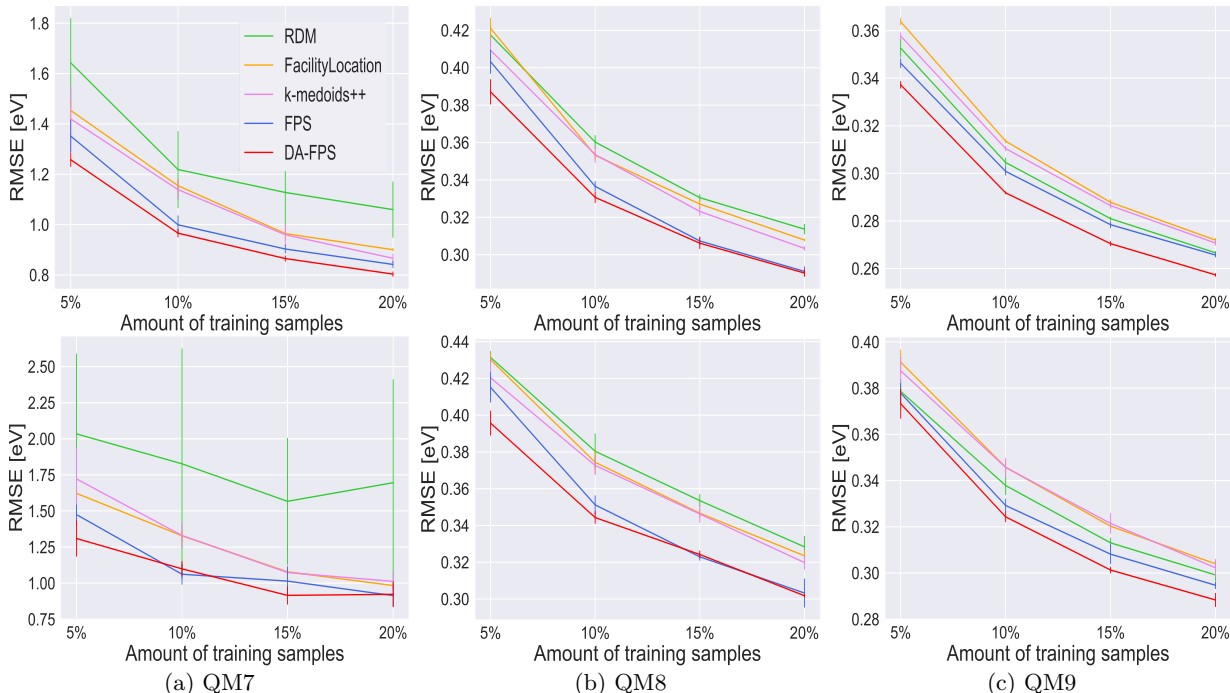

Figure 3: RMSE for regression tasks on QM datasets using KRR with Gaussian kernel (top row) and FNN (bottom row) trained on sets of various sizes, expressed as a percentage of the available data points, and selected with different sampling strategies. Error bars represent the standard deviation over five runs. The performances of FPS (blue lines) may be close to that of DA-FPS (red lines) when we consider the RMSE, particularly for larger training set sizes, e.g., on QM7 and for larger set sizes on QM8. Nevertheless, DA-FPS still leads to the most competitive performances across datasets. The legend in the leftmost graph in the top row applies to all graphs. DA-FPS is initialized with $\mathcal{L}_\mathcal{X} = \emptyset$, $k = 100$, and $u = 3\%$ of the available data, independently of the dataset.

## J Evaluation DA-FPS using the root mean squared error (RMSE)

In this appendix we use the root mean squared error (RMSE) to evaluate the performance of DA-FPS against the baselines used in Section 8. Given true target values $\{y_i\}_{i=1}^{n_u}$ and the predicted values $\{\tilde{y}_i\}_{i=1}^{n_u}$ the RMSE is defined as $\text{RMSE} := \sqrt{\frac{1}{n_u} \sum_{i=1}^{n_u} |y_i - \tilde{y}_i|^2}$, where $n_u$ is the number of unlabeled points in the data pool. Similarly to the MAE, the RMSE provides information on the average quality of the prediction. The difference is in the fact that RMSE penalizes large errors more than the MAE, thus providing insight on the robustness of the predictions. The graphs in Fig. 3 compare the performances of DA-FPS with the baselines and illustrate the RMSE of the predictions for the KRR (top row) and FNN (bottom row). The illustrated results suggest that the performances of FPS may be closer to that of DA-FPS when we consider the RMSE. This is particularly evident on the smaller QM7 and for larger set sizes on the QM8. Note that the RMSE penalizes large errors more than the MAE, thus giving more relevance to outlier error values. As we know from Climaco & Garcke (2024), FPS leads to a substantial decrease in maximum prediction error and hence reduces the amount and magnitude of large error values. Nevertheless, DA-FPS still leads to competitive performances in terms of the RMSE, highlighting its robustness against large errors.

## K Additional experiments

In this section we present experiments considering two additional datasets unrelated to quantum chemistry, two additional sampling strategies, and one additional kernel method. The new datasets are the Concrete Compressive Strength dataset and the Electrical Grid Stability Simulated Dataset, from the public UCI ML repository (Dua & Graff, 2017). The Electrical Grid Stability dataset contains 10000 data points represented

by 12-dimensional vectors. The target variable for regression is the stability margin (stab), reflecting grid stability. The Concrete dataset consists of 1030 data points described by 8-dimensional vectors. The target variable is the compressive strength in megapascals (MPa). In Section G.1 we provide additional details on the datasets and preprocessing procedures. We also consider the QM8 to evaluate the performance of the additional sampling strategies and regression model in a quantum chemistry context.

The additional sampling strategies are the Twinning algorithm (Vakayil & Joseph, 2022), and the facility location with a Gaussian similarity function (FacLocG) as defined in Bhatt et al. (2024). Fine-tuning is required for the Gaussian width of the similarity function in facility location, and we follow the methodology outlined in Bhatt et al. (2024), with complete details and hyperparameters described in Section M.1 of this appendix. Section M.1 also includes the hyperparameters used for DA-FPS. In these experiments, we compare DA-FPS with the additional sampling strategies alongside the baseline methods used in the previous section, including RDM, $k$-medoids++, facility location and FPS. The Twinning algorithm implementation from Vakayil & Joseph (2022) only allows the selection of subsets of the size that can be expressed as an integer "$r$" representing the inverse of the partitioning ratio, i.e., for obtaining a training subset consisting of 20% of the data points, we must set $r = \frac{100}{20} = 5$. Consequently, to use the Twinning algorithm, we consider training set sizes similar to those considered in the previous section, but that can be expressed as an integer ratio. Specifically, we consider training set sizes of 5%, 10%, 16.67% and 20% associated with a ratio $r$ of 20, 10, 6 and 5, respectively.

We use KRR with a Cauchy kernel as an additional regression method. We take the definition of Cauchy kernel used in Basak (2008). We describe the Cauchy kernel, its hyperparameters and the hyperparameters' optimization process in later in Section M.2. For each sampling strategy and set size, the training set selection process is independently run five times considering different initializations, that is, different initial point or random seed. Accordingly, we report for each analyzed model the average and the standard deviation of the results for five runs. We consider three distinct evaluation metrics to analyze prediction performance. We use the MAE and RMSE, introduced in Section 8 and Appendix J, respectively, which quantify the average quality of the prediction. In addition, we also compute the Maximum Absolute Error (MAXAE) of the predictions. The MAXAE is defined as MAXAE:$= \max_{1 \leq i \leq n_u} |y_i - \tilde{y}_i|$, where $y_i$ and $\tilde{y}_i$ are the true and predicted values, respectively. The MAXAE provides information on the worst-case scenario, and thus on the robustness of the model's predictions. Fig. 4 presents the regression task results on the Concrete dataset, the Electrical Grid dataset, and the QM8 dataset, using KRR with a Cauchy kernel trained on datasets of various sizes, selected with different strategies. The top row of the figure shows the MAE of the predictions, the primary metric of interest. These results suggest that, overall, DA-FPS outperforms other methods across all datasets. The only exception occurs with a training set size of 5% on the QM8 dataset, where the Twinning approach performs better. Generally, the MAE plots indicate that Twinning is the second-best method regarding MAE. The middle row of Fig. 4 presents the RMSE results, which indicate that DA-FPS consistently achieves strong performance, ranking as either the best or second-best method across all scenarios. Unlike the MAE results, where Twinning is the second best-performing approach, the RMSE results highlight FPS as the method most closely aligned with DA-FPS in terms of performance. This distinction underscores the reliability of DA-FPS across varying evaluation metrics. Since RMSE assigns greater weight to larger prediction errors compared to MAE, these findings emphasize the capability of DA-FPS to manage and mitigate significant prediction errors effectively. The bottom row of Fig. 4 illustrates the MAXAE, further indicating DA-FPS as either the best or second-best method in every scenario. FPS emerges as the other best performing. It is worth to note that while the Twinning approach is the second best performing approach it terms of the MAE, it performs poorly in terms of MAXAE, especially when compared to DA-FPS and FPS. For example, Twinning is the worst-performing algorithm on the Concrete dataset with a training set size of 5% and the second-worst on the QM8 dataset, regardless of training set size. In both cases, Twinning performs even worse than random sampling. The results in Fig. 4 underscore the effectiveness and adaptability of DA-FPS across different datasets, sampling strategies, and error metrics. These additional experiments further indicate that DA-FPS is a robust and versatile approach for various regression tasks not only constrained to the quantum-chemistry domain.

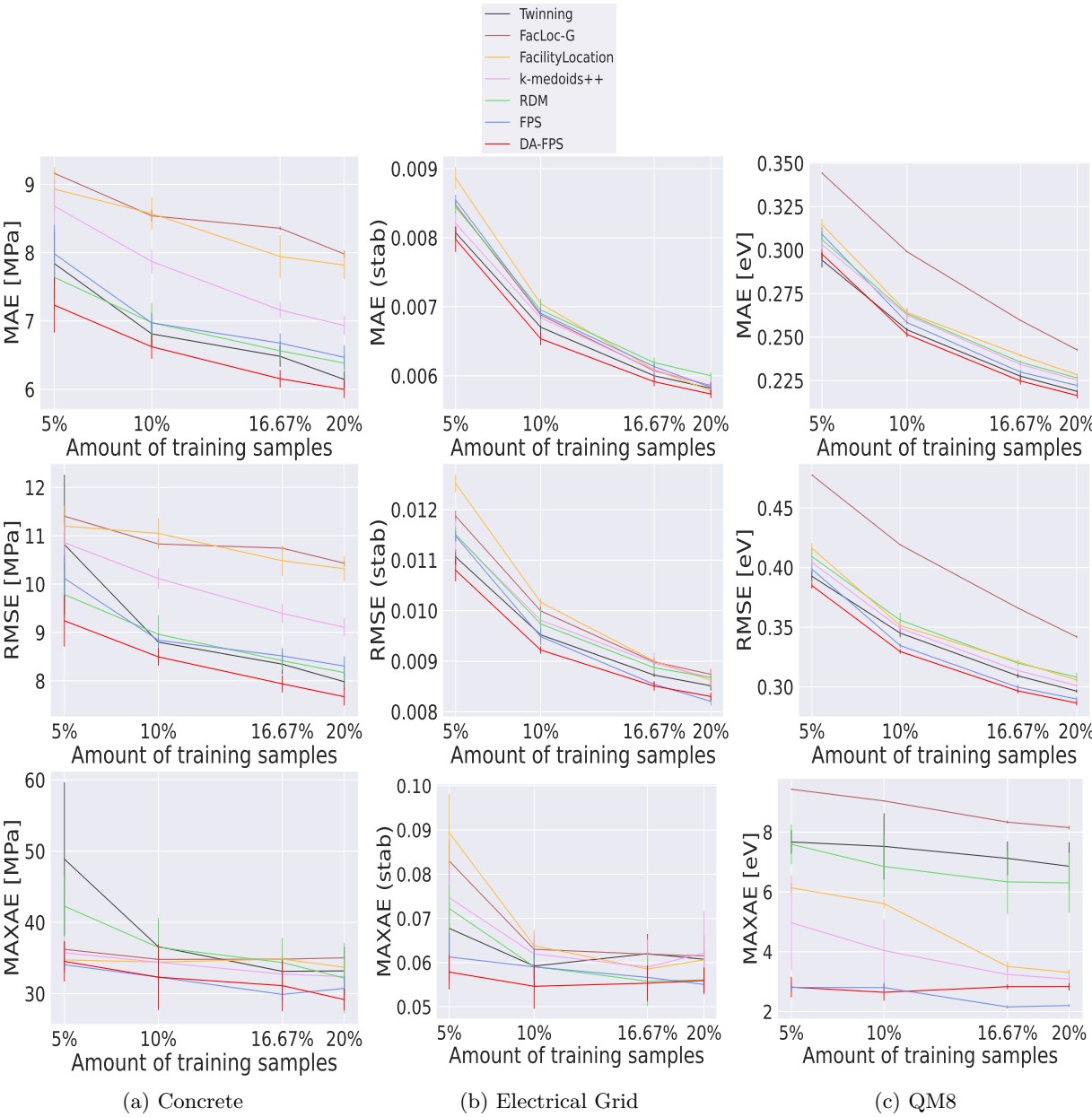

Figure 4: Results for regression tasks on the Concrete Compressive Strength, Electrical Grid Stability, and QM8 datasets. We use KRR with the Cauchy kernel trained on sets of various sizes, expressed as a percentage of the available data points, and selected with different sampling strategies. MAE (top row), RMSE (middle row) and MAXAE (bottom row) are shown for each training set size and sampling approach. Error bars represent the standard deviation of the results over five runs. DA-FPS (red lines) consistently showcases competitive performances across all metrics. For MAE, DA-FPS generally outperforms other methods, except Twinning (black lines) at 5% training set size on QM8. Twinning is the second-best method in terms of the MAE. For the RMSE, DA-FPS consistently ranks as the best or second-best. MAXAE results confirm DA-FPS as the best or second-approach, with FPS (blue lines) as the other most competitive approach. As for Twinning, despite strong MAE performance, it under-performs in MAXAE, sometimes worse than random sampling (green lines). Overall, DA-FPS delivers competitive performances across all metrics. DA-FPS is initialized with $\mathcal{L}_\mathcal{X} = \emptyset$ and $u = 3\%$, 1% and 3% and $k = 100$, 300 and 300 for the QM8, Concrete dataset and electricity dataset, respectively.

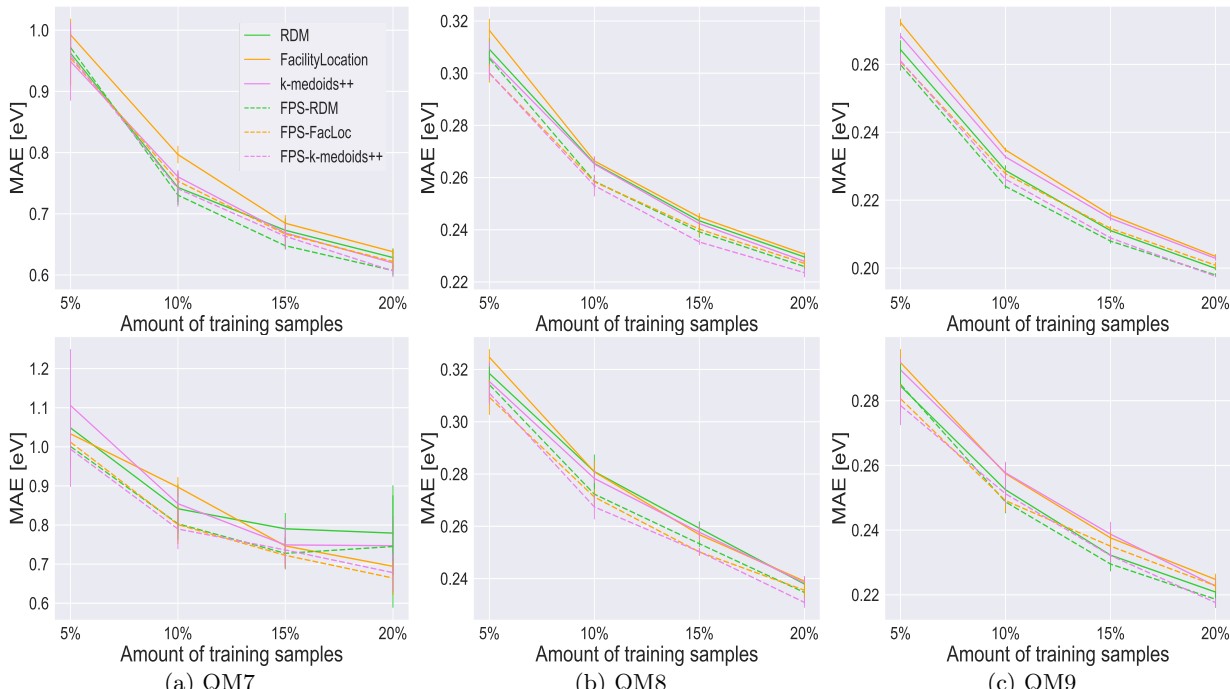

Figure 5: MAE for regression tasks on QM datasets using KRR with Gaussian kernel (top row) and FNN (bottom row) trained on sets of various sizes, expressed as a percentage of the available data points, and selected with different sampling strategies. Error bars represent the standard deviation over five runs. The modified versions of the baselines (dashed lines) lead to better performance than the respective original baselines (solid lines). The legend in the leftmost graph in the top row applies to all graphs. The modified baselines sample the first 3% of the points using FPS.

## L   Combining FPS with baselines

In this appendix we empirically investigate the benefits of combining the baseline sampling approaches used in Section 8 with FPS. We consider modified versions of the RDM, Facility location and $k$-medoids++. These modified versions of the baselines first select a prefixed amount of points with the FPS, the same amount we would consider for the DA-FPS, and then augment the selected set by sampling from the remaining points in the data pool according to their specific criteria. Consequently, in the early stage of the sampling process, the sets selected with our proposed approach coincide with those selected with FPS and the modified baselines. We refer to the modification of the baselines as FPS-RDM, FPS-FacLoc and FPS-$k$-medoids++. The experiments in this appendix are performed considering the same experimental set up as in Section 8.

Fig. 5 shows that FPS-RDM, FPS-FacLoc and FPS-$k$-medoids++ consistently outperform the associated baselines RDM, facility location and $k$-medoids++ in terms of the MAE for both, the KRR (top row) and FNN (bottom row). These results suggest that modifying the baselines by initially sampling with FPS leads to a reduction of the MAE independently of the dataset and trained model.

Fig. 6 compares DA-FPS with the modified baselines. Overall, DA-FPS leads to lower MAE of the regression models. Looking at the results in Fig. 6 obtained with FNN (bottom row), we can highlight two scenarios where DA-FPS is outperformed by one of the modified baselines: on the smaller QM7 and on the QM9 for the 5% training set size. These results suggest that, the advantages of using DA-FPS with respect to the modified baselines may be less evident on smaller datasets and training set sizes ($\leq 5\%$ of the available points), when using the FNN as learning model. Recall that, FNN predictions are prone to instability for lower training set sizes. Nonetheless, our experiments still indicate that, overall, DA-FPS is more competitive than the modified baselines in terms of the MAE of the predictions. In particular, no modified baseline

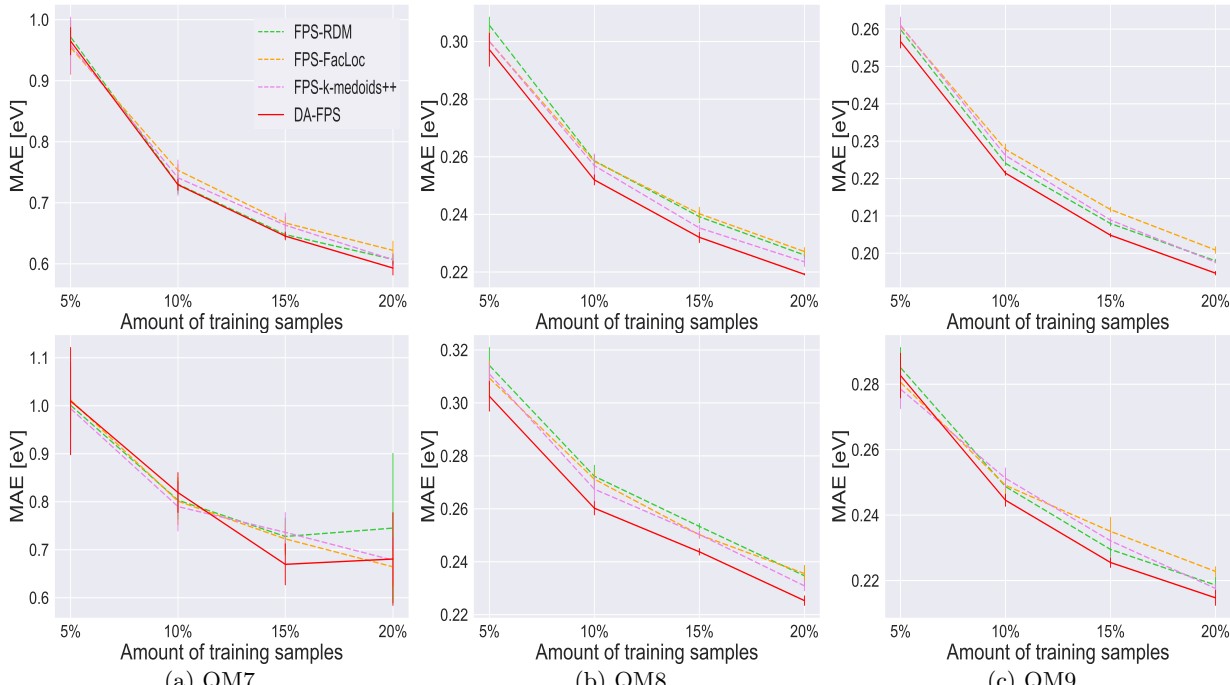

Figure 6: MAE for regression tasks on QM datasets using KRR with Gaussian kernel (top row) and FNN (bottom row) trained on sets of various sizes, expressed as a percentage of the available data points, and selected with different sampling strategies. Error bars represent the standard deviation over five runs. DA-FPS (red lines) outperforms the modified baselines. The legend in the leftmost graph in the top row applies to all graphs. DA-FPS is initialized with $\mathcal{L}_{\mathcal{X}} = \emptyset$, $k = 100$, and $u = 3\%$ of the available data, independently of the dataset. The modified baselines coincide with DA-FPS until 3% of the data is selected

can consistently outperform DA-FPS in any of the datasets considered. Moreover, according to our experiments, the comparative effectiveness of the modified baselines, in terms of the MAE, depends on the dataset considered, that is, on the underlying data distribution. For instance, in Fig. 6, if we consider KRR as the regression model (top row), FPS-RDM outperforms FPS-$k$-medoids++ on QM9. The opposite is true on QM8. The results with DA-FPS appear to be more robust to changes in the datasets.

## M Ablation study DA-FPS hyperparameters on ZINC dataset

In this appendix we analyze how the performances of DA-FPS may be affected as we vary the hyperparameter $u$, defining the amount of samples initially selected with uniform weights and the hyperparameter $k$, defining the amount of $k$-nearest neighbors considered for computing the weights. We perform this analysis on the ZINC dataset and use the KRR and FNN as regression models. We consider a version of the ZINC dataset consisting of 24000 molecules represented as vectors in $\mathbb{R}^{100}$. We aim to predict the molecules' LogP value, describing the molecules' solubility. In Appendix G we provide the details on the descriptors, label values, and preprocessing procedures we use. ZINC provides one additional application scenario to those already considered in Section 8.

### Hyperparameter "$u$"

To study how DA-FPS performs as the hyperparameter $u$ varies, we fix $k = 300$. The graphs in Fig. 7a illustrate how the performance of DA-FPS (dashed lines) changes as the parameter $u$ varies, compared to the baseline approaches (solid lines) for both, the KRR (top row) and FNN (bottom row). We consider $u = 0\%, 1\%, 2\%, 3\%$. For the low data budget of 5%, we see that random sampling and $k$-medoids++ lead to

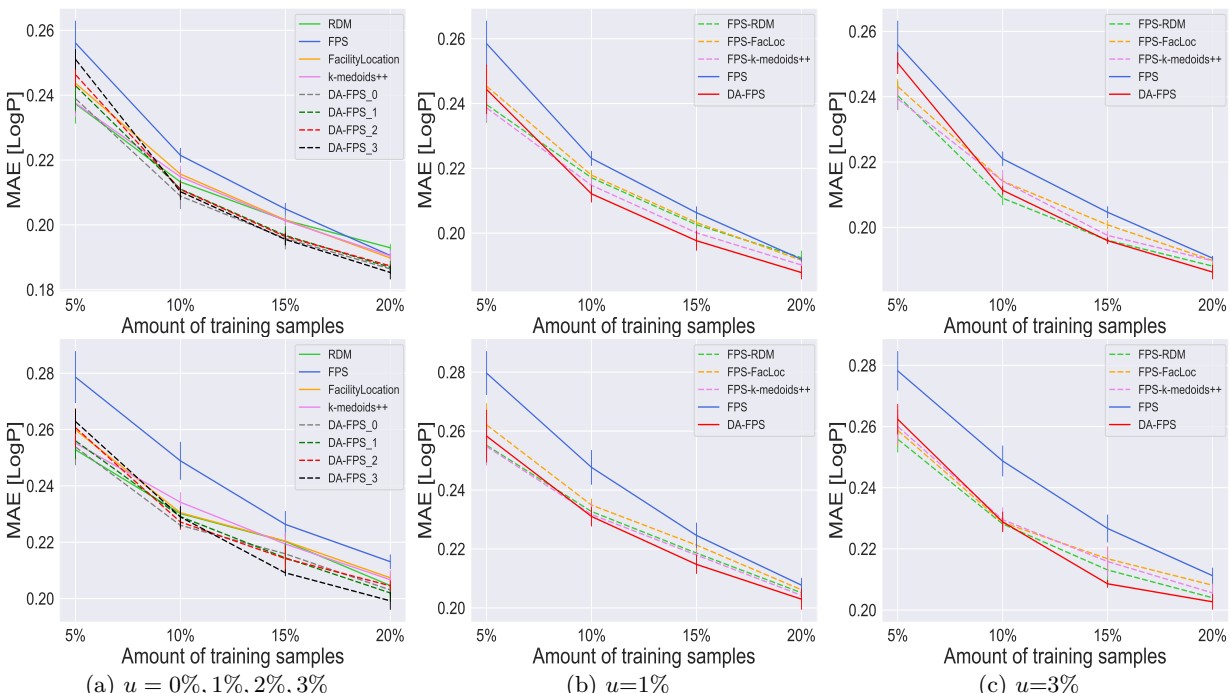

Figure 7: Sensitivity study DA-FPS hyperparameter "$u$": MAE for regression tasks on ZINC using KRR (top row) and FNN (bottom row) trained on sets of various sizes, expressed as a percentage of the available data points, and selected with different sampling strategies. DA-FPS is implemented with $k = 300$ and considering various values for $u$. In (a) DA-FPS is implemented with $u = 0\%, 1\%, 2\%, 3\%$. DA-FPS and the modified baselines are implemented with $u = 1\%$ in (b) and $u = 3\%$ in (c). Error bars represent the standard deviation over five runs. DA-FPS is initialized with $\mathcal{L}_{\mathcal{X}} = \emptyset$ and $k = 300$.

better results than DA-FPS, particularly if we consider larger values of $u$ for DA-FPS. This is more evident with KRR. On the contrary, for the larger training set sizes of 10%, 15% and 20%, DA-FPS consistently outperforms all the baselines independently of the choice of $u$. Moreover, for the smallest training set size of 5% we see that the smaller the value of $u$ the more accurate the average predictions obtained considering training set selected with DA-FPS, independently of the regression model. We note that, such a trend may change or even be reverted if we consider larger training set sizes of 10%, 15% and 20%. This is particularly evident in the graph of Fig. 7a related to the FNN model (bottom row), where, for training set sizes of 15% and 20%, the larger the value of $u$ the more accurate the average predictions with DA-FPS. Thus, from our experiments, we see that the parameter $u$ may affect the performance of DA-FPS differently, depending on the training set size and regression model considered. Nonetheless, for larger training set sizes the increased effectiveness of DA-FPS compared to the baselines is consistent across the different choices of $u$.

The graphs in Fig. 7b compare DA-FPS and the modified baselines. They show the results obtained by initializing DA-FPS and the modified baselines with $u = 1\%$ of the available data points. That is, DA-FPS and the modified baselines coincide with FPS until 1% of the available data points has been selected. The results align with those of the experiments performed in Section 8. In particular, the graphs in Fig. 7b show that, DA-FPS tends to outperform the modified baselines in terms of the MAE of the regression models, particularly for larger training set sizes ($> 5\%$).

Fig. 7c illustrates the performance of DA-FPS and the modified baselines considering $u = 3\%$ of the available data. The graphs suggest that, by increasing the parameter $u$ form 1% to 3% the gap between the modified baselines and DA-FPS reduces for KRR and increases FNN. This indicates that the choice of $u$ has an impact on the relative effectiveness of the modified baselines and DA-FPS and that such impact also depends on

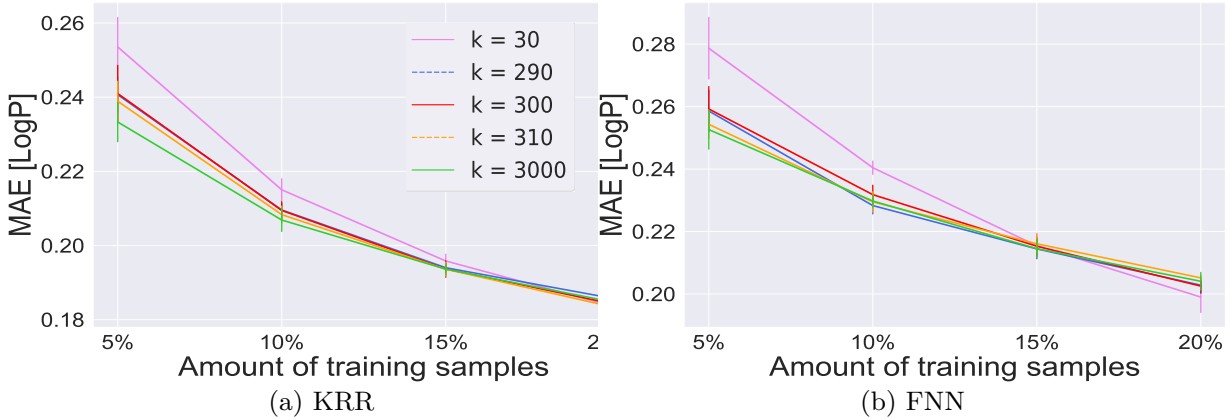

Figure 8: Sensitivity study DA-FPS hyperparameter "$k$": MAE for regression tasks on ZINC using KRR with Gaussian kernel (a) and FNN (b) trained on sets of various sizes, and selected with DA-FPS considering $u = 1\%$ and $k = 30, 290, 300, 310, 3000$. Error bars represent the standard deviation over five runs. The performance of DA-FPS remains stable when the hyperparameter $k$ is within a certain range ($k = 290, 300, 310$). Choosing $k$ too small may lead to a notable performance decline. Legend in the left graph applies to both graphs. DA-FPS is initialized with $\mathcal{L}_{\mathcal{X}} = \emptyset$ and $u = 1\%$ of the available data.

the model used for the regression task. Nonetheless, independently of the choice $u$, DA-FPS is the best or second best performing for the larger training set sizes ($> 10\%$).

**Hyperparameter "$k$"**

To study how DA-FPS performs as the hyperparameter $k$ varies, we fix $u=1\%$. We choose $u=1\%$ because, according to the results in Fig. 7, it provides the better overall results across the various training set sizes considered. Fig. 8 reports experiments where we investigate the effects of varying the DA-FPS parameter $k$ for regression tasks with KRR and FNN on the ZINC dataset. We select training sets of various sizes with DA-FPS, considering different values of $k$ ($k = 30, 290, 300, 310, 3000$).

These exemplary results suggest that the proposed value $k = 300$ (used for the experiments in Fig. 7) falls within a range where small changes would not negatively impact the effectiveness of our approach on the ZINC datasets. Using $k = 290$ and $310$ does not lead to substantial differences, independently of the training set size. However, applying changes of an order of magnitude to $k$ could potentially affect the approach. For instance, changing the value of $k = 300$ by a factor of 10 to $k = 30$ leads to a significant decrease in performance for the training set size of 5%, independently of the regression model. The rough magnitude of $k$ likely depends on the data dimension and distribution and is generally investigated in the context of density estimation. We expect further insights from that domain.

**M.1 Hyperparameter optimization for additional experiments**

In this section we follow along Bhatt et al. (2024) and describe the fine-tuning process for optimizing the hyperparameters of the facility location algorithm with the Gaussian similarity function. In addition, we report the hyperparameters considered for DA-FPS used for the additional experiments.

For the facility location method using the Gaussian similarity function, the fine-tuning process involves selecting an appropriate kernel width, denoted as $\gamma$, to prevent the function's gains from saturating when new data points are added to the training set. That is, given a finite pool of data points $\mathcal{D}_{\mathcal{X}} \subset \mathbb{R}^n$, a subset $S_k \subset \mathcal{D}_{\mathcal{X}}$, consisting of $k$ elements, and $f(S_k) := \sum_{\boldsymbol{x} \in \mathcal{D}_{\mathcal{X}}} \max_{\hat{\boldsymbol{x}} \in S_k} e^{-\gamma \|\boldsymbol{x} - \hat{\boldsymbol{x}}\|_2} \in \mathbb{R}^+$ value of the facility location function evaluated on $S_k$, we aim to choose $\gamma$ to maximize the gains $f(S_{k+1}) - f(S_k)$. The optimization procedure consists of computing the gains for various values of $\gamma$ and analyzing their behavior. The optimal value

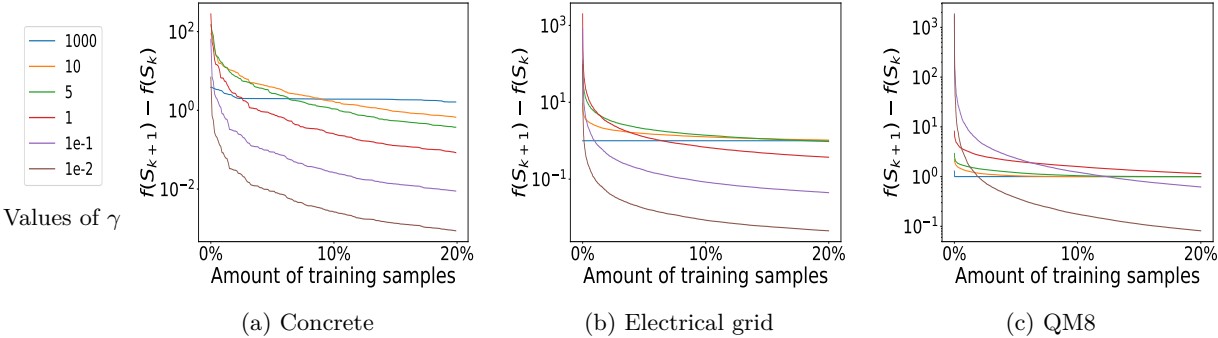

Figure 9: Gains from adding new elements to the selected sets for the QM8, Concrete, and Electrical Grid datasets using the facility location method with the Gaussian similarity function. Gains are shown for various values of the kernel width, $\gamma = 1000, 10, 5, 1, 0.1$ and $0.01$.

$\gamma$ is chosen to maximize gains for larger training sets while maintaining the ability to capture interactions between data points.

Figure 9 illustrates the gains obtained from adding new elements to the selected sets for the Concrete, Electrical grid, and QM8 datasets. We initialize the greedy selection process with the same data point, independently of the value of $\gamma$. Low values of $\gamma$ result in diminishing gains as the training set size grows, while excessively high values, such as $\gamma = 1000$, cause the kernel to approximate a diagonal matrix, failing to capture data point interactions. Based on the experimental results, we set $\gamma$ to 1 for QM8, 10 for the Concrete dataset, and 10 for the Electrical grid dataset.

For the DA-FPS we follow the same heuristic approach used for experiments in Section 8 and set $u = 3\%$, $1\%$ and $3\%$ and $k = 100, 300$ and $300$ for the QM8, Concrete dataset and electricity dataset, respectively.

### M.2  Cauchy Kernel

In this section we define the Cauchy kernel and describe the optimization process implemented to fine-tune the kernel hyperparameter and the regularization hyperparameter for the kernel ridge regression weights optimization problem. Given data points $\boldsymbol{x}_i, \boldsymbol{x}_j \in \mathbb{R}^d$, we follow along Basak (2008) and define the Cauchy kernel as follows:

$$k(\boldsymbol{x}_i, \boldsymbol{x}_j) = \frac{1}{1 + \left( \frac{\|\boldsymbol{x}_i - \boldsymbol{x}_j\|_2}{\gamma_c} \right)^2} \tag{33}$$

The hyperparameter $\gamma_c$ of the Cauchy kernel and $\lambda$, the regularization parameter of kernel ridge regression problem, are fine-tuned using the following process: first, we conduct a cross-validation grid search to identify the best hyperparameters for each training set size used in the experiments. The training subsets are obtained through random sampling. Then, we calculate the average of the best hyperparameter pairs across all training set sizes, which is subsequently used to build the final model. The tensor-grid search explores 6 points for each hyperparameter, ranging from $10^{-6}$ to $10^{-2}$. It is important to note that in our experiments we do not use an optimal set of hyperparameters for each selection strategy and training set size. This choice ensures that we focus on analyzing the qualitative behavior of a fixed model, where the only variable influencing the prediction quality is the selection of the training set.

## N   Alternative to Theorem 7.1

Theorem 7.1 states that DA-FPS provides a $2k$-optimal result to the optimization problem in (10), which considers dynamic-weights, that is, the weights are iteratively updated any time a new point is selected. We can consider a simplified version of the optimization problem in (10) by considering for each data point $\boldsymbol{x} \in \mathcal{D}_{\mathcal{X}}$ a fixed weight $\omega(\boldsymbol{x}) \in \mathbb{R}^+$, which is determined a-priori and does not depend on the selected set.

Such a simplified scenario has been already studied in literature. In particular, the authors of Dyer & Frieze (1985) show that it is possible to find solutions that are $\sigma$-optimal, with $\sigma := \min\{3, 1 + \alpha\}$, where $\alpha := \frac{\max_{\boldsymbol{x} \in \mathcal{D}_\mathcal{X}} \omega(\boldsymbol{x})}{\min_{\boldsymbol{x} \in \mathcal{D}_\mathcal{X}} \omega(\boldsymbol{x})}$. That is, it is possible to find solutions that are at least 3-optimal. With the following theorem we attempt to extend the result provided in Dyer & Frieze (1985) into our scenario with dynamic weights.

**Theorem N.1.** *Given set $\mathcal{D}_\mathcal{X} = \{\boldsymbol{x}_i\}_{i=1}^n \subset \mathbb{R}^d$, subset $O_\mathcal{X} \subset \mathcal{D}_\mathcal{X}$, optimal solution to the problem in (10) with $|O_\mathcal{X}| = b \in \mathbb{N}^+$, $b < n$, and $\mathcal{L}_\mathcal{X} \subset \mathcal{D}_\mathcal{X}$, $|\mathcal{L}_\mathcal{X}| = b$, subset selected with Algorithm 1 initialized with $\mathcal{L}_\mathcal{X} = \emptyset$ and $u = 0$, we have*

$$W_{\mathcal{L}_\mathcal{X}, \mathcal{D}_\mathcal{X}}^k \leq \sigma \gamma W_{O_\mathcal{X}, \mathcal{D}_\mathcal{X}}^k, \tag{34}$$

*where $W_{\mathcal{L}_\mathcal{X}, \mathcal{D}_\mathcal{X}}^k$ and $W_{O_\mathcal{X}, \mathcal{D}_\mathcal{X}}^k$ are the estimated weighted fill distances of $\mathcal{L}_\mathcal{X}$ and $O_\mathcal{X}$ in $\mathcal{D}_\mathcal{X}$, respectively, defined as in (11). Moreover,*

$$\gamma := \max_{j=1,\ldots,b+1} \frac{\omega_{\mathcal{L}_{j-1}}^k(\boldsymbol{x}_j)}{\omega_{O_\mathcal{X}}^k(\boldsymbol{x}_j)} \tag{35}$$

*and*

$$\sigma := \min\{3, 1 + \alpha\} \quad with \quad \alpha := \max_{\substack{i,j=1,\ldots,b+1 \\ i<j}} \frac{\omega_{\mathcal{L}_{j-1}}^k(\boldsymbol{x}_j)}{\omega_{\mathcal{L}_{i-1}}^k(\boldsymbol{x}_i)}. \tag{36}$$

*For each $j = 1,\ldots,b$, $\mathcal{L}_j := \{\boldsymbol{x}_1,\ldots,\boldsymbol{x}_j\}$ is the set of cardinality $j$ obtained with Algorithm 1. We set $\mathcal{L}_0 = \emptyset$. The weights $\omega_{\mathcal{L}_{j-1}}^k(\boldsymbol{x}_j)$ and $\omega_{O_\mathcal{X}}^k(\boldsymbol{x}_j)$ in (35) and (36) are computed according to the same principle as in (8).*

*Proof.* Let $O_\mathcal{X} := \{\boldsymbol{o}_1,\ldots,\boldsymbol{o}_b\}$ be an optimal solution to the optimization problem in (10). Moreover, let $\mathcal{L}_{b+1} := \{\boldsymbol{x}_1,\ldots,\boldsymbol{x}_{b+1}\}$ be the set of cardinality $b + 1$ obtained with Algorithm 1. First, note that by pigeonhole principle there exist $\boldsymbol{x}_i, \boldsymbol{x}_j \in \mathcal{L}_{b+1}$, with $1 \leq i < j \leq b + 1$ such that there exists a common closest element $\boldsymbol{o}_c \in O_\mathcal{X}$. Therefore, $\max\{\|\boldsymbol{x}_i - \boldsymbol{o}_c\|_2 \omega_{O_\mathcal{X}}^k(\boldsymbol{x}_i), \|\boldsymbol{x}_j - \boldsymbol{o}_c\|_2 \omega_{O_\mathcal{X}}^k(\boldsymbol{x}_j)\} \leq W_{O_\mathcal{X}, \mathcal{D}_\mathcal{X}}$. Next, we define the quantity

$$\beta := \frac{\omega_{\mathcal{L}_{j-1}}^k(\boldsymbol{x}_j)}{\omega_{\mathcal{L}_{i-1}}^k(\boldsymbol{x}_i)}$$

and consider two scenarios $\beta \leq 2$ and $\beta > 2$.

**First scenario:** $\beta \leq 2$. If we assume $\beta \leq 2$ we can prove the Theorem as follows

$$
\begin{aligned}
W_{\mathcal{L}_b, \mathcal{D}_\mathcal{X}}^k \leq & \omega_{\mathcal{L}_{j-1}}^k(\boldsymbol{x}_j) \min_{\boldsymbol{x} \in \mathcal{L}_{j-1}} \|\boldsymbol{x} - \boldsymbol{x}_j\|_2 \\
\leq & \omega_{\mathcal{L}_{j-1}}^k(\boldsymbol{x}_j) \|\boldsymbol{x}_i - \boldsymbol{x}_j\|_2 \\
\leq & \omega_{\mathcal{L}_{j-1}}^k(\boldsymbol{x}_j) \left(\|\boldsymbol{x}_i - \boldsymbol{o}_c\|_2 + \|\boldsymbol{x}_j - \boldsymbol{o}_c\|_2\right) \\
\leq & \frac{\omega_{\mathcal{L}_{j-1}}^k(\boldsymbol{x}_j)}{\omega_{O_\mathcal{X}}^k(\boldsymbol{x}_j)} \omega_{O_\mathcal{X}}^k(\boldsymbol{x}_j) \|\boldsymbol{x}_j - \boldsymbol{o}_c\|_2 \\
& + \frac{\omega_{\mathcal{L}_{j-1}}^k(\boldsymbol{x}_j)}{\omega_{\mathcal{L}_{i-1}}^k(\boldsymbol{x}_i)} \frac{\omega_{\mathcal{L}_{i-1}}^k(\boldsymbol{x}_i)}{\omega_{O_\mathcal{X}}^k(\boldsymbol{x}_i)} \omega_{O_\mathcal{X}}^k(\boldsymbol{x}_i) \|\boldsymbol{x}_i - \boldsymbol{o}_c\|_2 \\
\leq & \gamma(1 + \beta) W_{O_\mathcal{X}, \mathcal{D}_\mathcal{X}}^k \\
\leq & \sigma \gamma W_{O_\mathcal{X}, \mathcal{D}_\mathcal{X}}^k.
\end{aligned}
$$

The first inequality follows from the fact that $W_{\mathcal{L}_{i+1}, \mathcal{D}_\mathcal{X}}^k \leq W_{\mathcal{L}_i, \mathcal{D}_\mathcal{X}}^k$ for all $i = 1,\ldots,b$. This is shown in Step 1 of the proof of Theorem 7.1. The second inequality follows from the fact that $\boldsymbol{x}_i \in \mathcal{L}_{j-1}$ since $i < j$. The last inequality follows from the assumption that $\beta \leq 2$, thus, we have that $\beta \leq \min\{\alpha, 2\}$ which implies that $1 + \beta \leq \sigma$.

**Second scenario:** $\beta > 2$. Consider $1 = i < j \leq b+1$. Note that by how the weights are defined in (8), we have that $\omega^k_{\mathcal{L}_{i-1}}(\boldsymbol{x}_i) = \omega^k_{\mathcal{L}_0}(\boldsymbol{x}_1) = \omega^k_{\emptyset}(\boldsymbol{x}_1) = k$ and that for each $j = 2, \ldots, b+1$ we have $1 \leq \omega^k_{\mathcal{L}_{j-1}}(\boldsymbol{x}_j) \leq k$. Thus, if $1 = i < j \leq b+1$, it follows that

$$\beta = \frac{\omega^k_{\mathcal{L}_{j-1}}(\boldsymbol{x}_j)}{\omega^k_{\mathcal{L}_{i-1}}(\boldsymbol{x}_i)} = \frac{\omega^k_{\mathcal{L}_{j-1}}(\boldsymbol{x}_j)}{k} \leq 1,$$

which contradicts the assumption $\beta > 2$, so it holds $i > 1$. Next, consider $1 \leq l < i < j \leq b+1$, and $\boldsymbol{x}_l$ to be the closest point to $\boldsymbol{x}_j$ when $\boldsymbol{x}_i$ is chosen, then we have that

$$
\begin{aligned}
W^k_{\mathcal{L}_b, \mathcal{D}_{\mathcal{X}}} &\leq \omega^k_{\mathcal{L}_{j-1}}(\boldsymbol{x}_j) \min_{\boldsymbol{x} \in \mathcal{L}_{j-1}} \|\boldsymbol{x} - \boldsymbol{x}_j\|_2 \\
&\leq \omega^k_{\mathcal{L}_{j-1}}(\boldsymbol{x}_j) \|\boldsymbol{x}_l - \boldsymbol{x}_j\|_2 \\
&\leq \left|\left\{\bar{\boldsymbol{x}} \in \mathcal{D}_{\mathcal{X}} \text{ such that } \|\boldsymbol{x}_j - \bar{\boldsymbol{x}}\|_2 \leq \min\{\|\boldsymbol{x}_j - \boldsymbol{x}_l\|_2 + \frac{\epsilon_{\mathcal{X}}}{|\mathcal{L}_{i-1}|}, \rho_k(\boldsymbol{x}_j)\}\right\}\right| \|\boldsymbol{x}_l - \boldsymbol{x}_j\|_2 \\
&= \omega^k_{\mathcal{L}_{i-1}}(\boldsymbol{x}_j) \min_{\boldsymbol{x} \in \mathcal{L}_{i-1}} \|\boldsymbol{x} - \boldsymbol{x}_j\|_2 \\
&\leq \omega^k_{\mathcal{L}_{i-1}}(\boldsymbol{x}_i) \min_{\boldsymbol{x} \in \mathcal{L}_{i-1}} \|\boldsymbol{x} - \boldsymbol{x}_i\|_2 \\
&\leq \omega^k_{\mathcal{L}_{i-1}}(\boldsymbol{x}_i) \|\boldsymbol{x}_i - \boldsymbol{x}_l\|_2 \\
&\leq \omega^k_{\mathcal{L}_{i-1}}(\boldsymbol{x}_i) \left(\|\boldsymbol{x}_i - \boldsymbol{x}_j\|_2 + \|\boldsymbol{x}_j - \boldsymbol{x}_l\|_2\right).
\end{aligned}
\tag{37}
$$

The second inequality follows from the fact that $\boldsymbol{x}_l \in \mathcal{L}_{j-1}$, thus the distance between $\boldsymbol{x}_j$ and the closest element in $\mathcal{L}_{j-1}$ is smaller than $\|\boldsymbol{x}_j - \boldsymbol{x}_l\|_2$. This is also relevant for the third inequality: The value of the weight $\omega^k_{\mathcal{L}_{j-1}}(\boldsymbol{x}_j)$, which is the amount of data points in the ball centered in $\boldsymbol{x}_j$ with radius $r^k_{\mathcal{L}_{j-1}}(\boldsymbol{x}_j) := \min\left\{\min_{\boldsymbol{x} \in \mathcal{L}_{j-1}} \|\boldsymbol{x} - \boldsymbol{x}_j\|_2 + \frac{\epsilon_{\mathcal{X}}}{|\mathcal{L}_{j-1}|}, \rho_k(\boldsymbol{x}_j)\right\}$, is less or equal the amount of data points contained in the ball centered in $\boldsymbol{x}_j$ with the larger radius of $\min\{\|\boldsymbol{x}_j - \boldsymbol{x}_l\|_2 + \frac{\epsilon_{\mathcal{X}}}{|\mathcal{L}_{i-1}|}, \rho_k(\boldsymbol{x}_j)\}$. The equality follows from the fact that we assume $\boldsymbol{x}_l$ to be the closest point to $\boldsymbol{x}_j$ when $\boldsymbol{x}_i$ is chosen, that is, $r^k_{\mathcal{L}_{i-1}}(\boldsymbol{x}_j) = \min\{\|\boldsymbol{x}_j - \boldsymbol{x}_l\|_2 + \frac{\epsilon_{\mathcal{X}}}{|\mathcal{L}_{i-1}|}, \rho_k(\boldsymbol{x}_j)\}$. The fourth inequality is true because if we assume it was false then $\boldsymbol{x}_j$ would have been selected before $\boldsymbol{x}_i$.

If we now assume that $\|\boldsymbol{x}_i - \boldsymbol{x}_j\|_2 \leq \|\boldsymbol{x}_j - \boldsymbol{x}_l\|_2$ by the inequalities in (37) we would have

$$\omega^k_{\mathcal{L}_{j-1}}(\boldsymbol{x}_j) \|\boldsymbol{x}_j - \boldsymbol{x}_l\|_2 \leq 2\omega^k_{\mathcal{L}_{i-1}}(\boldsymbol{x}_i) \|\boldsymbol{x}_j - \boldsymbol{x}_l\|_2,$$

which implies that

$$\frac{\omega^k_{\mathcal{L}_{j-1}}(\boldsymbol{x}_j)}{\omega^k_{\mathcal{L}_{i-1}}(\boldsymbol{x}_i)} \leq 2,$$

which is a contradiction since we are assuming $\beta > 2$. Thus, we have that $\|\boldsymbol{x}_i - \boldsymbol{x}_j\|_2 > \|\boldsymbol{x}_j - \boldsymbol{x}_l\|_2$. Therefore, from equation (37), we have that

$$
\begin{aligned}
W^k_{\mathcal{L}_b, \mathcal{D}_{\mathcal{X}}} &\leq 2\omega^k_{\mathcal{L}_{i-1}}(\boldsymbol{x}_i) \|\boldsymbol{x}_i - \boldsymbol{x}_j\|_2 \\
&\leq 2\omega^k_{\mathcal{L}_{i-1}}(\boldsymbol{x}_i) \left(\|\boldsymbol{x}_i - \boldsymbol{o}_c\|_2 + \|\boldsymbol{x}_j - \boldsymbol{o}_c\|_2\right) \\
&\leq 2\frac{\omega^k_{\mathcal{L}_{i-1}}(\boldsymbol{x}_i)}{\omega^k_{O_{\mathcal{X}}}(\boldsymbol{x}_i)} \omega^k_{O_{\mathcal{X}}}(\boldsymbol{x}_i) \|\boldsymbol{x}_i - \boldsymbol{o}_c\|_2 \\
&\quad + 2\frac{\omega^k_{\mathcal{L}_{i-1}}(\boldsymbol{x}_i)}{\omega^k_{\mathcal{L}_{j-1}}(\boldsymbol{x}_j)} \frac{\omega^k_{\mathcal{L}_{j-1}}(\boldsymbol{x}_j)}{\omega^k_{O_{\mathcal{X}}}(\boldsymbol{x}_j)} \omega^k_{O_{\mathcal{X}}}(\boldsymbol{x}_j) \|\boldsymbol{x}_j - \boldsymbol{o}_c\|_2 \\
&\leq 2\gamma(1 + \beta^{-1}) W^k_{O_{\mathcal{X}}, \mathcal{D}_{\mathcal{X}}} \\
&\leq \sigma \gamma W^k_{O_{\mathcal{X}}, \mathcal{D}_{\mathcal{X}}}
\end{aligned}
$$

The last inequality follows from the facts that $\beta > 2 \Rightarrow \alpha > 2 \Rightarrow 1 + \alpha > 3 \Rightarrow \sigma = 3$. Thus, since $2(1 + \beta^{-1}) \leq 3$, we have that $\sigma \geq 2(1 + \beta^{-1})$. $\square$

With Theorem N.1 we provide an alternative result to Theorem 7.1 for the optimality of the solution provided by DA-FPS. Note that in Theorem N.1 we explicitly link the quality of approximation of DA-FPS with the ratio between the computed and optimal weights. In particular, one of the terms in the approximation factor is $\gamma := \max_{j=1,\dots,b+1} \frac{\omega^k_{\mathcal{L}_{j-1}}(\boldsymbol{x}_j)}{\omega^k_{O_{\mathcal{X}}}(\boldsymbol{x}_j)}$, that is, the ratio between the weights related to the set selected with DA-FPS and those of an optimal set. Note that the simplest upper bound for $\gamma$ is $\gamma \leq k$. This is because, for how we defined them in (8), the weights value is at least 1 and at the most $k$, independently of the set considered to define them. Thus, using the simplest upper bound for $\gamma$, according to Theorem N.1, DA-FPS achieves approximations that are $3k$-optimal. This rate represents a less favorable worst-case scenario compared to the $2k$-optimal rate given by Theorem 7.1. Nonetheless, we think this result is relevant because it highlights how the relationship between the weights an optimal set and those associated with the selected set can be connected to the approximation error of DA-FPS. Moreover, by explicitly linking the quality of DA-FPS's approximation to $\gamma$ and $\sigma$, we aim to highlight a possible path for improving the optimality constant by identifying a bound for either of these two quantities. Future work should focus on this direction.

