# OpenReview forum: "Density-Aware Farthest Point Sampling"
_TMLR — Accepted by TMLR_

### Review · Reviewer_2mgk · 2025-10-03

**Summary Of Contributions:**

This paper studies the problem of data selection for efficient training in the scenario where a large pool of unlabeled data is available and the learner wants to select a subset in order to obtain labels for it.

The proposed algorithm is based on designing a probability distribution to sample from to form the training set. The algorithm relies on a theorem that relates the excess error to a discrepancy measure referred to as weighted fill distance. Then, the algorithm constructs a distribution that (approximately) minimizes this weighted fill distance.

The authors provide experiments showing improvements compared to the baselines.

**Audience:**

Yes

**Audience Explanation:**

the topic is interesting.

**Broader Impact Concerns:**

not applicable.

**Claims And Evidence:**

No

**Claims Explanation:**

**Notation and presentation**
The notation is hard to follow. Please add a table of notation (for example: D_X, L_X, p_D, p_L, p_X^D, p_X^L, ψ_{L_X}, W(·), and the empirical/k-NN counterparts) and expand the discussion of your algorithmic choices (choice of metric, role of k, etc.). The current presentation is quite dense and would benefit from a short “roadmap” paragraph at the start of each section.

**Definition of the weighted fill distance**
Please clarify whether the weighted fill distance is a random variable or not. Let Q denote the training-source distribution induced by the selection policy and P the data-source distribution. If Q = P pointwise, then p_X^L = p_X^D and the weight ψ_{L_X}(x) = 1 − p_X^L(x)/p_X^D(x) vanishes everywhere, making the population weighted fill distance zero. Does this imply that sampling according to P is optimal? If not, please explain what fails in finite samples. In general, I think this confusion is another example of bad notation.

**Theorem 4.2 (tightness)**
When Q = P, the density term in the bound disappears and the remainder is driven by training-error and noise terms. In particular, it seems we are upper bounding a quantity by that quantity and some additional terms. It shows in trivial cases, the theorem is not tight. Please provide conditions under which this theorem is tight.

**Sampling distribution vs. active selection**
The introduction frames the method as designing a sampling distribution, but Algorithm 1 greedily selects individual points. Please reconcile these perspectives and define p_X^L precisely in your notation: is it the empirical measure on L_X or a k-NN–smoothed estimate?

**Numerical results**
The improvements over baselines appear marginal. To provide a baseline, please add a full-data (or 90%) ceiling: train the same model on all labels (or 90% train / 10% test) and report test performance. Please ensure validation handling is fair: use a fixed held-out validation set shared by all methods (not touched by the selector)

**Requested Changes:**

please see above.

---

> ### Author Response · Authors · 2025-11-04
> **We thank the reviewer for the time and the constructive comments. We found them very useful.**
>
> Notation and presentation:
> -  In the revised version, we added a notation table after the conclusion. This table is referenced at the end of the introduction. We also added a short "roadmap" paragraph at the start of each section (starting from section 3). We expanded the discussion of algorithmic choices in the last paragraph of Section 5.3. we addressed the role of k, the role of u, and the choice of metric. (See  point 5 general answers to reviewers)
>
> Definition of the weighted fill distance:
>  - In the revised version we clarify that the weighted fill distance is a deterministic quantity (first paragraph following definition 4.1.). The answer to the reviewer's question is yes, the  training data would be sampled "optimally" if they were sampled according to the data-source distribution. Optimal is meant as minimizing the weighted fill distance, which is the main goal of our sampling strategy. We note that the notion of optimality is not only connected to distributional matching, but also to geometric coverage of the nonzero support of $p_{X\_{D}}$ in the feature space. In the revised version, in the second paragraph following definition 4.1, we discussed possible implications of the weighted fill distance being zero. We included: "Note that the weighted fill distance can be zero under different circumstances. This occurs, for instance, when $\psi_{L\_{X}}(x)= 0$ for all $x\in X$, or when any point in the non-zero support of $p_{X\_{{D}}}$ is in $L\_{X}$[...]. In the first case, the fact that $\psi_{L\_{X}}(x)= 0$ for all $x \in X$ implies that the marginals of the training and data source distributions are perfectly aligned [...]. In the second case, the fact that each point in the non-zero support of $p_{X\_{D}}$ is in ${L}\_{X}$ implies that $L\_{X}$ already contains all points associated with data locations where a point from the data source distribution could arise. That is, the relevant regions of the data space are already included in the selected set."
>
> Theorem 4.2:
>  - We thank the reviewer for these valuable suggestion. We do not claim that our bound is tight. We explicitly characterize our result as a "qualitative" upper bound in the main text. The main goal of the bound is to identify the weighted fill distance as the main quantity to optimize. Nonetheless, the review observation clearly highlighted the importance of ensuring tightness at least for the case when $p_{X\_{L}} = p_{X\_{D}}$. This motivated us to improve the statement of Theorem 4.2 in the revised version. In particular, by simply applying a less coarse bound in the last step of the proof of theorem 4.2 we now multiply the training error and noise term by a factor that quantifies the distributional mismatch between $p_{X\_{L}}$ and $p_{X\_{D}}$. In the revised version, when $p_{X\_{L}} = p_{X\_{D}}$, the bound  reduces to upper bounding the prediction error by itself, which is the best achievable bound in this scenario. After Theorem 4.2 we discuss the implications of this improved bound.
>
> Sampling distribution vs. active selection:
> - In the revised version, we clarified that "Our goal is to design a data sampling strategy, inducing a training data distribution marginal $p_{{X}\_{L}}$, to sample a set $L\_{X}$ with minimal weighted fill distance." (Last sentence of Section 4). In our framework, $p_{X\_{L}}$ represents the distribution induced by our selection strategy. Since this distribution is not known explicitly, we estimate it using $\hat{p}^k_{X\_{L}}$, a k-NN smoothed estimate based on the currently selected points.
>  In the revised manuscript we also mention " In this work we assume we do not know a priori the explicit form of the training data distribution marginal $p_{X\_{L}}$ that we want to use to sample points[...]. Instead, we develop a data selection strategy that induces a training data distribution marginal $p_{X\_{L}}$, which we estimate from the selected $L\_{X}$ as $\hat{p}\_{X\_{L}}$. The selection procedure we propose aims at selecting a set $L\_{X}$ that minimizes a data-driven estimation of the weighted fill distance..." (2nd paragraph Section 5).
>
>
> Numerical results:
> - We appreciate this suggestion and agree that including a 90%-data baseline would provide a useful performance ceiling, indicating how close DA-FPS approaches the ideal case where most labels are available. However, our study specifically targets the low-data regime (<20% of labeled data), which reflects realistic scenarios where label acquisition is costly or computationally constrained. The 90%-data baseline represents an upper bound beyond our target setting as well as our computational resources for the case of large datasets like QM9. We ensure that the models is trained following the same procedure independently of the selected training set. We are after the relative performance. For KRR hyperparameters we optimized for random sampling.

---

### Review · Reviewer_pVQd · 2025-10-04

**Summary Of Contributions:**

The authors proposed density aware farthest point sampling, which extends farthest point sampling to be density aware so that higher density regions in the feature space are sampled more often, which helped reducing covariate shift in down stream training.

- Key strength: density aware is principled and model agnostic, can also be extend to other distance based sampling schemes, e.g., coresets
- Weaknesses: the method requires density estimate, which is itself nontrivial and can be hard in higher dimensions. Assumption B1 is interesting in the sense that requires the conditional expectation of Y given X to be very predictive and Y has very little noise. This should not be hidden in the appendix.

**Audience:**

Yes

**Audience Explanation:**

Model agnostic method for data selection is useful where data labeling is costly.

**Claims And Evidence:**

Yes

**Claims Explanation:**

Fig.2 showed numerical results in one type of regression tasks that the proposed method is never worse than existing methods and sometimes better, which support the claim on the effectiveness of the proposal. In appendix the authors showed theoretical results.

**Requested Changes:**

I think the assumptions should be made clear, at least qualitatively in the main text. The authors should also make comments on that one needs to estimate the density and consequence when it is not estimated well.

---

> ### Author Response · Authors · 2025-11-04
> **We thank the reviewer for the time and clear feedback**
>
> Reviewer comment:
> - The assumptions should be made clear, at least qualitatively, in the main text. The authors should also comment on the need to estimate the density and the consequences when it is not estimated well.
>
> Response:
> - In Section 4, in the paragraph preceding Theorem 4.2, we clarified the assumptions qualitatively and added key equations to make both the mathematical conditions and their intuition explicit in the main text.
> In addition, we extended the conclusion section discussing the need to correctly estimate data density and potential consequences of poor estimation. We mention "A key aspect of DA-FPS is the need to estimate the data densities. Poor density estimation in DA-FPS can lead to ineffective weights that cause oversampling of high-density regions, generating redundant information in the selected set, and undersampling of low-density regions, leading to underrepresentation of important areas of the data space in the training set. This can lead to degraded model performance, for instance, through increased prediction errors in underrepresented regions." Note that we also discussed the effects of the choice of k on the DA-FPS performance at the end of section 5.3 and in the ablation study in Appendix M.

---

### Review · Reviewer_rsAe · 2025-10-29

**Summary Of Contributions:**

The paper introduces an approach for training data selection which is model-agnostic (i.e. makes no assumptions about the structure or parametrisation of the model) and is passive - particuarly suited for settings where labeled data is scarce / expensive to generate.

The presented idea is Density-Aware farthest point sampling which is an extension of Farthest point sampling, which adds a weighting term reflecting the relative concentration of labelled to unlabelled points.   Based on this, they bound the generalisation error that depends linearly on the WFD.   They propose a greedy algorithm, leveraging KDEs to approximate the distributions.  They show that the algorithm provides a 2k-approximation of the estimated WFD, where k is the nhd size of the KDE.

They demonstrate the efficacy of the approach on molecular property regression datasets, compared to a number of sensible baseline approaches.

**Additional Comments:**

None

**Audience:**

Yes

**Audience Explanation:**

The problem is of suitable generality and applicability across a wide range of domains.    The theory and generalisation bound is of indepdnent interest - with connections to coreset construction, and subset selection.

**Claims And Evidence:**

Yes

**Claims Explanation:**

Generaliation bound - well supported by proof.
DA-FPS algorithm - clear discussion how it is implemented, computational complexity & wall-clock run times.
Approximation guarantee for the estimate Weighted FPS
Empirical evidence on a number of challenging molecular regression datasets, and strong comparisons with baseline model-agnostic point selection processes.

**Requested Changes:**

There are no major changes required.

1. Making the the relationship between the theoretical bound and the implemented algorithm.  So thm 4.2 applies to the weighted fill distance, but DA-FPS uses a surrogate.    Clarify the gap between the theoretical bound and the implemented algorithm. Theorem 4.2 applies to the true weighted fill distance.   No need for a new theorem, but adding a short subsection or paragraph explicitly stating: (i) what is provably minimised, (ii) what remains conjectural about its relationship to the true weighted fill distance.

2. Can you explain the effect of the hyperparameter u.  Approximation theorem is provd for u=0, but then u>0 in the experiments.  Maybe just explicitly noting this, and adding some remark on what the authors expect to happen theorywise more generally.

3. Provide some heuristic guidance for k and u.

---

> ### Author Response · Authors · 2025-11-04
> **We thank the reviewer for the time and the useful comments**
>
> Response to 1:
> Concerning (i), DA-FPS attempts to minimize the estimated weighted fill distance defined in (11) with an approximation error bounded by Theorem 7.1. To make this point more explicit, we added a sentence at the beginning of section 7 stating "We establish that DA-FPS achieves a $2k$-approximation for the estimated weighted fill distance minimization problem in (10)."  Moreover, in section 7 we mention "Theorem 7.1 gives an upper bound on the approximation error of DA-FPS for the estimated weighted fill distance minimization problem defined in (10)."
>
> Regarding (ii), in Section 7 we now mention "It is important to note that the bound in Theorem 7.1 does not account for the quality of the density ratio estimation itself, which also depends on the parameter $k$. In other words, it does not quantify how well the estimated weighted fill distance in (11) relates to the true weighted fill distance in (2). Addressing this issue is an open problem for future work and is closely related to the broader field of density ratio estimation.
>
> The estimated weighted fill distance in (11) may not accurately reflect the true weighted fill distance in (2), particularly when the training set is small or the data are high-dimensional. According to multivariate density estimation theory~\citep{Wang2009}, limited sample sizes lead to biased density estimates in (5), which in turn cause the behavior the estimated weights $\omega^k_{\mathcal{L}\_{\mathcal{X}}}(\mathbf{x})$ to deviate from that of the true weights $\psi_{\mathcal{L}\_{\mathcal{X}}}(\mathbf{x})$ in (3). Consequently, DA-FPS may appear effective for the estimated problem in (10) but may fail to provide reliable performance for the true problem, especially for small training sets under high-dimensional settings where density estimation is challenging. To mitigate these limitations, we first sample attempting to ensure broad coverage of the data space by using the hyperparameter $u > 0$."
>
>
> Response to 2:
> In the revised version we now explicitly write: "We note that, the theoretical approximation guarantee in Theorem 7.1 applies only to $u=0$.  Nonetheless, using $u>0$ offers practical benefits: it ensures coverage across different regions of the data space before density-aware sampling begins, and it allows us to accumulate sufficient points to improve the reliability of density estimation needed for effective weight computation. The choice of $u$ is critical to DA-FPS effectiveness. For instance, if set too small, sparse regions may be underrepresented in the training set. This imbalance can result in low model performances in sparse regions of the dataset with a consequential increase in the average error. We speculate that setting $u >0$ may lead to improved approximation guarantees for DA-FPS with respect to minimizing the true weighted fill distance in (2). This is because the initial uniform sampling phase helps ensure that the selected set $\mathcal{L}_{\mathcal{X}}$ has a more balanced representation of the data space, which can lead to more accurate density estimates and weight computations in subsequent iterations. However, formalizing this intuition and deriving precise approximation bounds for DA-FPS with $u > 0$ remains an open problem for future research."
>
> Response to 3:
> We added a discussion at the end of Section 8 providing heuristic guidance for choosing the hyperparameters k and u. Among other things, we mention "... we think that an ``optimal'' choice $u$ depends on what the user wants to achieve. Qualitatively, if the user is interested in ensuring low MAXAE, at the cost of higher MAE, a larger $u$ is recommended. If the user is mainly interested in performing well on average in the region of high data density, a smaller $u$ is recommended. [...] we also analyze how changing the DA-FPS hyperparameter $k$ affects performance. [...] We think that the choice of $k$ should take into account at least two aspects: First, $k$ should not be too small, because it would lead to high variance in the values of the weights, and consequently, to noticeable differences in the values of weights associated with points lying in regions with similar true density. Second, $k$ should not be too large as the computational cost of DAFPS depends linearly on $k$."

---

### Author Response · Authors · 2025-11-04
**Summary of changes**

## Summary of changes

We thank all reviewers for their constructive feedback. Below we summarize the main changes made to address the concerns raised:


1. Notation and Presentation (2mgk): To improve readability, we added a comprehensive notation table after the conclusion. This table is referenced at the end of the introduction. We also added a short "roadmap" paragraph at the start of each section (starting from section 3) to guide the reader through the paper.


2. Clarity on definition of the weighted fill distance (2mgk): In the revised version we clarified that "The weighted fill distance is a deterministic quantity...". The revised version also includes a discussion of the case when $p_{\mathcal{X}\_{\mathcal{L}}} = p_{\mathcal{X}\_{\mathcal{D}}}$ pointwise and more generally on the implications of the weighted fill distance being zero (paragraph following Definition 4.1).

3. Tightness Theorem 4.2 (2mgk):  We enhanced the statement of Theorem 4.2 by applying a less coarse bound that multiplies training error and noise terms by a factor quantifying distributional mismatch. When $p_{\mathcal{X}\_{\mathcal{L}}} = p_{\mathcal{X}_{\mathcal{D}}}$, the bound now reduces to upper bounding the prediction error by itself, which is the best achievable bound in this scenario. We discussed these implications of the improved Theorem 4.2 in the paragraphs following it.


4. Reconciliation of Sampling distribution vs. active selection (2mgk): We clarified that "Our goal is to design a data sampling strategy, inducing a training data distribution marginal $p_{\mathcal{X}\_{\mathcal{L}}}$, to sample a set $\mathcal{L}_{\mathcal{X}}$ with minimal weighted fill distance."  (End of second paragraph after Theorem 4.2).


5. Discussion of algorithmic choices (2mgk): We expanded our discussion at the end of Section 5.3 regarding the role of k, role of u, and choice of metric. We mention "
 The behavior of the algorithm is primarily controlled by two hyperparameters: the neighborhood size $k$ for the $k$NN density estimation and the parameter $u$ for initial uniform selection. The parameter $k$ controls how we estimate density and directly affects the weight values $\omega^k_{\mathcal{L}_{\mathcal{X}}}(\mathbf{x})$. [...]The parameter $u$ determines the amount of points selected with uniform weight: setting $u = 0$ recovers pure density-aware selection from the start, while $u = b$ reduces to standard FPS. [...] Note that we are considering the Euclidean distance in our formulation of the (estimated) weighted fill distance. However, other distance metrics can be used depending on the data characteristics and the specific application. We follow common practice and use the Euclidean distance for general-purpose applications".

6. Heuristic guidance for hyperparameters k and u (rsAe): We added a discussion at the end of Section 8 providing heuristic guidance for choosing the hyperparameters k and u. Among other things, we mention "... we think that an ``optimal'' choice $u$ depends on what the user wants to achieve. Qualitatively, if the user is interested in ensuring low MAXAE, at the cost of higher MAE, a larger $u$ is recommended. If the user is mainly interested in performing well on average in the region of high data density, a smaller $u$ is recommended. [...] we also analyze how changing the DA-FPS hyperparameter $k$ affects performance. [...] We think that the choice of $k$ should take into account at least two aspects: First, $k$ should not be too small, because it would lead to high variance in the values of the weights, and consequently, to noticeable differences in the values of weights associated with points lying in regions with similar true density. Second, $k$ should not be too large as the computational cost of DAFPS depends linearly on $k$."

7. Clarity on assumptions (pVQd): In Section 4, we clarified assumptions in the paragraph preceding Theorem 4.2. We made the mathematical formulation and their intuition explicit in the main text.

8. Density estimation (pVQd): We extended the conclusion section discussing the need to correctly estimate data density and potential consequences of poor estimation. We mention "A key aspect of DA-FPS is the need to estimate the data densities. Poor density estimation in DA-FPS can lead to ineffective weights that cause oversampling of high-density regions, generating redundant information in the selected set, and undersampling of low-density regions, leading to underrepresentation of important areas of the data space in the training set. This can lead to degraded model performance, for instance, through increased prediction errors in underrepresented regions."

---

### Decision · Action_Editor_VMic · 2026-01-26

**Recommendation:** Accept as is

**Audience:**

Yes

**Audience Explanation:**

The paper would be of interest to application areas where labels are expensive to obtain. Broaderly, areas such as active learning and experimental design consider different but closely related problems, for which comparisons of the problem setups and approaches could be worthy for future considerations.

**Claims And Evidence:**

Yes

**Claims Explanation:**

This paper considers the problem of subsampling from a fixed set of unlabeled training examples for labeling, to minimize the expected performance on underlying data distribution the set was drawn from. The paper develops an upper bound on the prediction error based on the weighted fill distance, and propose the Density-Aware Farthest Point Sampling (DA-FPS) algorithm for subsampling based on density estimation and weighted fill distance. Experiments verify the effectiveness of the algorithm against baselines such as FPS. After rebuttal, all reviewers are positive about the contributions of the paper.